# GPROF-NN: a neural-network-based implementation of the Goddard Profiling Algorithm

**Simon Pfreundschuh**[1,2]**, Paula J. Brown**[2]**, Christian D. Kummerow**[2]**, Patrick Eriksson**[1]**, and Teodor Norrestad**[3,a]

[1]Department of Space, Earth and Environment, Chalmers University of Technology, 41296 Gothenburg, Sweden
[2]Department of Atmospheric Science, Colorado State University, Fort Collins, CO 80523, United States of America
[3]independent researcher
[a]formerly at: Department of Space, Earth and Environment, Chalmers University of Technology, 41296 Gothenburg, Sweden

**Correspondence:** Simon Pfreundschuh (simon.pfreundschuh@chalmers.se)

**Abstract.** The Global Precipitation Measurement (GPM) mission measures global precipitation at a temporal resolution of a few hours to enable close monitoring of the global hydrological cycle. GPM achieves this by combining observations from a spaceborne precipitation radar, a constellation of passive microwave (PMW) sensors, and geostationary satellites. The Goddard Profiling Algorithm (GPROF) is used operationally to retrieve precipitation from all PMW sensors of the GPM constellation. Since the resulting precipitation rates serve as input for many of the level 3 retrieval products, GPROF constitutes an essential component of the GPM processing pipeline.

This study investigates ways to improve GPROF using modern machine learning methods. We present two neural-network-based, probabilistic implementations of GPROF: GPROF-NN 1D, which (just like the current GPROF implementation) processes pixels individually, and GPROF-NN 3D, which employs a convolutional neural network to incorporate structural information into the retrieval. The accuracy of the retrievals is evaluated using a test dataset consistent with the data used in the development of the GPROF and GPROF-NN retrievals. This allows for assessing the accuracy of the retrieval method isolated from the representativeness of the training data, which remains a major source of uncertainty in the development of precipitation retrievals. Despite using the same input information as GPROF, the GPROF-NN 1D retrieval improves the accuracy of the retrieved surface precipitation for the GPM Microwave Imager (GMI) from 0.079 to 0.059 mm h$^{-1}$ in terms of mean abso-

lute error (MAE), from 76.1 % to 69.5 % in terms of symmetric mean absolute percentage error (SMAPE) and from 0.797 to 0.847 in terms of correlation. The improvements for the Microwave Humidity Sounder (MHS) are from 0.085 to 0.061 mm h$^{-1}$ in terms of MAE, from 81 % to 70.1 % for SMAPE, and from 0.724 to 0.804 in terms of correlation. Comparable improvements are found for the retrieved hydrometeor profiles and their column integrals, as well as the detection of precipitation. Moreover, the ability of the retrievals to resolve small-scale variability is improved by more than 40 % for GMI and 29 % for MHS. The GPROF-NN 3D retrieval further improves the MAE to 0.043 mm h$^{-1}$; the SMAPE to 48.67 %; and the correlation to 0.897 for GMI and 0.043 mm h$^{-1}$, 63.42 %, and 0.83 for MHS.

Application of the retrievals to GMI observations of Hurricane Harvey shows moderate improvements when compared to co-located GPM-combined and ground-based radar measurements indicating that the improvements at least partially carry over to assessment against independent measurements. Similar retrievals for MHS do not show equally clear improvements, leaving the validation against independent measurements for future investigation.

Both GPROF-NN algorithms make use of the same input and output data as the original GPROF algorithm and thus may replace the current implementation in a future update of the GPM processing pipeline. Despite their superior accuracy, the single-core runtime required for the operational processing of an orbit of observations is lower than that of GPROF. The GPROF-NN algorithms promise to be a simple

and cost-efficient way to increase the accuracy of the PMW precipitation retrievals of the GPM constellation and thus improve the monitoring of the global hydrological cycle.

# 1 Introduction

The Goddard Profiling Algorithm (GPROF, Kummerow et al., 2015) is the operational precipitation retrieval algorithm for the passive microwave (PMW) observations from the radiometer constellation of the Global Precipitation Measurement (GPM, Hou et al., 2014) mission, whose objective is to provide consistent global measurements of precipitation at a temporal resolution of a few hours. The precipitation retrieved by GPROF serves as input for the Integrated Multi-Satellite Retrievals for GPM (IMERG, Huffman et al., 2020), which can be considered the state of the art of global precipitation measurements. The algorithm thus constitutes an essential component of the global observation system that enables monitoring of the hydrological cycle for the benefit of science and society.

The development of GPROF was originally motivated by the Tropical Rainfall Measurement Mission (TRMM, Simpson et al., 1996), the precursor of the GPM mission, and thus dates back almost 30 years (Kummerow and Giglio, 1994b, c). Due to the conceptual and computational complexity of simulating PMW observations of clouds and precipitation, the algorithm was and remains based on a retrieval database consisting of observations and corresponding profiles of hydrometeors and precipitation rates. Nonetheless, the algorithm has undergone several updates since its conception. Methodologically, the most fundamental modification was the introduction of the Bayesian retrieval scheme in Kummerow et al. (1996), which is used in the algorithm up to the present. Following this, algorithm updates were mostly focused on improving the retrieval database and the incorporation of ancillary data into the retrieval. While the first version of GPROF still used handcrafted hydrometeor profiles to generate the retrieval database, these were soon replaced by profiles from a mesoscale weather model (Kummerow et al., 1996). An important improvement was the replacement of the model-derived database by an observationally generated database for the GPROF 2010 algorithm (Kummerow et al., 2011, 2015), which helped reduce errors caused by misrepresentation of atmospheric states in the database. The 2014 version of GPROF (Kummerow et al., 2015) introduced the first fully parametric version of the algorithm, which was designed to be applicable to all sensors of the GPM constellation. This version of GPROF became the operational PMW precipitation retrieval of the GPM mission.

This study focuses on the computational method that is used to produce the retrieval results from the retrieval database used by GPROF. Since its introduction in Kummerow et al. (1996), the currently used Bayesian method

has not received much consideration, mainly because the database and the incorporation of ancillary data were deemed to be more relevant for improving the accuracy of the retrieval. However, two disadvantages of the current retrieval method have become apparent with the introduction of the much larger, observationally generated retrieval databases into the algorithm (Elsaesser and Kummerow, 2015): first, the retrieval database must be compressed into self-similar clusters to reduce the processing time. This lossy compression may limit the extent to which the current algorithm can benefit from the size and representativeness of observationally generated retrieval databases. This is expected to affect retrievals of high rain rates due to their scarcity in the retrieval database. Second, the accuracy of the retrieval results depends on the uncertainties assigned to the database observations. Since there is no principled way to calculate these uncertainties, they need to be tuned heuristically for each sensor.

While GPROF is currently based on a data-driven method to solve Bayesian inverse problems, more general machine learning techniques have recently gained popularity for application in precipitation retrievals. Deep neural networks (DNNs), which have enabled a number of significant breakthroughs in different scientific fields (Silver et al., 2016; Jumper et al., 2021), have in recent years been explored for retrieving precipitation from satellite observations. Convolutional neural networks (CNNs) are especially appealing for this application because of their ability to leverage spatial patterns in image data. This property sets them apart from traditional retrieval methods and shallow machine-learning techniques, which are limited in their ability to use this information by computational complexity (Duncan et al., 2019) or the need for feature engineering or manual incorporation of spatial information through techniques such as convective–stratiform discrimination (Gopalan et al., 2010).

Shallow neural networks have long been used to retrieve precipitation from PMW observations (Staelin and Chen, 2000; Surussavadee and Staelin, 2008). The Passive microwave Neural network Precipitation Retrieval (PNPR) presented in Sanò et al. (2015, 2016, 2018) and the work by Tang et al. (2018) are among the more recent algorithms that use neural networks for retrieving precipitation from PMW observations. They employ relatively shallow neural networks and retrieve precipitation in a pixel-wise manner, thus neglecting the spatial structure in the observations. Other recent work demonstrates the ability of CNNs to leverage spatial information in satellite observations. Examples of this are IR-based retrievals by Sadeghi et al. (2019), PMW-based precipitation detection (Li et al., 2021), and retrievals combining PMW with IR observations (Gorooh et al., 2022) and gauge measurements (Moraux et al., 2019).

A shortcoming of the aforementioned studies is that none of them addresses the inherent uncertainty of the precipitation retrievals. Retrieving precipitation from PMW observations constitutes an inverse problem, whose ill-posed char-

acter leads to significant uncertainties in the retrieval results. Traditionally, these uncertainties are handled using Bayesian statistics. However, because the algorithms mentioned above neglect the probabilistic character of the retrieval, there is no way to reconcile them with the Bayesian approach.

Moreover, existing precipitation retrievals that make use of DNNs (Moraux et al., 2019; Sadeghi et al., 2019; Li et al., 2021; Gorooh et al., 2022) are experimental retrievals that are currently not used operationally. The design of an operational retrieval algorithm for the GPM PMW observations needs to address a number of additional requirements, such as the handling of observations from different sensors and the retrieval of multiple output variables. Furthermore, because GPM is an ongoing mission, continuity of the output variables must be ensured, which further constrains the design of the retrieval algorithm.

This study explores the use of DNNs for the operational retrieval of precipitation rates and hydrometeor profiles from the PMW observations from the GPM constellation. To this end, we present two PMW precipitation retrieval algorithms that provide probabilistic precipitation estimates and can be used in the operational processing pipeline for the GPM PMW observations.

**GPROF-NN 1D** This algorithm uses a fully connected neural network to retrieve single-column hydrometeor profiles and rain rates based on the observed brightness temperature vector. It thus uses exactly the same input data as the standard GPROF algorithm.

**GPROF-NN 3D** This algorithm extends the GPROF-NN 1D algorithm by incorporating spatial information into the retrieval using a CNN. It produces the same output as GPROF and GPROF-NN 1D but processes all observations simultaneously, thus allowing the algorithm to combine information from pixels across the swath.

The proposed algorithms are based on quantile regression neural networks (QRNNs, Pfreundschuh et al., 2018), which can be used to predict the posterior distribution of a Bayesian solution of the retrieval, given that the assumed a priori distribution of the Bayesian solution is the same as the distribution of the neural network's training data. Because of this, the GPROF-NN retrievals can produce all of GPROF's retrieval outputs, which include a probability of precipitation and an uncertainty estimate of the predicted precipitation in the form of terciles of the posterior distribution.

Before a retrieval can replace the current operational version of GPROF, it is imperative to establish its ability to improve the retrieval accuracy to avoid degradation of the GPM data products. A balanced evaluation of the accuracy of precipitation retrievals is difficult because it depends on the statistics of the data used in the assessment. Data-driven retrievals generally yield the most accurate results when evaluated on data with the same distribution as the data used for their training. At the same time, evaluation against independent measurements may distort the evaluation when these measurements deviate significantly from the training data. In this study, the retrieval performance of the GPROF-NN algorithms is evaluated and compared to that of GPROF using a held-out part of the retrieval database. This provides the most direct estimate of the benefits of the neural-network-based retrievals because it avoids the distorting effects of using test data from a different origin. Moreover, the nominal accuracy of both the GPROF and GPROF-NN algorithms provides a reference for future validation against independent measurements. More specifically, this study employs the newly developed GPROF-NN algorithms to answer the following two questions.

1. Can a DNN that uses the same input information as GPROF provide more accurate retrievals of surface precipitation and vertical hydrometeor profiles?

2. What is the potential of using a CNN to incorporate spatial information into the retrieval to further improve the accuracy of the retrievals within the current processing pipeline?

This study uses the upcoming version of the GPROF algorithm, GPROF 2021 (also known as GPROF V7, in the GPM Precipitation Processing System; NASA, 2021). The retrieval performance is assessed for two sensors of the GPM constellation: the GPM Microwave Imager (GMI) and the Microwave Humidity Sounder (MHS, Bonsignori, 2007). In addition to the evaluation against the data from the retrieval database, the study also presents a case study of the retrieved surface precipitation from overpasses of both GMI and MHS, which are compared to reference measurements from the GPM combined product (CMB, Grecu et al., 2016) and ground-based radar measurements from the Multi-Radar Multi-Sensor (MRMS, Smith et al., 2016) product suite.

## 2 Data and methods

The GPROF-NN algorithms make use of the same data as the original GPROF algorithm. This data, which we refer to as the retrieval database, defines, for all three algorithms, the input for the retrieval and the precipitation and hydrometeor profiles that the retrieval aims to reproduce. Because of its fundamental importance for all retrievals considered here, this section provides an overview of the retrieval database. This is followed by a brief description of the current GPROF algorithm and the implementation of the GPROF-NN retrievals.

### 2.1 The retrieval database

The GPROF retrieval database is made up of pairs of retrieval input data and corresponding output. The input comprises PMW observations and ancillary data. The output consists of

**Table 1.** Retrieval quantities in the retrieval database.

| Retrieval variable | Unit | Type |
|---|---|---|
| Surface precipitation | $\mathrm{mm\,h^{-1}}$ | Scalar |
| Convective precipitation | $\mathrm{mm\,h^{-1}}$ | Scalar |
| Cloud water path | $\mathrm{kg\,m^{-2}}$ | Scalar |
| Rain water path | $\mathrm{kg\,m^{-2}}$ | Scalar |
| Ice water path | $\mathrm{kg\,m^{-2}}$ | Scalar |
| Cloud water content | $\mathrm{g\,m^{-3}}$ | Profile |
| Rain water content | $\mathrm{g\,m^{-3}}$ | Profile |
| Snow water content | $\mathrm{g\,m^{-3}}$ | Profile |
| Latent heating | $\mathrm{K\,h^{-1}}$ | Profile |

**Table 2.** Channels of the GMI and MHS sensors used for the retrievals in this study.

| Channel | Freq. (GHz) | Pol. |
|---|---|---|
| GMI-1 | 10.6 | V |
| GMI-2 | 10.6 | H |
| GMI-3 | 18.7 | V |
| GMI-4 | 18.7 | H |
| GMI-5 | 23 | V |
| GMI-6 | 37 | V |
| GMI-7 | 37 | H |
| GMI-8 | 89 | V |
| GMI-9 | 89 | H |
| GMI-10 | 166 | V |
| GMI-11 | 166 | H |
| GMI-12 | $183 \pm 3$ | V |
| GMI-13 | $183 \pm 7$ | V |

| Sensor | Freq. (GHz) | Pol. |
|---|---|---|
| MHS-1 | 89 | V |
| MHS-2 | 157 | V |
| MHS-3 | $183 \pm 1$ | H |
| MHS-4 | $183 \pm 3$ | H |
| MHS-5 | 190.31 | V |

the values of the retrieved quantities. GPROF's retrieval outputs include surface precipitation; profiles and path integrals of rain, snow, and cloud water; and latent heating profiles. A listing of all retrieval targets and corresponding units is provided in Table 1.

Since the available channels and the viewing geometries vary between the sensors of the GPM constellation, a separate database is generated for each sensor type. A crucial difference between the retrieval databases for GMI and the other sensors of the GPM constellation is that the database for GMI uses real observations, whereas the databases for the other sensors are constructed using simulations. The varying resolutions and viewing geometries of different sensors are taken into account by resampling and averaging the simulated observations and retrieval results to the observation footprints of the corresponding sensor. The channels of the GMI and MHS sensors that are used in this study are listed in Table 2.

The databases for GPROF 2021 are derived from 1 year (October 2018 to September 2019) of retrieved hydrometeor profiles from the GPM CMB product (Grecu et al., 2016). This data are complemented with surface precipitation from the currently operational Microwave Integrated Retrieval System (Boukabara et al., 2011), which adds light precipitation in areas where no echo is detected by the GPM Dual-Frequency Precipitation Radar. Observations over sea ice and snow-covered surfaces are handled separately. For sea ice, precipitation is derived from the ERA5 reanalysis (Hersbach et al., 2020). For snow-covered surfaces, precipitation is derived from several years of co-locations with gauge-corrected radar measurements from MRMS (Smith et al., 2016).

The ancillary data that serve as additional retrieval inputs are derived from reanalysis datasets. They consist of 2 m temperature ($T_{2\,\mathrm{m}}$), total column water vapor (TCWV), the surface type, and an air lifting index (ALI) that encodes information on atmospheric convergence in mountainous areas. The ancillary data for the databases used in this study were derived from the ERA5 reanalysis (Hersbach et al., 2020).

A detailed description of the retrieval database and the derivation of the data it contains can be found in the GPROF

Algorithm Theoretical Basis Document (ATBD) (Passive Microwave Algorithm Team Facility, 2022). The training data for the GPROF-NN retrievals consists of the data from the retrieval database. The training data are stored in an intermediate format to simplify the loading of the data during training of the neural network. The format and the creation process of the training data are both described in detail in Sect. B1 in Appendix B.

## 2.2 The GPROF algorithm

The current implementation of GPROF uses a Bayesian scheme to retrieve precipitation and hydrometeor profiles, which works by resampling the profiles in the database based on the similarity of the observations and ancillary data. GPROF uses ancillary data to split the database into separate bins. This reduces the number of profiles for which weights must be computed and helps to constrain the retrieval. Moreover, the profiles in each bin are clustered to limit the number of profiles that need to be processed. A detailed description of the implementation of GPROF is provided in Appendix A.

## 2.3 The GPROF-NN algorithms

The principal objective guiding the design of the GPROF-NN algorithms was to develop a neural-network-based retrieval that operates on the same input data and provides the same output as GPROF so that it can replace the current implementation in a future update. Although GPROF's retrieval scheme is defined on independent pixels, the algorithm pro-

cesses full orbits of observations and corresponding ancillary data. Both GPROF-NN retrievals were therefore designed to process the same input format as GPROF, which corresponds to each sensor's level 1C observations in their native spatial sampling, which is remapped to a common grid where required. The output from all retrievals is on the same grid as the input.

GPROF produces multiple probabilistic outputs: a probability of precipitation and the mode and terciles of the posterior distribution of precipitation. An implementation based on standard regression neural networks would not provide any principled way to produce these probabilistic outputs due to the incompatibility of deterministic regression with the Bayesian retrieval formulation used in GPROF. The implementation of the GPROF-NN retrievals uses quantile regression neural networks (QRNNs) to overcome this limitation. As shown in Pfreundschuh et al. (2018), when trained on a dataset distributed according to the a priori distribution of a Bayesian retrieval, QRNNs learn to predict quantiles of the Bayesian posterior distribution. They thus provide a simple and efficient way to reconcile neural network retrievals with the Bayesian framework employed by GPROF.

QRNNs predict a sequence of quantiles, which allows reconstructing the cumulative distribution function (CDF) of the a posteriori distribution of any scalar retrieval quantity. Since the distribution of a scalar variable is fully described by its CDF, any relevant statistic of the a posteriori distribution can be derived from the predicted CDF. The GPROF-NN retrievals use the predicted CDF to derive the most likely and mean surface precipitation (the latter of which is identical to the solution that would have been obtained with standard mean squared error regression), the terciles of the posterior distribution, and the probability of precipitation. Figure 1 illustrates the principle of the GPROF-NN retrievals: the retrieval employs a neural network to predict a vector of values for each pixel in the input observations. The elements of this vector correspond to a sequence of quantiles of the a posteriori distribution. These quantiles are used to reconstruct a piece-wise linear approximation of the CDF of the distribution from which the retrieval results are derived.

### 2.3.1 Training objectives

A neural network can be trained to predict a quantile $\hat{x}_\tau$ of a given conditional distribution by training it to minimize the quantile loss function $\mathcal{L}_\tau$ corresponding to the quantile fraction $\tau$ (Koenker and Hallock, 2001):

$$\mathcal{L}_\tau(\hat{x}_\tau, x) = \left(\tau - \mathbb{I}_{x \leq \hat{x}_\tau}\right)\left(x - \hat{x}_\tau\right), \tag{1}$$

where $\hat{x}_\tau$ is the predicted quantile, $x$ is the reference value from the training data, and $\mathbb{I}_{x \leq \hat{x}}$ is the indicator function taking the value 1 when the condition $x \leq \hat{x}$ is true and 0 otherwise.

This principle can be extended to a sequence of quantiles corresponding to quantile fractions $\tau_1, \ldots, \tau_N$ by minimizing the mean of the loss functions corresponding to each quantile fraction:

$$\mathcal{L}_{\tau_1, \ldots, \tau_N}(\hat{\boldsymbol{x}}, x) = \frac{1}{N} \sum_{i=0}^{N} \mathcal{L}_{\tau_i}(\hat{x}_i, x), \tag{2}$$

where $\hat{x}_i$ is the $i$th component of the vector of predicted quantiles $\hat{\boldsymbol{x}}$. The GPROF-NN retrievals use this loss function with 128 equally spaced quantiles ranging from $\tau_1 = 0.001$ to $\tau_{128} = 0.999$ for all scalar retrieval variables.

A difficulty with predicting quantiles of precipitation is that lower quantiles may become degenerate due to the high probability of no precipitation. For example, it is impossible to predict empirical quantiles with $0 < \tau < 0.5$ for a pixel with 50 % probability of precipitation. To allow monitoring of the ability of the network to correctly predict retrieval uncertainty up to the degeneracy induced by non-raining pixels, we replace rain rates of non-raining pixels with random values from a log-uniform distribution that are smaller than the smallest rain rate in the training data. During the retrieval, predicted precipitation rates that are smaller than this threshold are set to zero. The threshold is chosen as $10^{-4}\,\mathrm{mm\,h^{-1}}$ and thus has negligible impact on mean or accumulated precipitation.

An additional advantage of the application of the quantile loss function is that the training can be performed on transformed retrieval outputs without changing the statistical properties of the network predictions given that the transformation function is strictly monotonic. The training of all scalar, non-negative retrieval quantities uses a log-linear transformation function of the form

$$f(x) = \begin{cases} \log(x) & \text{if } x < 1 \\ x - 1 & \text{if otherwise.} \end{cases} \tag{3}$$

In addition to avoiding the prediction of negative values, we found this to slightly increase retrieval accuracy for quantities that vary by multiple orders of magnitude, which precipitation rates and hydrometeor concentrations typically do.

For hydrometeor profiles, the retrieval is implemented in a slightly different manner. To reduce the number of network outputs, the posterior mean of hydrometeor profiles is predicted directly using mean squared error regression. Since the output of GPROF contains only the posterior mean of the hydrometeor concentrations, it was deemed unnecessary to predict their full posterior distribution at each level using quantile regression. To avoid the prediction of negative concentrations, rectified linear unit (ReLU) activation functions are applied to the network outputs corresponding to hydrometeor concentrations.

### 2.3.2 GPROF-NN 1D

The GPROF-NN 1D retrieval was designed to use the same input information and produce the same output as the

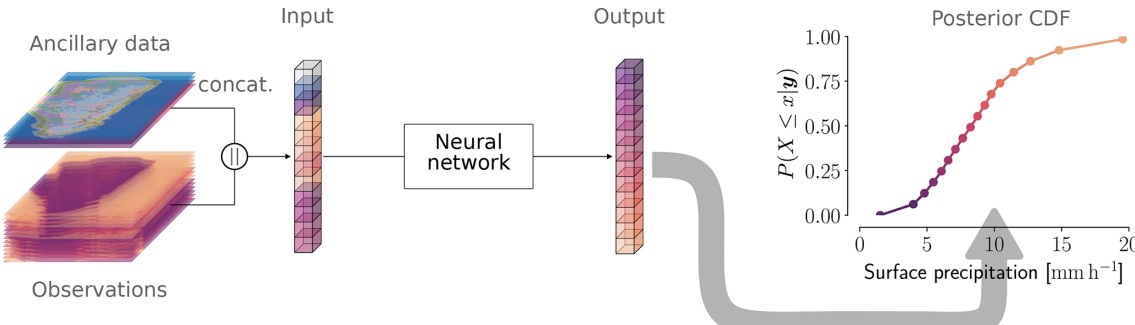

**Figure 1.** The basic principle of the implementation of the GPROF-NN retrievals. A Bayesian solution of the retrieval is obtained by predicting, for each input pixel, a sequence of quantiles of the a posteriori distribution that is used to reconstruct its CDF. The predicted CDF is then used to derive the scalar retrieval results.

Bayesian scheme used by GPROF. GPROF-NN 1D thus operates on single pixels of brightness temperatures and ancillary data and predicts the corresponding precipitation and hydrometeor profiles.

The neural network architecture used for the GPROF-NN 1D retrieval is illustrated in Fig. 2. A single network is trained to predict all retrieval variables (see Table 1) using the training objectives described in Sect. 2.3.1. The network consists of a shared body and a separate head for each retrieved variable. Bodies and heads are built-up of blocks consisting of a fully connected layer followed by layer normalization (Ba et al., 2016) and Gaussian error linear unit (GELU) (Hendrycks and Gimpel, 2016) activation functions. During development we have experimented with different numbers of blocks in each body ($N_b$) and each of the heads ($N_h$) but found only a marginal impact on the retrieval performance, and we settled for a configuration with $N_b = 6$ and $H_h = 4$.

Detailed descriptions of the neural network training and the retrieval processing for GPROF-NN 1D are provided in Sects. B2 and. B3, respectively.

### 2.3.3   GPROF-NN 3D

The GPROF-NN 3D retrieval extends the GPROF and GPROF-NN 1D algorithms by incorporating structural information into the retrieval. To achieve this, the GPROF-NN 3D algorithm employs a CNN that performs the retrieval for all pixels in the swath simultaneously.

The network architecture for the GPROF-NN 3D algorithm, illustrated in Fig. 3, consists of an asymmetric encoder–decoder structure followed by a separate head for each retrieved variable. The stages of the encoder and decoder are built up of what we refer to here as Xception blocks (Fig. 3a) because they are based on the Xception architecture introduced in Chollet (2017). Each block consists of two depth-wise separable convolutions with a kernel size of 3 followed by group normalization layers with 32 groups and GELU activation functions. The first block in each stage of the encoder additionally contains a $3 \times 3$ max-pooling layer

with a stride of 2 following the first $3 \times 3$ convolution layer. Each downsampling block in the encoder is followed by $N = 4$ standard Xception blocks. The stages of the decoder consist of a bilinear up-sampling layer followed by a single Xception block. The network architecture was chosen with the aim of maximizing the depth and width of the network while keeping the time required for processing an orbit low. Symmetric padding is performed before all convolution operations with a kernel size larger than one in order to conserve the input size.

Additional complexity in the training of the GPROF-NN 3D retrieval derives from the requirement to operate on the same data as GPROF, which means that input and output data must be on the native observation grid of each sensor. This is problematic because the viewing geometries of PMW sensors break the translational symmetry of digital images that constitutes one of the inductive biases of CNNs (Goodfellow et al., 2016). For example, geolocated pixels of conical scanners do not lie on a rectangular grid, which causes shapes to appear differently depending on their position in the swath. Figure 4 illustrates this for GMI observations. The rectangular shapes shown in Fig. 4a are distorted when the observations are plotted on a uniform grid (Fig. 4b).

Moreover, because GPROF currently only uses the central 21 pixels of the CMB product for the generation of the retrieval database, the values of the retrieval targets in the GPROF database are known only at the central pixels of the GMI swath. The location of these pixels is marked by the light stripe in Fig. 4. The neural network can thus learn the spatial structure of precipitation only from the central part of the GMI observations.

The training of the GPROF-NN 3D retrieval employs a customized data augmentation scheme to account for the aforementioned characteristics of the training data. Training samples for the GPROF-NN 3D retrieval are transformed to simulate the effect of observing each training scene at varying locations of the sensor swath. The transformations are applied randomly when a training sample is loaded, thus ensuring that the network rarely or never sees a training scene

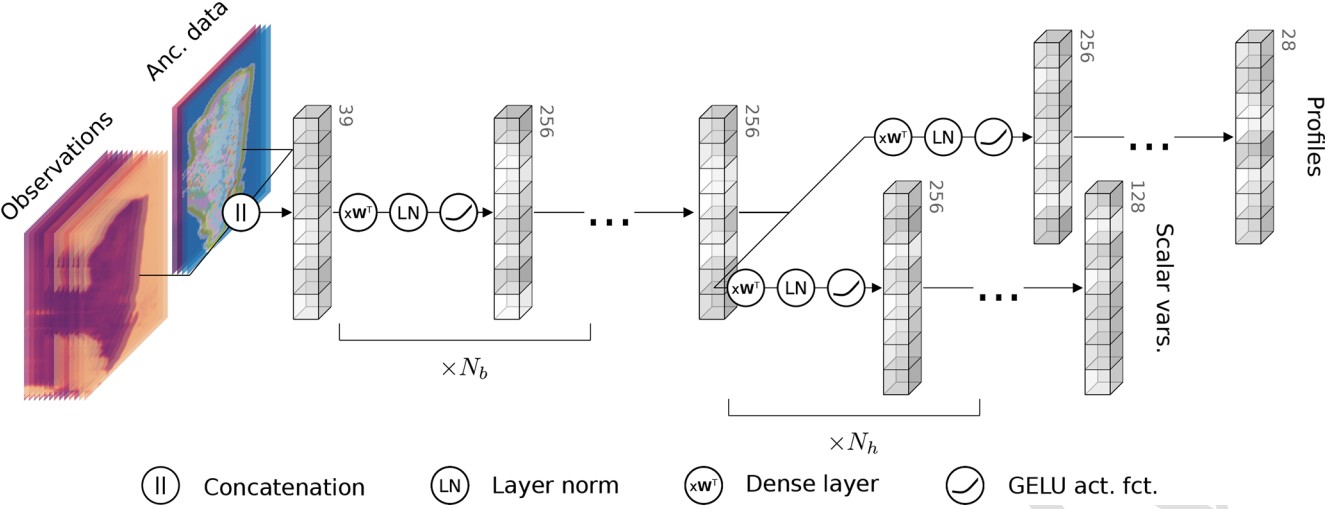

**Figure 2.** Illustration of the neural network architecture used in the GPROF-NN 1D algorithm. The network consists of a common body and one head for each retrieval variable. Each block in body and head consists of a fully connected layer, layer norm, and GELU activation function.

## (a) Xception block

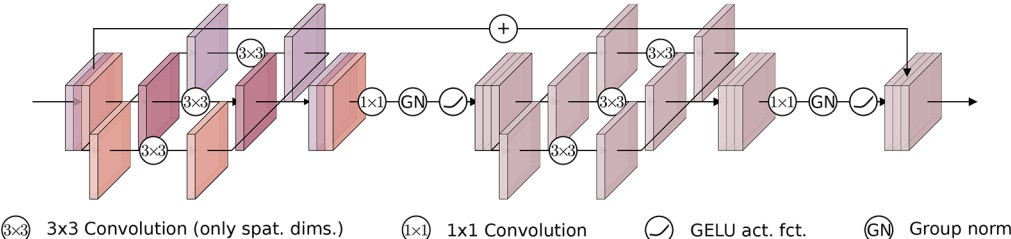

## (b) GPROF-NN 3D architecture

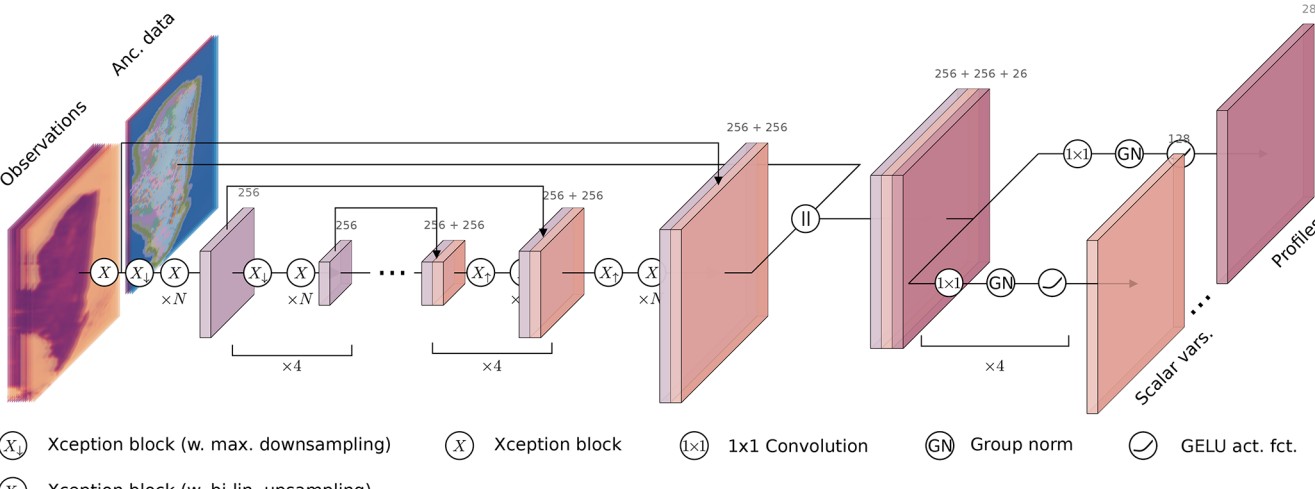

**Figure 3.** The neural network architecture of the GPROF-NN 3D retrieval. Panel **(a)** displays the structure of the Xception blocks (Chollet, 2017) that form the building blocks of the GPROF-NN 3D model. An Xception block consists of two depth-wise separable convolutions followed by a group normalization layer and a GELU activation function. Panel **(b)** shows how the Xception blocks are used in an asymmetric encoder-decoder structure that forms the body of the network. Output from the body is combined with the ancillary data to form the inputs to the separate heads that predict the retrieval results for each of the retrieved variables.

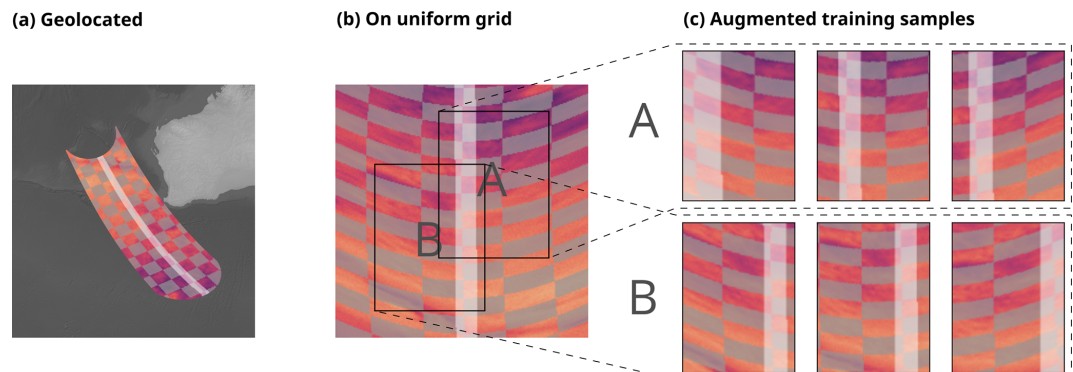

**Figure 4.** The effect of GMI's conical viewing geometry on observed features. Panel **(a)** displays geolocated observations of the 10.6 GHz channel (colored background). Grey squares mark equilateral shapes with a side length of 200 km oriented along the swath. The highlighted stripe located at the swath center marks the region where the values of the retrieved variables are known. Panel **(b)** shows the same observations viewed as an image on a uniform grid. Panel **(c)** shows six synthetically generated training inputs based on two input regions marked in panel **(b)**. The first row shows three synthetic samples that simulate the effect of viewing the input in region A at a different position across the GMI swath. The second row shows the corresponding transformations for the input in region B.

from the same perspective. The transformations also vary the relative location of the pixels at which values of retrieval variables are known across the full width of the swath instead of always being located at its center. Examples of transformed inputs for GMI are displayed in Fig. 4c.

A detailed description of the training and the retrieval processing for the GPROF-NN 3D retrieval are provided in Sects. B2 and B3, respectively.

### 2.3.4 Extension to other sensors

The GPROF retrieval for GMI is special because it is the only sensor of the GPM constellation for which the retrieval inputs used in the database correspond to real observations. For the other sensors, the observations used to construct the retrieval database are simulated. Since the simulations take into account the effect of the different viewing geometries and resolutions, GPROF-NN 1D inherits its ability to handle observations from different sensors directly from the design of retrieval database.

For the GPROF-NN 3D algorithm this is not the case. The problem for sensors other than GMI is that the retrieval database contains simulated observations only at the central pixels of the GMI swath (the highlighted pixels in Fig. 4a). To obtain two-dimensional training scenes that are sufficiently wide to train a CNN, we make use of an intermediate CNN-based model to "retrieve" simulated brightness temperatures across the full GMI swath. The extended simulated brightness temperatures are then remapped from the GMI viewing geometry to the viewing geometry of the target sensor. While this approach is certainly not ideal with respect to the realism of the generated scenes, it was the simplest and currently only feasible way to extend the GPROF-NN 3D retrieval to other sensors than GMI using only currently available data from the GPROF database. A detailed description

of the procedures involved in generating the training data for different sensors is provided in Sect. B1 in Appendix B.

Moreover, since the databases for other sensors rely on simulations, it is not guaranteed that the distribution of brightness temperatures in the database matches those of actual observations. The simulations are therefore corrected using a surface type and total TCWV-dependent correction that matches the quantiles of the conditional distributions of simulated and real observations. The GPROF algorithm's correction distinguishes three different surface types. However, the GPROF-NN algorithms use a correction with all 18 surface types because the correction used by GPROF was found to be too crude over land surfaces leading to artifacts in the retrieval results.

## 3   Results

This section presents the results of the evaluation of GPROF and the novel GPROF-NN algorithms. The first part evaluates the retrievals using a held-out test dataset. The remainder of this section presents a case study of precipitation retrievals from Hurricane Harvey followed by a brief assessment of the processing times of the different algorithms.

### 3.1   Assessment on held-out test data

The held-out test data comprise observations from the retrieval database from the first 3 d of every month. These data have not been used for training the neural network retrievals. It is, however, derived from the same data sources and thus stems from the same distribution as the training data.

Table 3 lists the number of pixels with precipitation information used for testing the retrievals. The evaluation of the GPROF-NN 3D retrieval uses spatially contiguous scenes of the same size as the ones used during its training. Since

**Table 3.** Number of pixels with precipitation information in the test datasets used to evaluate the retrievals.

| Sensor | GPROF and GPROF-NN 1D | GPROF-NN 3D |
|--------|----------------------|-------------|
| GMI    | 50 435 584           | 14 218 203  |
| MHS    | 24 975 877           | 4 945 165   |

these scenes typically do not cover all of the pixels with precipitation information, the test data for the GPROF-NN 3D retrievals contain fewer pixels that can be used for evaluation. The lower number of test pixels for MHS is due to the coarser resolution of the observations, which leads to a smaller number of observations over sea ice and snow and an additional reduction of the pixels available for evaluation of the GPROF-NN 3D retrieval.

### 3.1.1 Precipitation and hydrometeor paths

As described in Table 1, the scalar variables retrieved by GPROF are surface and convective precipitation and the column-integrated concentrations of cloud droplets, rain, and snow. They are denoted as cloud water path (CWP), rain water path (RWP), and ice water path (IWP), respectively. Scatter plots of the retrieval results for these five quantities evaluated over all surfaces are displayed in Fig. 5 for GMI and Fig. 6 for MHS. The frequencies in all plots have been normalized column-wise to ensure that results for high reference values remain visible.

Consistent improvements in the accuracy of the surface precipitation retrieved by GMI are observed between GPROF and GPROF-NN 1D and between GPROF-NN 1D and GPROF-NN 3D. The improvements are most pronounced for light rates between $10^{-2}$ and $10^{-1}$ mm h$^{-1}$ but are consistent across the full range of values. The comparably bad performance of GPROF for light precipitation is likely due to the tuning of the assigned uncertainties to yield good results for heavier rain that is more relevant for rainfall accumulation.

For convective precipitation, the results of GPROF deviate noticeably from the diagonal. The results of GPROF-NN 1D slightly improve upon those of GPROF. Although the mode of the distribution is still displaced from the diagonal, the GPROF-NN 3D algorithm yields the best agreement with the reference data. For the path-integrated quantities, similar improvements between GPROF and GPROF-NN 1D as well as GPROF-NN 1D and GPROF-NN 3D are observed. Large cloud water path values are underestimated by all retrievals, which is likely because these values are associated with precipitation but difficult to distinguish from it. Due to the lack of a cloud water path signal in raining profiles, all algorithms resort to predicting the climatology in the presence of significant rain.

The results for MHS, displayed in Fig. 6, paint a similar picture. Although the overall accuracy of all retrievals is lower than for GMI, GPROF-NN 1D consistently yields

more accurate results than GPROF. The GPROF-NN 3D retrieval also yields further consistent improvements compared to the GPROF-NN 1D retrieval.

Quantitative measures of the retrieval accuracy for surface precipitation of the three retrieval algorithms are displayed in Table 4 for GMI and Table 5 for MHS. Similar tables for the other retrieval quantities are provided in Tables C1–C8 in Appendix C. Each table displays bias, mean absolute error (MAE), mean squared error (MSE), the symmetric mean absolute percentage error (SMAPE$_t$) for all test samples with a reference value that exceeds a quantity-specific threshold $t$ and the correlation. The error metrics confirm the qualitative findings from Figs. 5 and 6: the neural network implementations outperform GPROF in terms of all considered metrics. Moreover, the GPROF-NN 3D algorithm further improves upon the performance of the GPROF-NN 1D algorithm. The same tendency is observed for MHS, albeit with lower overall accuracy.

Since the surface type has a considerable effect on the lower-frequency observations used in the retrieval, its impact on the retrieval of surface precipitation is assessed in Fig. 7. Figure 7 displays bias, MSE, MAE, SMAPE, and correlation for principal surface types. For the analysis, original GPROF surface types have been grouped into ocean (surface type 1), dense vegetation (surface types 3–5), sparse vegetation (6–7), snow (surface types 8–11), and coast (surface types 12–15). Even when the different surface types are considered separately, the results of the surface precipitation retrieval show the same pattern as the scatter plots in Fig. 5. The results of the GPROF-NN 1D retrieval are generally more accurate than those of GPROF, and the results of the GPROF-NN 3D algorithm are more accurate than those of the GPROF-NN 1D algorithm. These findings are mostly consistent across the considered surface types and both sensors. Exceptions are the biases of the GPROF-NN 1D algorithm for GMI over densely vegetated surfaces and for MHS over snow, which are larger than those of GPROF, and the MSE of the GPROF-NN 3D algorithm for MHS over snow, which is slightly larger than that of the GPROF-NN 1D retrieval. We suspect this is caused by the relative scarcity of the observations in the retrieval database.

Figure 8 displays the geographical distribution of bias, MSE, and SMAPE for GMI in $5° × 5°$ boxes. As could be expected from the previous results, the magnitudes of the biases of GPROF are considerably larger than for the other two algorithms. Furthermore, GPROF exhibits consistent biases across geographical regions such as the northwestern Atlantic and northwestern Pacific, which is less the case for the neural network algorithms. Although spatial distribution of the MSE mostly reflects the global distribution of precipitation, a gradual decrease in MSE can be observed between the results of GPROF and GPROF-NN 1D and between GPROF-NN 1D and GPROF-NN 3D. More consistent patterns are visible in the SMAPE: the largest errors for all three retrievals occur over land surfaces, which likely re-

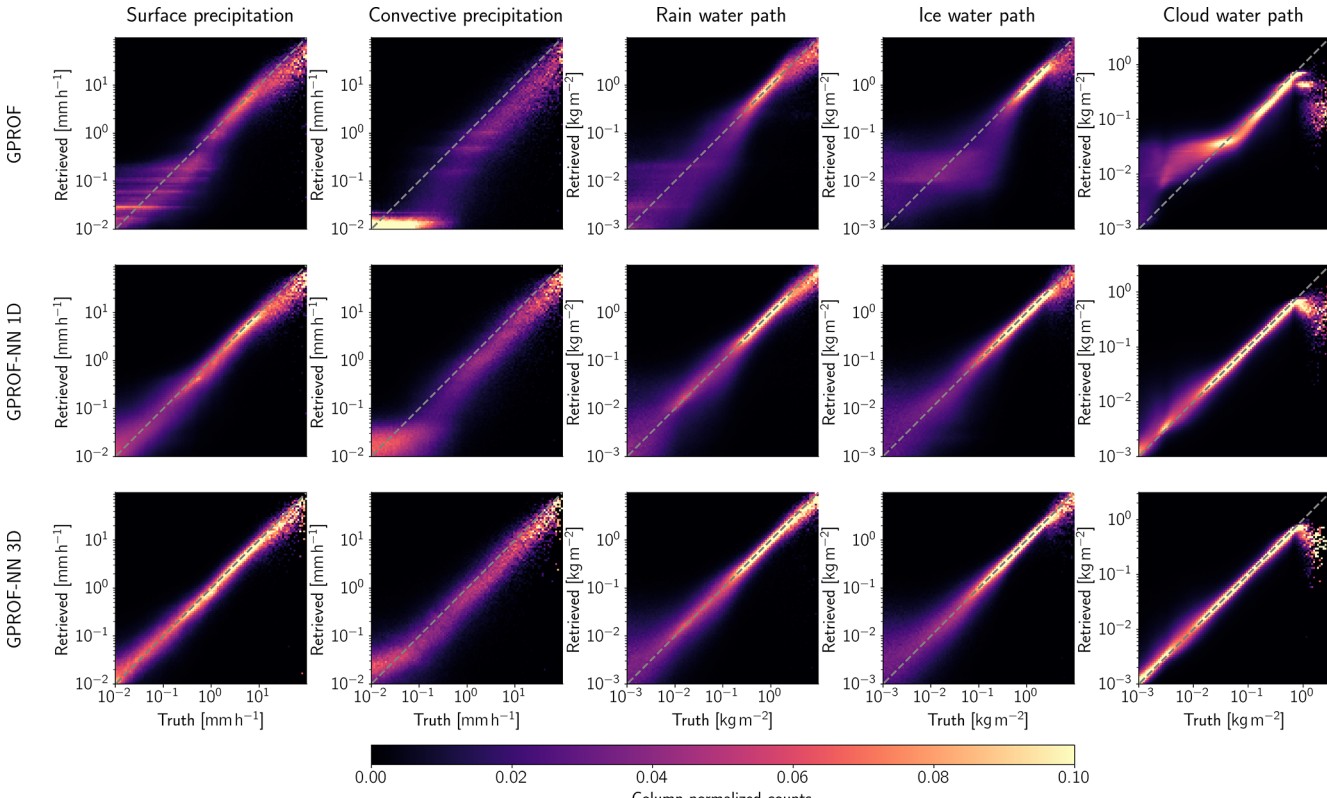

**Figure 5.** Scatter plots of scalar retrieval targets for the three retrieval algorithms for GMI. Rows display the results for the GPROF, GPROF-NN 1D, and GPROF-NN 3D algorithms, respectively. Columns display the results for different retrieval targets. Frequencies in the plots have been normalized column-wise, i.e., per bin of the reference value.

**Table 4.** Mean error metrics and estimated standard deviation for surface precipitation retrieved from GMI observations.

| Metric | GPROF | GPROF-NN 1D | GPROF-NN 3D |
|---|---|---|---|
| Bias [$\mathrm{mm\,h^{-1}}$] | $-0.0029 \pm 0.0001$ | $-0.0024 \pm 0.0001$ | $-0.0006 \pm 0.0001$ |
| MAE [$\mathrm{mm\,h^{-1}}$] | $0.0788 \pm 0.0001$ | $0.0585 \pm 0.0001$ | $0.0444 \pm 0.0001$ |
| MSE [$\mathrm{mm\,h^{-1}}$] | $0.1965 \pm 0.0001$ | $0.1379 \pm 0.0001$ | $0.0983 \pm 0.0001$ |
| $\mathrm{SMAPE_{0.01}}$ [%] | $76.0598 \pm 0.0139$ | $69.5382 \pm 0.0127$ | $56.0040 \pm 0.0181$ |
| Correlation | $0.7971$ | $0.8470$ | $0.8966$ |

flects the decrease in information content due to the reduced contrast in the lower-frequency channels. Over ocean, errors are generally higher in the sub-tropics and tropics compared to higher latitudes. Although these patterns are observed in the results of all algorithms, a clear and globally consistent decrease in SMAPE can be observed between the GPROF, GPROF-NN 1D, and GPROF-NN 3D retrievals.

The corresponding results for MHS are provided in Fig. C1. Again, although the errors are slightly larger, the results are qualitatively similar.

### 3.1.2 Predicted retrieval uncertainties and probabilistic rain detection

In addition to quantitative precipitation estimates, GPROF produces estimates of the first and second tercile of the posterior distribution of surface precipitation, which provide an uncertainty estimate for the retrieved mean surface precipitation, as well as a probabilistic classification of pixels into raining and non-raining pixels based on an estimated probability of precipitation. Due to their probabilistic nature, similar results can be produced using the GPROF-NN algorithms. From the predicted quantiles, the terciles can be inferred by interpolating them to the fractions $\frac{1}{3}$ and $\frac{2}{3}$, respectively. The probability of precipitation is calculated by using the pre-

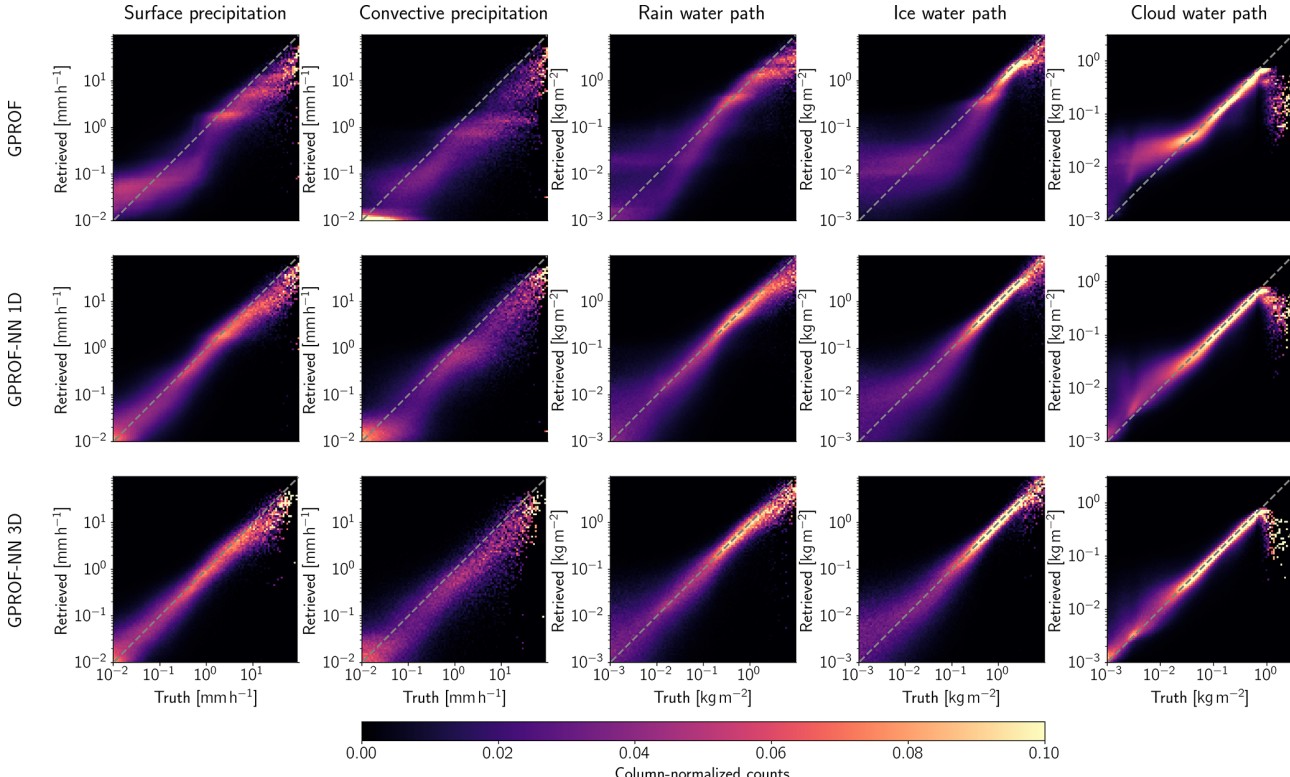

**Figure 6.** The same as Fig. 5 but for MHS.

**Table 5.** Mean error metrics and estimated standard deviation for surface precipitation retrieved from MHS observations.

| Metric | GPROF | GPROF-NN 1D | GPROF-NN 3D |
|---|---|---|---|
| Bias [$\mathrm{mm\,h^{-1}}$] | $-0.0110 \pm 0.0001$ | $-0.0066 \pm 0.0001$ | $-0.0018 \pm 0.0001$ |
| MAE [$\mathrm{mm\,h^{-1}}$] | $0.0846 \pm 0.0001$ | $0.0609 \pm 0.0001$ | $0.0487 \pm 0.0001$ |
| MSE [$\mathrm{mm\,h^{-1}}$] | $0.2317 \pm 0.0001$ | $0.1682 \pm 0.0001$ | $0.1087 \pm 0.0001$ |
| $\mathrm{SMAPE_{0.01}}$ [%] | $80.8641 \pm 0.0190$ | $68.4961 \pm 0.0185$ | $62.3086 \pm 0.0377$ |
| Correlation | $0.7239 \pm 0.0000$ | $0.8040 \pm 0.0000$ | $0.8400 \pm 0.0000$ |

dicted posterior distribution to calculate the probability of the retrieved surface precipitation to be larger than the smallest non-zero rain rate in the training data.

To assess the accuracy of the uncertainty estimates from GPROF and the GPROF-NN algorithms, Table 6 lists the calibration, i.e., the frequency with which each predicted tercile was larger than the true surface precipitation. The evaluation of the results for the GPROF-NN algorithms was performed with the replacement of non-raining values described in Sect. 2.3.1, which allows us to account for degenerate quantiles. Since no such mechanism is available for GPROF, it is not possible to evaluate the calibration of the predicted terciles without their effect. At least partially because of this, the calibration of GPROF deviates from the nominal frequencies. For the GPROF-NN algorithms, however, both algorithms yield frequencies that are close to the expected frequencies of $\frac{1}{3}$ and $\frac{2}{3}$, respectively.

The quality of the raining or non-raining classification is assessed in Fig. 9, which displays the calibration of the predicted probability and the receiver operating characteristic (ROC) curve. The predicted probabilities are fairly well calibrated for all algorithms and sensors. Nonetheless, the GPROF-NN algorithms yield results that are slightly closer to the diagonal. The results for the ROC curves are analogous: the GPROF-NN 1D algorithm yields better precipitation detection than GPROF and the GPROF-NN 3D retrieval in turn yields slightly better performance than the 1D version. In terms of classification skill, worse performance is achieved for GMI than for MHS by all algorithms, but again the relative performance of the retrievals is the same.

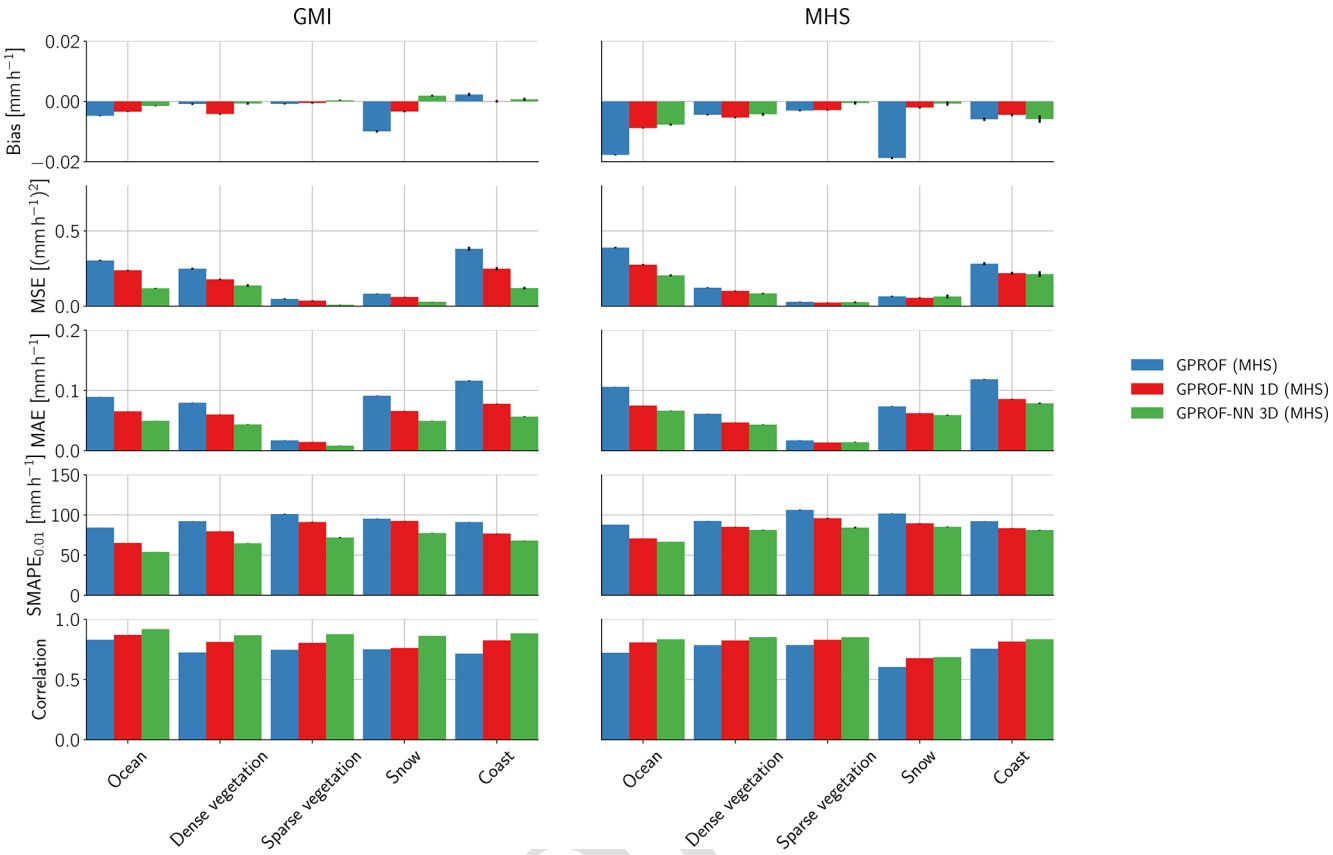

**Figure 7.** Bias, MSE, MAE, $SMAPE_{0.01}$, and correlation of the retrieved surface precipitation with respect to the surface type and retrieval algorithm. The results for the GMI sensor are displayed in the first column, and results for the MHS sensor are displayed in the second column. Error bars mark one standard deviation around the mean.

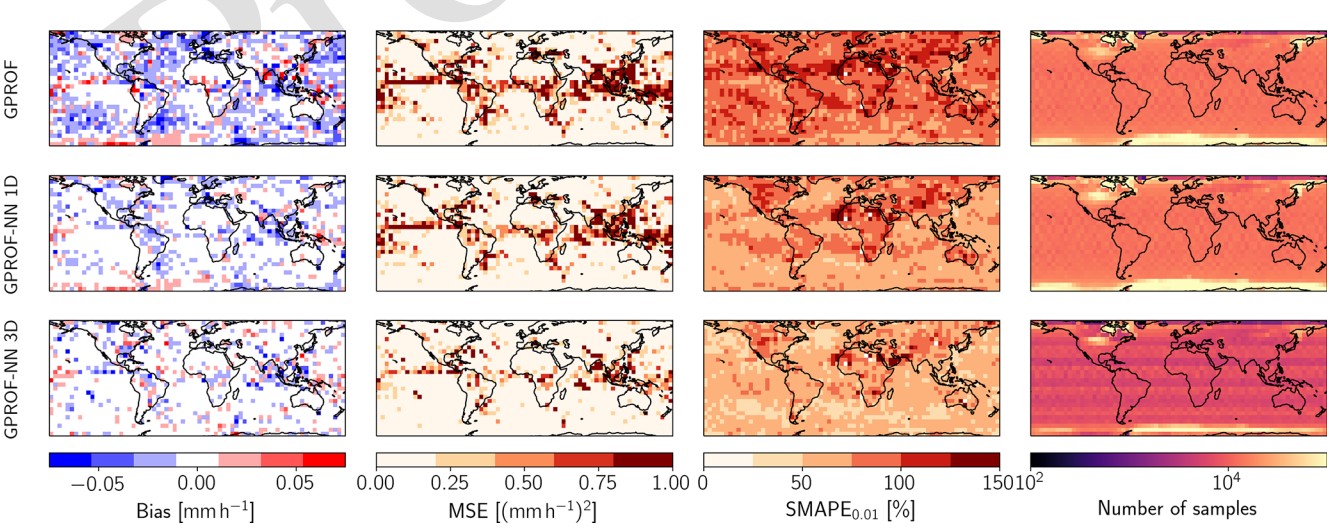

**Figure 8.** Spatial distributions of bias (column 1), MSE (column 2), $SMAPE_{0.01}$ (column 3), and the counts in each $5° \times 5°$ box (column 4) for the GPROF (row 1), GPROF-NN 1D (row 2), GPROF-NN 3D (row 3) algorithms for GMI.

**Table 6.** Calibration of the predicted terciles of the posterior distribution of surface precipitation for the three retrieval algorithms and the GMI and MHS sensors.

| Tercile | Nominal | GMI | | | MHS | | |
|---------|---------|-------|------------|------------|-------|------------|------------|
| | | GPROF | GPROF-NN 1D | GPROF-NN 3D | GPROF | GPROF-NN 1D | GPROF-NN 3D |
| First | 0.333 | 0.461 | 0.351 | 0.349 | 0.274 | 0.34 | 0.326 |
| Second | 0.667 | 0.514 | 0.652 | 0.654 | 0.480 | 0.664 | 0.649 |

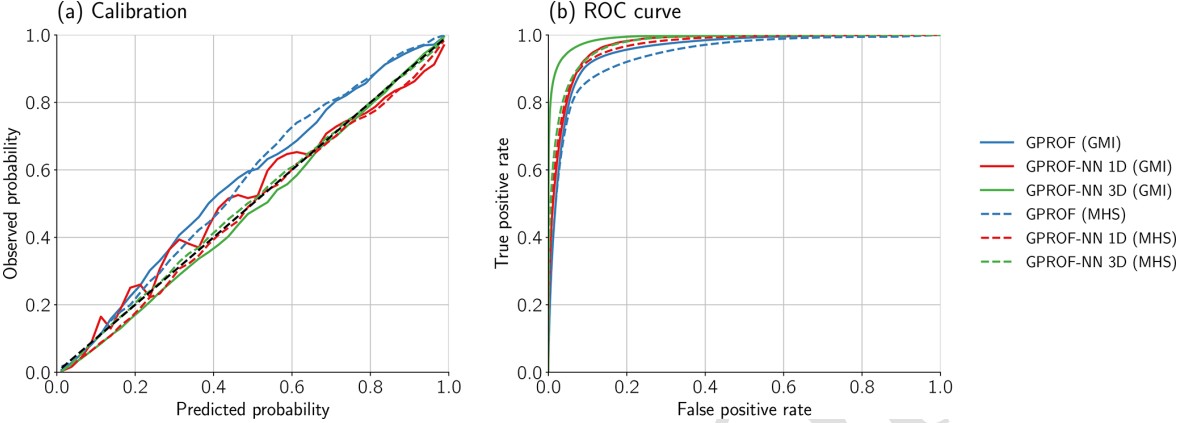

**Figure 9.** Calibration **(a)** and receiver operating characteristic (ROC, **b**) for the predicted probability of precipitation.

### 3.1.3 Effective resolution

Next, we aim to assess the impact of the retrieval method on the effective resolution of the retrieved precipitation fields, which is important for hydrologic applications. For this, we adopt the approach from Guilloteau et al. (2017), who have studied the effective resolution of the previous version of GPROF for the GMI and the TRMM Microwave Imager sensors. A 1D Haar wavelet decomposition in along-track direction over all 128 pixel sequences in the test data is performed to calculate the effective resolution. We do not consider observations for different surface types separately. Following, Guilloteau et al. (2017), we examine energy spectra, correlation coefficients, and Nash–Sutcliffe (NS) efficiency of the coefficients of the wavelet decomposition for the reference and retrieved surface precipitation. The results are displayed in Fig. 10.

An obvious difference to the results from Guilloteau et al. (2017) is that the energy spectrum of the reference precipitation field is not monotonically decreasing. The reason for this is that the reference precipitation field in the retrieval database is smoothed using an averaging filter adapted to the footprint size of the respective sensor. For GMI, the GPROF-NN 3D algorithm has the highest variability in the retrieved precipitation field, followed by the GPROF-NN 1D algorithm and GPROF. However, the variability of all retrievals remains lower than that of the reference field. The correlation of the wavelet coefficients at different scales (Fig. 10b) is highest for the GPROF-NN 3D algorithm, followed by the

GPROF-NN 1D algorithm and GPROF. The same pattern is observed for the NS efficiency. In terms of effective resolution, defined following Guilloteau et al. (2017) as the smallest scale at which the NS efficiency exceeds 0.5, the GPROF-NN 3D algorithm for the GMI sensor achieves a resolution of 13.5 km, which is the distance between consecutive pixels in along-track direction and thus the smallest spatial scale that can be resolved in this analysis. The effective resolution is 14.1 km for the GPROF-NN 1D algorithm and 23.1 km for the GPROF algorithm.

For MHS, the effective resolutions of GPROF and GPROF-NN 1D of 104 and 73 km, respectively, are significantly higher than for GMI. Since the resolution is averaged over the viewing angles of the cross-track scanner and because of its generally lower sensitivity to precipitation, a certain degradation of the resolution was expected. Despite this, the GPROF-NN 3D algorithm achieves a resolution of 22.3 km, which is close to that of GPROF for GMI.

### 3.1.4 Profile retrieval variables

In addition to precipitation fields and path-integrated hydrometeor concentrations, GPROF retrieves concentration profiles of rain, snow, and cloud water. The retrieval database also contains latent heating rates as a target variable, but there is currently no plan to include them in the operational output of GPROF 2021. For this study, latent heating rates were nonetheless included in the output of the GPROF-NN retrievals to investigate the feasibility of the retrieval.

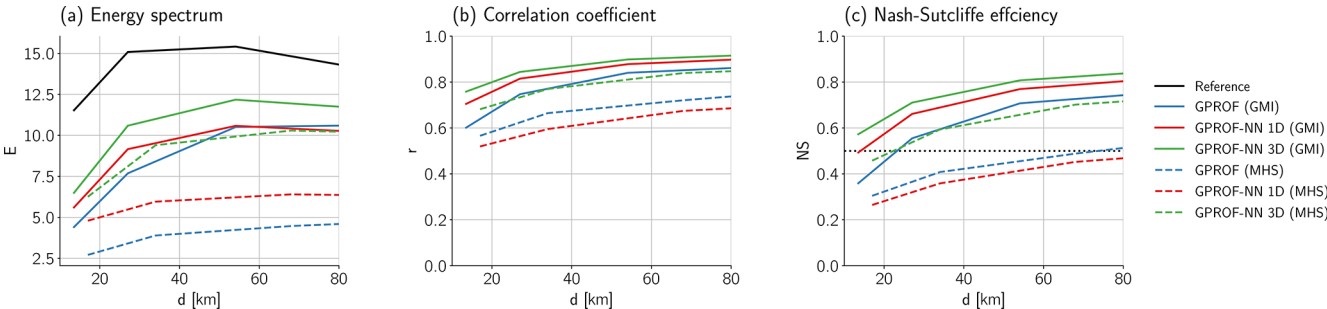

**Figure 10.** Spatial variability of retrieved and reference fields. Panel **(a)** shows the average of the total energy defined as the sum of the squared wavelet coefficients at different length scales for the reference and retrieved surface precipitation fields. Panel **(b)** shows the correlation coefficient between the coefficients of the reference and the retrieved precipitation field. Panel **(c)** shows the corresponding Nash–Sutcliffe efficiency.

The error statistics for the profile retrievals are displayed in Fig. 11. For GMI, the results are qualitatively similar to those observed for the scalar retrieval variables: the GPROF-NN 1D retrieval has slightly lower biases than GPROF, with the GPROF-NN 3D algorithm yielding the lowest biases. Similar patterns are observed for MSE, SMAPE, and correlation throughout most of the atmosphere. For rain and snow water content, the SMAPE of the GPROF-NN 3D retrieval increases and even exceeds that of GPROF-NN 1D at the topmost levels where the hydrometeors are present. This is presumably due to the scarcity of profiles with hydrometeors at these altitudes, leading to decreased accuracy for the more complex neural network model employed by the GPROF-NN 3D algorithm.

For MHS, the retrievals exhibit slightly lower accuracy, but qualitatively the results are very similar to those from GMI. One exception are the biases for cloud water content, which are slightly larger than those of GPROF. It is not quite clear what causes this, but given that the biases remain comparable to those of GPROF and the results for GMI, we do not consider these deviations critical.

## 3.2 Case study: Hurricane Harvey

All of the results presented above were based on a test dataset with the same statistics as the retrieval database. While for GMI the observations can be expected to be consistent with those in the database, this is not necessarily the case for sensors for which the retrieval database contains mostly simulated observations. While a comprehensive analysis of the retrieval performance on real observations is outside the scope of this study, this section presents retrieval results from two overpasses over Hurricane Harvey to provide an indication as to whether the performance characteristics of the retrieval algorithm can be expected to carry over to real observations.

The first considered overpass is from the GPM Core Observatory over Hurricane Harvey and occurred on 25 August 2017 at 11:50 UTC. Figure 12 shows the retrieved surface precipitation and reference measurements from the CMB

product and MRMS. The retrieved precipitation fields exhibit marked differences in structure: the GPROF retrieval produces large areas of low precipitation covering large parts of the scene that are not present in the CMB or MRMS measurements. This is consistent with the overestimation of light precipitation observed in Fig. 5. These artifacts are reduced in the results of the GPROF-NN 1D algorithm and practically absent in the results of the GPROF-NN 3D retrieval.

A quantitative assessment of the retrieval results is provided in Table 7, which shows bias, MSE, and correlation, as well as the precision and recall of the retrieved precipitation flag. The precision is the fraction of correctly detected raining pixels of all pixels predicted to be raining, and the recall is the fraction of all truly raining pixels that are correctly detected.

All statistics were calculated using the CMB product and the MRMS ground-based measurements as a reference. The reference measurements were averaged to the footprint of the GMI 18.7 GHz channel, taking into account the rotation of the pixels across the swath. Only measurements with a radar quality index of at least 0.8 were used for the comparison against MRMS retrievals.

The accuracy of all retrievals is lower when compared to MRMS than when compared to CMB. This is likely because all GPROF retrievals are designed to reproduce the retrieval database, which is to a large extent derived from the CMB product. The GPROF-NN retrievals yield more accurate results than GPROF across all considered metrics except for the recall, which is lower for GPROF-NN 1D than for GPROF. Interestingly, GPROF-NN 1D achieves lower MAE, MSE, and bias, as well as higher correlation than GPROF-NN 3D in the comparison against MRMS, while the two perform similarly in the comparison against CMB.

Figure 13 presents retrieved surface precipitation from an overpass of the MHS sensor on board the NOAA-18 satellite over the same storm at 13:58 UTC. Because no co-located CMB measurements are available for this overpass, only the MRMS measurements are shown as reference measure-

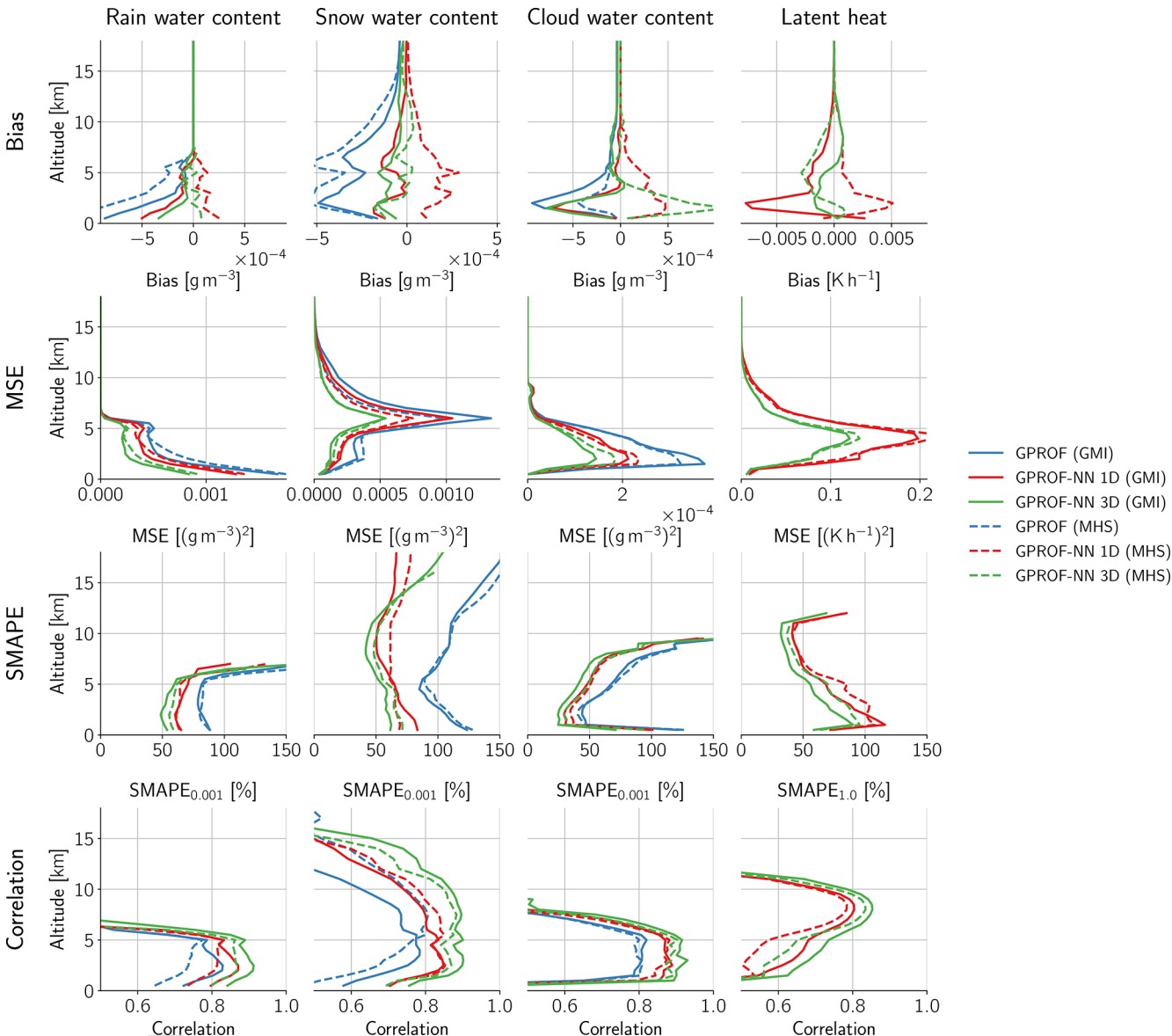

**Figure 11.** Error statistics of the retrieved profile variables. Columns show the errors for the different retrieved variables, whereas rows show altitude averaged bias, RMSE, SMAPE, and correlation, respectively.

**Table 7.** Accuracy metrics for surface precipitation retrieved from GMI PMW observations of Hurricane Harvey for the overpass on 25 August 2017 at 11:50:00 UTC. Each metric is calculated with respect to the surface precipitation from the CMB product as well as the surface precipitation from MRMS as reference.

| Retrieval | Bias [mm h$^{-1}$] | | MSE [(mm h$^{-1}$)$^2$] | | Correlation | | Precision | | Recall | |
| --- | --- | --- | --- | --- | --- | --- | --- | --- | --- | --- |
| | CMB | MRMS | CMB | MRMS | CMB | MRMS | CMB | MRMS | CMB | MRMS |
| GPROF | 0.346 | 0.355 | 2.691 | 8.299 | 0.892 | 0.651 | 0.9 | 0.82 | 0.82 | 0.81 |
| GPROF-NN 1D | 0.245 | 0.145 | 1.944 | 4.927 | 0.914 | 0.701 | 0.95 | 0.9 | 0.90 | 0.75 |
| GPROF-NN 3D | 0.248 | 0.184 | 1.953 | 6.12 | 0.923 | 0.676 | 0.95 | 0.9 | 0.90 | 0.87 |

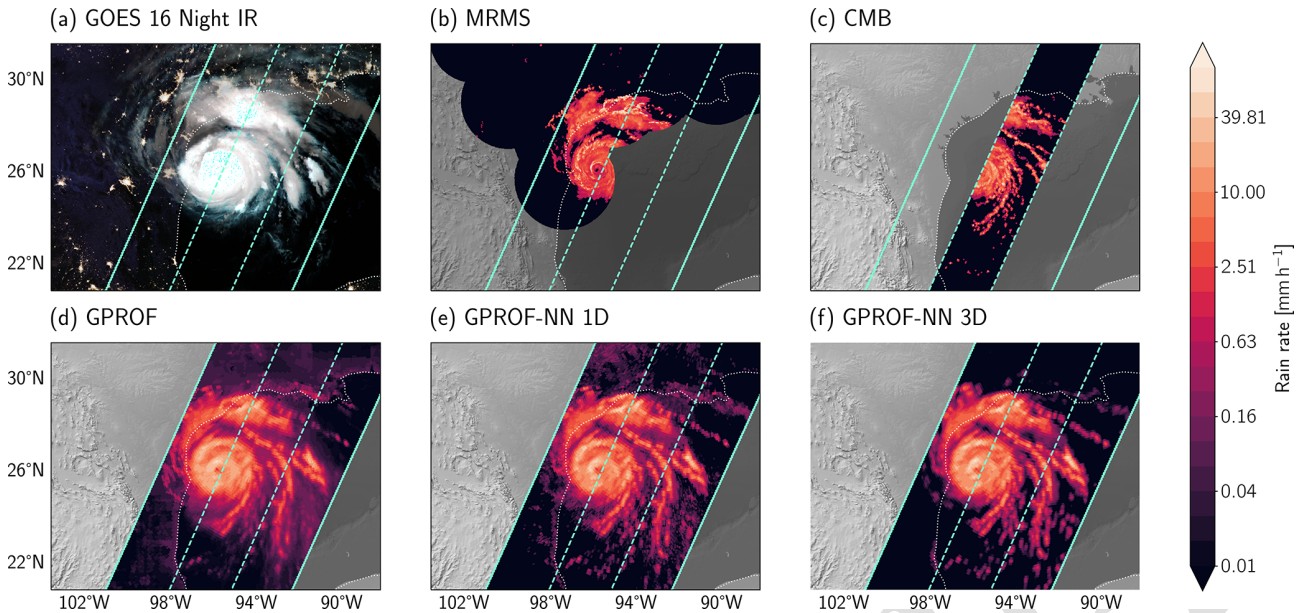

**Figure 12.** Surface precipitation during Hurricane Harvey on 25 August 2017 at 11:50:00 UTC retrieved from GMI. Panel **(a)** shows a GOES 16 Night IR composite (generated using the `night_ir_with_background_hires` composite in satpy, Raspaud et al., 2021), which merges infrared observations from the Advanced Baseline Imager (ABI, Schmit et al., 2005) on GOES 16 and NASA black marble imagery (NASA, 2022). Panel **(b)** shows ground-based precipitation measurements from MRMS for which the radar quality index exceeds 0.8. Panel **(c)** shows retrieved surface precipitation from the CMB product. Panels **(d)**, **(e)**, and **(f)** show the retrieved surface precipitation fields from GPROF, GPROF-NN 1D, and GPROF-NN 3D, respectively.

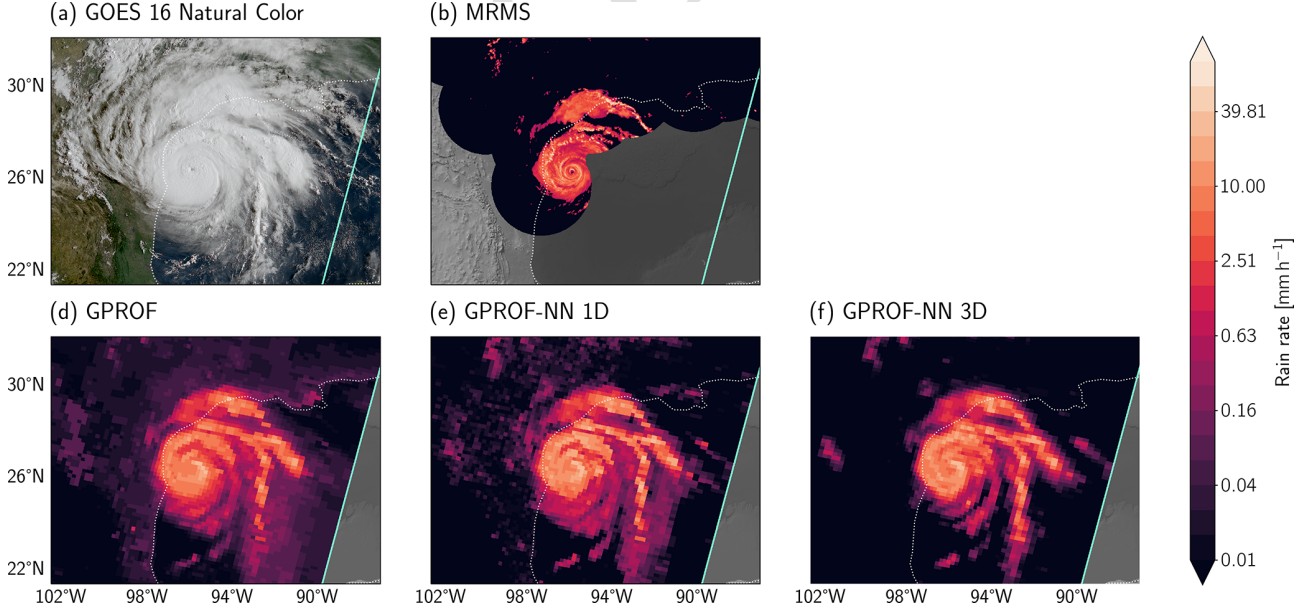

**Figure 13.** Surface precipitation during Hurricane Harvey on 25 August 2017 at 13:58:00 UTC retrieved from MHS on NOAA-18. Panel **(a)** shows a natural color composite from the ABI on GOES 16 (generated using the `natural_color` composite in Satpy; Raspaud et al., 2021). Panel **(b)** shows ground-based precipitation measurements from MRMS for which the radar quality index is at least 0.8. Panel **(c)** shows retrieved surface precipitation from the CMB product. Panels **(d)**, **(e)**, and **(f)** show the retrieved surface precipitation fields from GPROF, GPROF-NN 1D, and GPROF-NN 3D, respectively.

**Table 8.** Accuracy metrics for surface precipitation retrieved from MHS PMW observations of Hurricane Harvey for the overpass on 25 August 2017 at 13:58 UTC. The metrics are calculated against the MRMS surface precipitation estimates. TS1

| Retrieval | Bias [$mm\,h^{-1}$] | MSE [$(mm\,h^{-1})^2$] | Correlation | ☰sion | Recall |
|---|---|---|---|---|---|
| GPROF | 0.11 | 2.602 | 0.749 | 0.88 | 0.12 |
| GPROF-NN 1D | 0.259 | 4.031 | 0.751 | 0.9057 | 0.094 |
| GPROF-NN 3D | 0.152 | 3.168 | 0.759 | 0.948 | 0.052 |

ments. GPROF predicts low precipitation rates across large parts of the scene and even in cloud-free areas. This tendency is reduced in the results of the GPROF-NN 1D retrieval and even more so in the results of GPROF-NN 3D. The GPROF-NN retrievals also generally yield better agreement with MRMS over land. Furthermore, the rain bands of the hurricane are better defined in the results of the GPROF-NN 3D retrievals, which is consistent with the increased effective resolution of the retrievals.

Accuracy metrics for comparing the MHS retrievals with MRMS are shown in Table 8. The MRMS measurements were averaged to the MHS observation footprints, taking into account the changes in footprint size and shape across the swath. For MHS, GPROF has the lowest bias, MAE, and MSE and higher recall than GPROF-NN 1D. These results do not show any clear improvements for the GPROF-NN retrievals. However, the GPROF-NN 3D retrievals improve the retrieval in terms of all metrics compared to GPROF-NN 1D, suggesting that the GPROF-NN 3D can make use of the spatial information in the observations despite being trained on simulated observations.

### 3.3 Processing time

GPROF is used to process PMW observations from a constellation of sensors spanning several decades of observations. Therefore, the processing time must not be excessively high. Although neural networks are generally efficient to evaluate, this often assumes dedicated hardware, which can not be expected to be available at the processing centers yet.

We measure the processing time required for retrieving precipitation from a full orbit of observations using a single CPU core of an Intel Xeon Gold 6234 CPU to assess the computational complexity of the three retrievals. The processing time here includes all steps from reading a GPROF input file to writing the corresponding output file. The input and output files are the same for all three algorithms, excluding differences in the retrieval results.

The results are displayed in Fig. 14. The processing of a single GMI file takes about 4 min for GPROF but only about 2 min for the GPROF-NN retrievals. Because of the lower number of pixels in a single orbit, all retrievals are significantly faster for MHS. However, here the GPROF-NN retrievals are also significantly faster than GPROF. This shows that, even in the absence of dedicated hardware, the GPROF-

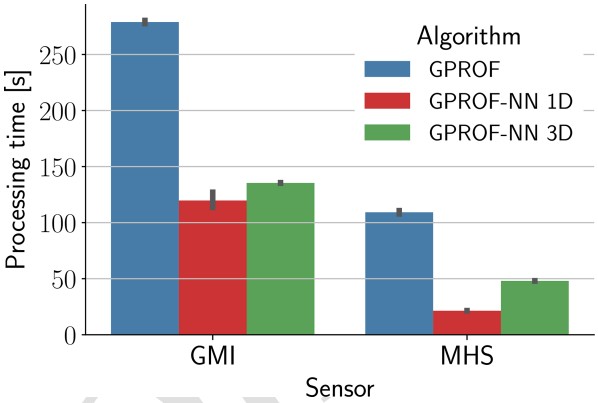

**Figure 14.** Single CPU core processing time for an orbit of observations for the three retrieval algorithms for GMI and MHS. Error bars show the range of 1 standard deviation around the mean for five executions of each retrieval.

NN retrievals process observations faster than the current implementation.

## 4 Discussion

This study presented two novel neural-network-based implementations of the GPROF retrieval algorithm and evaluated their performance for the GMI and MHS sensors against the current implementation.

### 4.1 Retrieval performance

The evaluation of the GPROF-NN 1D algorithm against GPROF showed that retrieval accuracy and effective resolution can be improved by replacing the current retrieval method with a fully connected neural network. Although GPROF and the GPROF-NN 1D algorithm are both based on the same retrieval database and use the same information as retrieval input, the neural network provides more accurate results. A potential explanation for this may lie in the way the two algorithms handle observation uncertainties, which were shown by Elsaesser and Kummerow (2015) to have a significant effect on the retrieval accuracy. For GPROF, these uncertainties must be specified manually. Apart from sensor noise, observations from sensors other than GMI are affected by modeling errors in the simulated observations, which are

difficult to estimate and unlikely to be well described by the assumed Gaussian error model. In addition to this, uncertainties are inflated to account for the sparsity of the retrieval database and the effects of clustering. Since samples with low precipitation rates are generally better represented in the database, this likely makes the uncertainties too large for the retrieval of low precipitation rates, which may explain the inferior performance of GPROF for low precipitation rates observed in Figs. 5 and 6. The neural-network-based algorithms infer observation errors directly from the data and can thus handle arbitrary observation errors if they are accurately represented in the training data.

Another advantage of the neural network retrievals that may explain the improved accuracy is that they scale more easily to large retrieval databases. While the GPROF algorithm requires compressing the retrieval database in a way that causes information loss, the training of the neural networks uses the full database. However, even in the absence of clustering, Pfreundschuh et al. (2018) provided empirical evidence that neural-network-based retrievals are less affected by the curse of dimensionality, which means they yield more accurate results when limited data are available. Although the GPROF database is fairly large, heavy precipitation events are likely still underrepresented, which may be exacerbated by the clustering performed by GPROF. In this context, it may also be worth pointing out that since the database size only influences the training time of the GPROF-NN algorithms, they can potentially be applied with even larger retrieval databases than the one currently used, which may help to further improve the retrieval accuracy in the future.

The second important finding from this study is that by extending the retrieval to incorporate structural information, its accuracy can be further improved by about 20 % in terms of MAE, MSE, and SMAPE and 5 % in terms of correlation compared to the GPROF-NN 1D retrieval at the same time as the effective resolution in the along-track direction is decreased to its lower limit of 13.5 km for GMI and improved by 70 % for MHS. Because precipitation exhibits distinct spatial patterns in satellite observations, many algorithms make use of this information to improve precipitation retrievals (Kummerow and Giglio, 1994a; Sorooshian et al., 2000; Hong et al., 2004; Gopalan et al., 2010). Our results confirm that CNNs learn to leverage this information directly from the satellite imagery and that it can notably improve the retrieval accuracy, which is in agreement with the findings from other precipitation retrievals that employ CNNs (Tang et al., 2018; Sadeghi et al., 2019; Gorooh et al., 2022; Sanò et al., 2018).

As a concluding remark regarding the retrieval performance, it should also be noted that this study focused on the development of a generic retrieval algorithm applicable to all sensors of the GPM constellation within the operational constraints of the current GPROF retrieval. This means that the neural network models used were not optimized exhaustively and that the performance of neural-network-based PMW pre-

cipitation retrievals can likely be improved further by dedicated tuning of the architecture.

## 4.2   Limitations

It is important to consider the limitations of the results presented in this study. We have deliberately limited the evaluation of the retrieval accuracy to test data with the same statistical properties as the retrieval database. This was done to isolate the effect of the retrieval method from potential aliasing effects that would be introduced by the use of external validation data. The presented retrieval accuracy should therefore be interpreted as an upper bound on the accuracy that can be achieved with respect to external validation data. Since the GMI retrieval is trained using real observations, the performance on real observations can be expected to be close to the results presented, which was confirmed by the results from the GMI overpass over Hurricane Harvey (Fig. 12).

For other sensors, however, the observations in the database can only be simulated and may deviate significantly from true observations. As described in Sect. B1.2, the training of the GPROF-NN 3D retrieval requires an additional neural network model to generate simulated observations of sufficiently large extent to train a CNN. The results presented in Sect. 3.1 should therefore be seen as an assessment of the potential benefits of a CNN-based retrieval given a perfect retrieval database rather than the real-world retrieval accuracy.

The quantitative assessment of the accuracy of the MHS retrievals of Hurricane Harvey did not show any clear improvements for the GPROF-NN retrievals compared to GPROF. This can be due to multiple reasons. Firstly, the hurricane constitutes an extreme event and it is likely that the instantaneous MRMS precipitation rates used as reference measurements are themselves affected by considerable uncertainties. Secondly, given that the bulk of the precipitation in the considered scene is intense and over ocean, GPROF can be expected to work quite well. This makes it less likely to find clear improvements in this particular scenario. Finally, the accuracy of the neural-network-based retrievals may be limited by the modeling error of the simulations in the retrieval database. In principle, simulation errors could even cause the GPROF-NN retrievals to be less accurate than GPROF for real observations. Should this really be the case, the demonstrated potential of the GPROF-NN retrievals would imply that the quality of the simulations in the GPROF database limits the accuracy of the GPM PMW precipitation measurements and that future work should focus on improving the simulations.

## 5   Conclusions

The results presented in this study clearly demonstrate the potential of a neural-network-based implementation of GPROF to improve accuracy and effective resolution of

retrievals of precipitation and hydrometeor profiles. Both GPROF-NN retrievals have been designed as a drop-in replacement for GPROF and can be directly used in the operational GPM processing pipeline. The results presented in this study show that, given a perfect retrieval database, considerable improvements in the accuracy of GPROF can be achieved by replacing the current Bayesian scheme with a deep neural network that processes pixels independently. In addition, further improvements of similar magnitude can be achieved with a CNN-based implementation that incorporates structural information into the retrieval.

Although the results presented here cannot fully answer the question to what extent the improvements observed for the GPROF-NN algorithms carry over to operational application of the retrievals, they show the potential of the neural-network-based PMW retrievals. Upgrading GPROF to a neural-network-based retrieval thus has the potential of being a very cost-efficient way to improve global measurements of precipitation with the added advantage of being applicable even to historical observations. Furthermore, the results provide an important reference point, which, together with a future evaluation of the retrievals against independent measurements, is required to inform further development aiming to improve the accuracy of GPM PMW retrievals.

The GPROF implementations presented in this study constitute a first step towards a potential upgrade of GPROF to a neural-network-based implementation. The next step will be to run the GPROF-NN retrievals alongside GPROF 2021 for all sensors of the GPM constellation and to validate the retrieval results against independent validation data. The Python-based software package that implements the retrieval and training framework is made available together with all trained models as free software (Pfreundschuh, 2022).

Although the effective improvements that will be achieved in operational use still remain to be investigated, we take the results presented here as a promising indication of the potential of the GPROF-NN retrievals to improve PMW retrievals from the sensors of the GPM constellation. These algorithms may thus constitute a step towards improving our ability to measure the global hydrological cycle and its changes in a warming climate.

## Appendix A: The GPROF Bayesian retrieval scheme

At the base of the GPROF is a Bayesian retrieval method based on Monte Carlo integration of the profiles in the retrieval database. The database is split into bins using the ancillary data to reduce the number of profiles that must be processed for each pixel and better constrain the retrieval. Moreover, the profiles are clustered to further reduce the number of profiles to process. Fig. A1 illustrates the three components of the GPROF retrieval.

The binning of the profiles in the database is performed with respect to all ancillary variables, that is $T_{2\,m}$, TCWV, surface type, and the airlifting index (Fig. A1a). Each bin covers a range of 1 K in $T_{2\,m}$ and 1 kg m$^{-2}$ in TCWV around the closest corresponding integral value. If a bin contains less than 30 000 profiles, its profiles are combined with those from bins with neighboring $T_{2\,m}$ and TCWV values.

The profiles in each bin are combined into self-similar clusters and only the mean of the observations and retrieval targets and the number of observations is retained (Fig. A1b). The hierarchical clustering merges profiles with similar observations until the number of clusters per bin is less than 800.

The binning and clustering of the database is performed offline, i.e., during the development phase of the retrieval. The following scheme is applied to retrieve precipitation and hydrometeor profiles from the clustered database bins. The first step in the retrieval is the determination of the database bins to be used. The central bin is found from the surface type and airlifting index and the rounded input $T_{2\,m}$ and TCWV values. Profiles from this central bin and the two neighboring $T_{2\,m}$ bins are considered for the retrieval.

Let $(\boldsymbol{y}_1, \boldsymbol{x}_1, n_1), \ldots, (\boldsymbol{y}_N, \boldsymbol{x}_N, n_N)$ denote the profile clusters in the selected database bin, where $\boldsymbol{y}_i$ and $\boldsymbol{x}_i$ are the centroids of the observation and state vector, respectively, and $n_i$ is the corresponding number of profiles in the $i$th cluster. Assuming that the profiles in the database bins are distributed according to the a priori distribution $p(\boldsymbol{x})$, the expected value of $\boldsymbol{x}$ with respect to the corresponding posterior distribution $p(\boldsymbol{x}|\boldsymbol{y})$ can be approximated using

$$
\int_{\boldsymbol{x}} \boldsymbol{x}\, p(\boldsymbol{x}|\boldsymbol{y})\, \mathrm{d}\boldsymbol{x} = \int_{\boldsymbol{x}} \boldsymbol{x}\, \frac{p(\boldsymbol{y}|\boldsymbol{x})\, p(\boldsymbol{x})}{p(\boldsymbol{y})}\, \mathrm{d}\boldsymbol{x}
$$
$$
\approx \frac{\sum_i p(\boldsymbol{y}|\boldsymbol{x}_i)\boldsymbol{x}_i}{\sum_i p(\boldsymbol{y}|\boldsymbol{x}_i)}. \tag{A1}
$$

The conditional probability $p(\boldsymbol{y}|\boldsymbol{x}_i)$ of the input observation $\boldsymbol{y}$ given atmospheric state $\boldsymbol{x}_i$ is taken as the probability of the deviations of $\boldsymbol{y}$ from the observations $\boldsymbol{y}_i$ to be caused by the random error in the observations, which is assumed to be unbiased and Gaussian:

$$
p(\boldsymbol{y}|\boldsymbol{x}_i) = \frac{n_i}{\sqrt{\det(2\pi\mathbf{S})}} \exp\left\{ -\frac{1}{2}(\boldsymbol{y}-\boldsymbol{y}_i)^T \mathbf{S}^{-1} (\boldsymbol{y}-\boldsymbol{y}_i) \right\}, \tag{A2}
$$

where $\mathbf{S}$ is a diagonal covariance matrix. The observation error includes sensor noise and other causes of deviations of real observations from the observations in the database, such as calibration errors or modeling errors. It should be noted here that the assumption of Gaussian errors with a state-independent, diagonal covariance matrix is made for simplicity but is likely insufficient to accurately describe modeling errors that are state dependent and correlated between channels. As illustrated in Fig. A1c, Eq. (A1) corresponds to a resampling of the states in the database with case-specific

**(a) Stratified database**          **(b) Clustering**          **(c) Bayesian retrieval scheme**

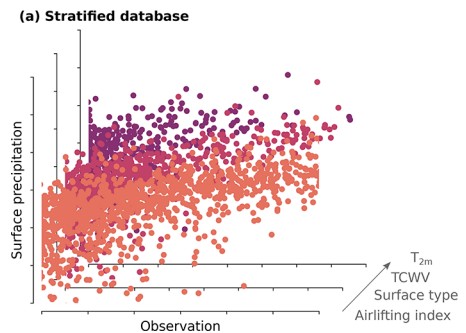 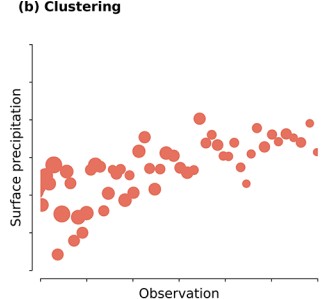 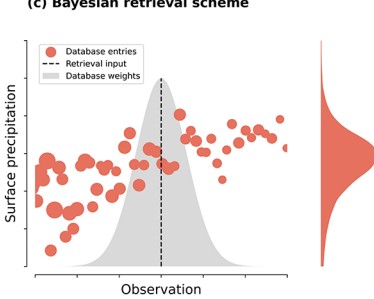

**Figure A1.** Components of the GPROF retrieval algorithm. Panel **(a)** illustrates the binning of the database with respect to the ancillary data, which consists of 2 m temperature ($T_{2\,m}$), total column water vapor (TCWV), surface type, and the airlifting index. Panel **(b)** illustrates the clustering of each database bin into self-similar clusters, with the size of the markers representing the number of profiles in each cluster. Panel **(c)** illustrates the Bayesian scheme that is used to approximate the posterior distribution of the retrieval, which corresponds to the filled curve to the right, by weighting the samples in the database.

weights calculated using Eq. (A2). This approach can be extended to approximate the probability density function of the posterior distribution or derive probabilities of certain characteristics of the a posteriori state, such as the presence of precipitation in a given observation.

## Appendix B: Implementation of the GPROF-NN retrievals

### B1 Training data

#### B1.1 Structure

The training data for the GPROF-NN retrievals are stored in an intermediate format to simplify the loading of the data during the training process. The data are organized into scenes measuring 221 contiguous GMI pixels in both along- and across-track directions. Each scene contains the GMI L1C brightness temperatures and the corresponding values of the retrieval quantities at the center of the GMI swath. For sensors other than GMI, each scene also contains the simulated brightness temperatures of the corresponding sensor.

#### B1.2 Generation

An overview of the data flow for the training data generation during the GPROF-NN retrievals is displayed in Fig. B1. The training data originate from four primary sources: the GPROF simulator files, which contain surface precipitation, hydrometeor profiles, and simulated brightness temperatures for an orbit of the GPM combined product. Surface precipitation over snow surfaces and sea ice are derived from MRMS and ERA5 data, respectively. These data are matched with GMI L1C-R brightness temperatures. The data are split into non-overlapping scenes measuring 221 scans and 221 pixels. For sensors other than GMI, the brightness temperature differences between actual and simulated GMI observations are

**Table B1.** Sizes of neural network models and the training data.

|                | Model parameters (GMI) | Training samples       |
| -------------- | ----------------------: | ---------------------- |
| GPROF-NN 1D    | 5 453 056              | 2 136 604 660 pixels   |
| GPROF-NN 3D    | 23 855 792             | 86 350 scenes          |

included and added to the simulated observations to provide a first-order correction for the modeling error in the observations.

Simulated brightness temperatures are only available where the hydrometeor profiles and surface precipitation are both known, i.e., at the center of the training scenes. Because this is insufficient to train a CNN with 2D convolutions for sensors other than GMI, an intermediate simulator retrieval is trained to retrieve simulated brightness temperatures from GMI observations. This retrieval is applied to the training data to fill in the simulated brightness temperatures across the entire GMI swath. The simulator neural network uses the same architecture as the GPROF-NN 3D retrieval.

### B2 Training

Table B1 lists the number of parameters of the neural networks used in the GPROF-NN retrievals together with the number of samples in the training data. Owing to its more complex network architecture, the neural network employed by GPROF-NN 3D has a larger number of parameters. The training data comprise 86 350 scenes of $221 \times 221$ GMI pixels. From those scenes, only the pixels with known surface precipitation are used for the training of the GPROF-NN 1D retrieval. The total number of pixels meeting this threshold is 2 136 604 660.

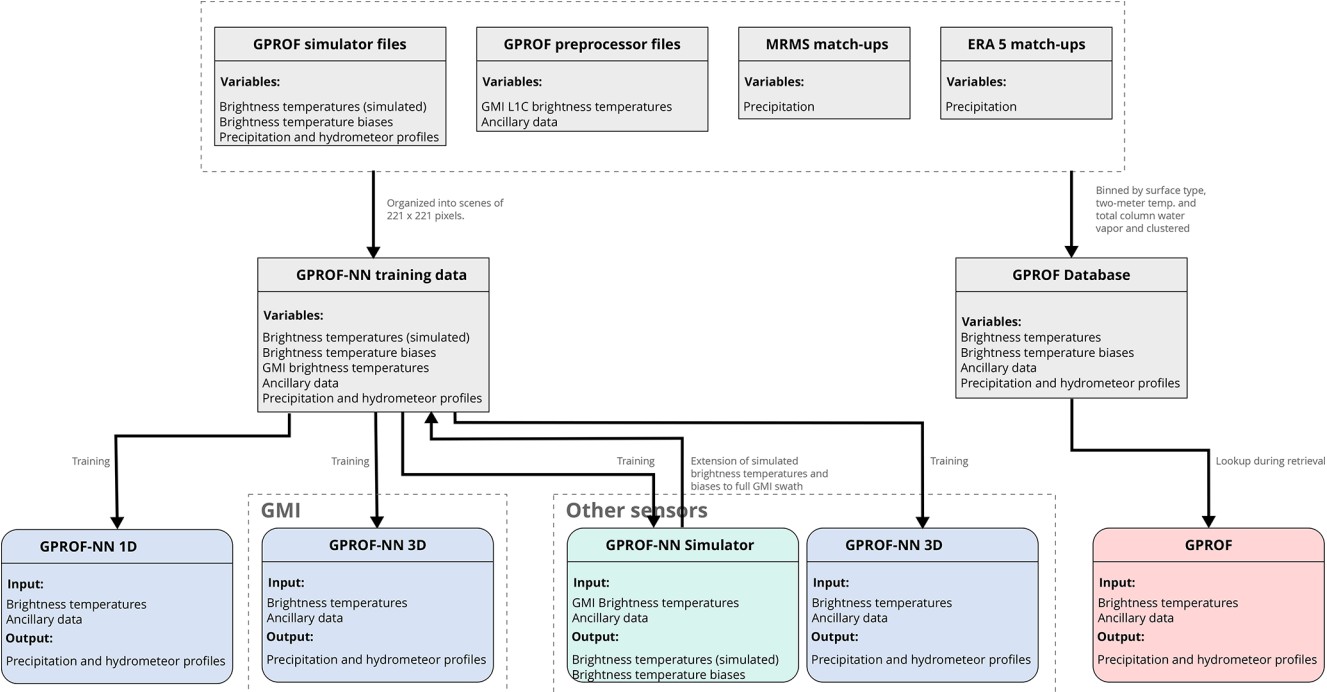

**Figure B1.** Data flow diagram for the generation and organization of the GPROF-NN training data. Grey rectangles represent datasets, and colored rectangles with rounded corners represent algorithms.

## B2.1 GPROF-NN 1D

The GPROF-NN 1D network is trained by simultaneously minimizing the sum of the losses of all retrieval variables. The training is performed over 70 epochs using the Adam optimizer (Kingma and Ba, 2014) with an initial learning rate of $5 \times 10^{-4}$ and a cosine-annealing learning rate schedule (Loshchilov and Hutter, 2016). Warm restarts are performed after 10, 30, and 50 epochs.

The following pre-processing steps are performed when the training data for the GPROF-NN 1D retrieval are loaded:

1. extracting pixels with known surface precipitation,

2. sampling Earth-incidence angle and interpolating inputs and outputs (cross-track scanners),

3. applying the brightness temperature correction (sensors other than GMI),

4. normalizing and encoding the input,

5. replacing zeros,

6. adding thermal noise (sensors other than GMI),

7. shuffling training samples.

The retrieval outputs in the GPROF-NN training data are known only at a limited number of pixels at the center of each scene. Only these pixels are extracted from each scene

in step (1). For cross-track-scanning sensors, the training data contain retrieval input and output for a sequence of discrete Earth-incidence angles. A random Earth-incidence angle is generated for each training sample, and the inputs and outputs are interpolated to that angle (2). If the sensor relies on simulations, the brightness temperatures are corrected using the method described in Sect. 2.3.4. The retrieval inputs are then encoded and normalized (4; see Sect. B2.4). Zero values of non-negative retrieval quantities are replaced by very small random values to avoid degenerate quantiles and problems with the application of the log-linear transformation Eq. (3) (5). Finally, thermal noise according to sensor specification is added to simulated observations for sensors other than GMI (6), and the loaded samples are shuffled (7).

## B2.2 GPROF-NN 3D

The training of the GPROF-NN 3D retrieval is performed over 70 epochs using the Adam optimizer (Kingma and Ba, 2014) with an initial learning rate of $5 \times 10^{-4}$ and a cosine-annealing learning rate schedule (Loshchilov and Hutter, 2016). Warm restarts are performed after 10, 30, and 50 epochs.

The following pre-processing steps are performed when the training data for the GPROF-NN 3D retrieval are loaded:

1. remapping of observations to the viewing geometry of sensor,

2. applying the brightness temperature correction (sensors other than GMI),

3. normalizing and encoding the input,

4. replacing zeros,

5. adding thermal noise and simulator error (sensors other than GMI),

6. shuffling training samples.

Each training scene is randomly remapped from the GMI swath to the viewing geometry of the sensor for which the training is performed (1, see Sect. B2.3). If the sensor relies on simulations, the correction described in Sect. 2.3.4 is applied to the brightness temperatures. The input for each scene is then encoded and normalized (3; see Sect. B2.4). Zero values of non-negative retrieval quantities are replaced by very small random values to avoid degenerate quantiles and avoid problems with the application of the log-linear transformation Eq. (3) (4). Thermal noise and a simulator error are added to the simulated observations for sensors other than GMI (5). The simulator error is modeled to be constant across each scene and determined from the MSE of the simulator network on the training data. Finally, the samples are shuffled (6).

## B2.3 Viewing geometry remapping

Because the largest part of the GPROF retrieval database is derived from collocations of GMI observations with the GPM CMB product, the spatial sampling of most training scenes corresponds to that of the GMI L1C-R product regardless of the sensor for which the database was generated. The viewing geometry of the observations in the database therefore does not match that of the other sensors of the GPM constellation. In addition, the values of the retrieval targets are only known at the center of the GMI swath. The distortions that occur towards the sides of the swath of GMI and the other sensors are therefore not well represented in the training data.

A custom data augmentation scheme is applied to overcome these limitations, which consists of a random remapping of the scenes to the viewing geometry of the target sensor. The remapping is implemented as follows.

1. A random center location $c_{out}$ in the swath of the target sensor is sampled.

2. The approximate positions $\mathbf{p}_{out}$ of $h \times w$ pixels in the swath of the target sensor in a two-dimensional Euclidean coordinate system centered on $c_{out}$ are calculated.

3. A random center location $c_{in}$ in the GMI swath is sampled.

4. The approximate positions $\mathbf{p}_{in}$ of all GMI pixels in the training scene in a two-dimensional Euclidean coordinate system centered on $c_{in}$ are calculated.

5. The retrieval inputs and outputs are interpolated from the positions $\mathbf{p}_{in}$ to the positions $\mathbf{p}_{out}$.

6. For cross-track-scanning sensors, the simulated brightness temperatures and retrieval outputs are interpolated to the Earth-incidence angles corresponding to the positions $\mathbf{p}_{out}$ in the output window.

The height $h$ and $w$ of the output window for GMI is 128 in the along-track direction and 96 in the across-track direction. Since many sensors have considerably wider swaths than GMI, the size of the output window is adapted to avoid cases where too many pixels lie outside the GMI swath. The width of the output window in across-track direction for MHS was set to 32 pixels.

## B2.4 Input normalization and encoding

The brightness temperatures and scalar ancillary data that constitute the input to the retrieval are normalized using minimum–maximum normalization. For each scalar input $x$, the minimum $x_{min}$ and maximum $x_{max}$ values in the training data are calculated. The values are then normalized to the range $[-1, 1]$ using

$$x_{\text{normalized}} = \frac{x - x_{min}}{x_{max} - x_{min}}. \tag{B1}$$

Missing values in the input are set to the value $-1.5$. Categorical ancillary data, i.e., the surface type and air-lifting index, are encoded using one-hot encoding.

## B3 Retrieval processing

The data flow for the application of the GPROF and GPROF-NN retrievals is displayed in Fig. B2. The first step, which is common for all three retrievals, is the augmentation of the GPM L1C data with ancillary data. This process is performed by the GPROF preprocessor application. A detailed description of the ancillary data and its derivation can be found in the GPROF ATBD (Passive Microwave Algorithm Team Facility, 2022). The GPROF preprocessor produces a binary file containing the observations and ancillary data. This file serves as input for both GPROF and the GPROF-NN retrievals.

## B3.1 GPROF-NN 1D

The processing of input observations for the GPROF-NN 1D retrieval involves the following steps:

1. flattening of retrieval inputs,

2. normalizing and encoding the input,

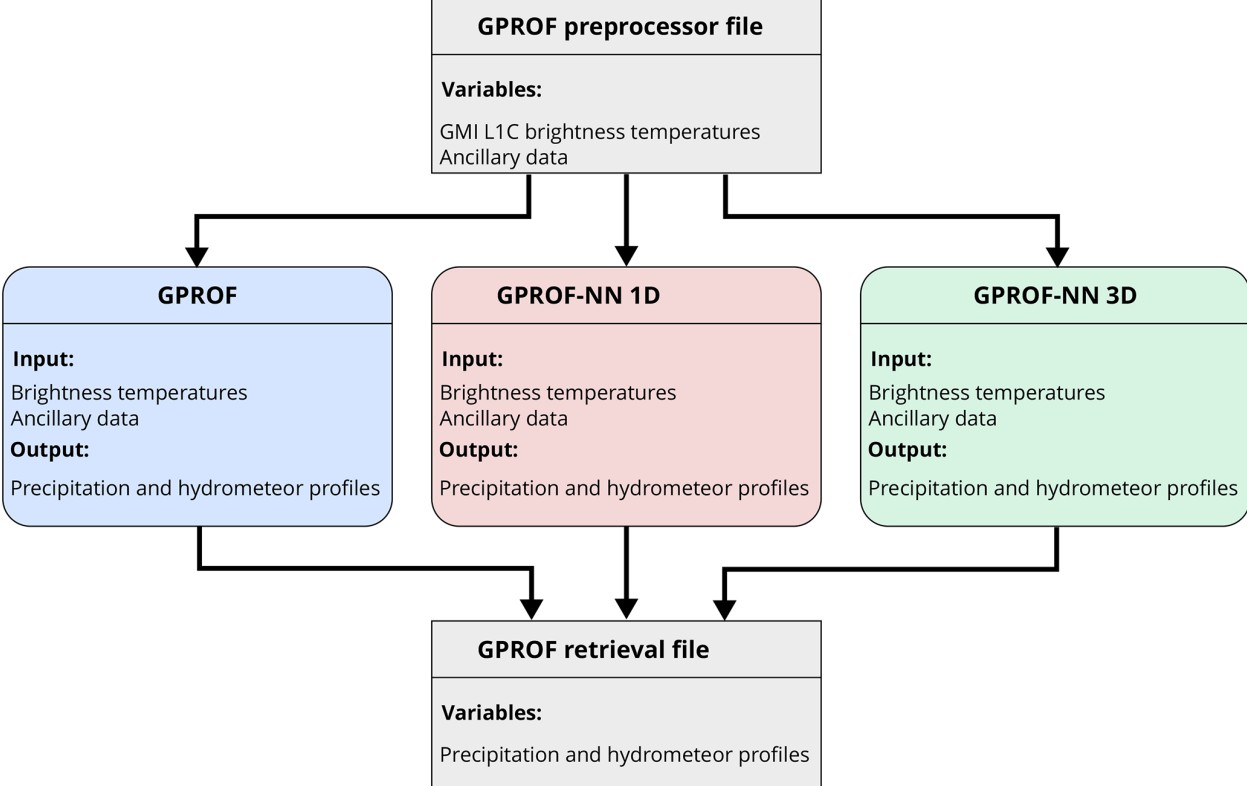

**Figure B2.** Data flow diagram for the application of the GPROF and GPROF-NN retrievals. The input for all retrievals is a GPROF preprocessor file, which is a binary file that contains the brightness temperatures and corresponding ancillary data. From this input, all retrievals produce the retrieval results, which are stored in a common binary format before being converted to HDF5 files.

3. evaluating the network batch-wise and calculating the posterior statistics,

4. reassembling the results into a swath structure,

5. writing the GPROF binary output file.

The observations and corresponding ancillary data are flattened into a list of inputs (1). All inputs are normalized and the categorical input variables are one-hot encoded using the same statistics as during training (2). The GPROF-NN 1D network is then used to calculate the posterior distributions of the retrieval targets from which the relevant posterior statistics are derived (3). Finally, the results for each pixel are reassembled into the original swath structure and written to the GPROF binary output format, which is converted to HDF5 format in a separate step.

### B3.2 GPROF-NN 3D

The processing of input observations for the GPROF-NN 3D retrieval involves the following steps:

1. normalizing and encoding the input,

2. padding the input,

3. evaluating the network and calculating the posterior statistics,

4. removing padding.

The input observations and ancillary data are normalized and encoded using the same statistics as during the training. The input observations are then padded using symmetric padding so that the dimension of the input data are a multiple of 32, which is required to fulfill symmetry requirements of the down- and up-sampling transformations in the neural network. The GPROF-NN 3D network is then evaluated, and the posterior statistics are calculated. Because the GPROF-NN 3D network employs a fully convolutional architecture, the results can be calculated for a full orbit of observations at once. However, since this may require excessive amounts of memory, the processing allows for optional tiling in the along-track direction. After removal of the padding, the retrieval results are written to the same binary format that is used by GPROF-NN 1D and GPROF.

## Appendix C: Error metrics

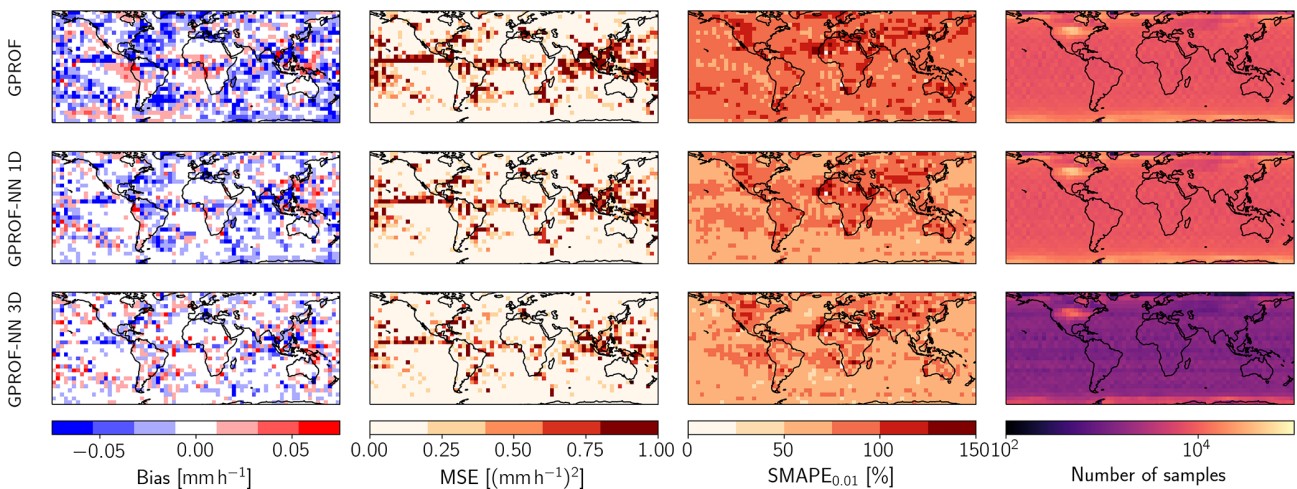

**Figure C1.** The same as Fig. 8 but for MHS.

**Table C1.** The same as Table 4 but for convective precipitation.

| Metric | GPROF | GPROF-NN 1D | GPROF-NN 3D |
|---|---|---|---|
| Bias [$\mathrm{mm\,h^{-1}}$] | $-0.0007 \pm 0.0001$ | $-0.0015 \pm 0.0001$ | $-0.0011 \pm 0.0001$ |
| MAE [$\mathrm{mm\,h^{-1}}$] | $0.0322 \pm 0.0001$ | $0.0239 \pm 0.0001$ | $0.0204 \pm 0.0001$ |
| MSE [$\mathrm{mm\,h^{-1}}$] | $0.1927 \pm 0.0001$ | $0.1298 \pm 0.0001$ | $0.0854 \pm 0.0001$ |
| $\mathrm{MAPE_{0.01}}$ [%] | $118.151 \pm 0.0391$ | $107.1976 \pm 0.0378$ | $92.8343 \pm 0.0542$ |
| Correlation | $0.6380$ | $0.7467$ | $0.8152$ |

**Table C2.** The same as Table 4 but for RWP.

| Metric | GPROF | GPROF-NN 1D | GPROF-NN 3D |
|---|---|---|---|
| Bias [$\mathrm{mm\,h^{-1}}$] | $0.0016 \pm 0.0000$ | $-0.0005 \pm 0.0000$ | $-0.0003 \pm 0.0000$ |
| MAE [$\mathrm{mm\,h^{-1}}$] | $0.0185 \pm 0.0000$ | $0.0127 \pm 0.0000$ | $0.0094 \pm 0.0000$ |
| MSE [$\mathrm{mm\,h^{-1}}$] | $0.0120 \pm 0.0000$ | $0.0086 \pm 0.0000$ | $0.0047 \pm 0.0000$ |
| $\mathrm{MAPE_{0.001}}$ [%] | $84.072 \pm 0.0287$ | $69.6918 \pm 0.0284$ | $61.8979 \pm 0.0315$ |
| Correlation | $0.8308$ | $0.8777$ | $0.9241$ |

**Table C3.** The same as Table 4 but for IWP.

| Metric | GPROF | GPROF-NN 1D | GPROF-NN 3D |
|---|---|---|---|
| Bias [$\mathrm{mm\,h^{-1}}$] | $-0.0022 \pm 0.0000$ | $-0.0006 \pm 0.0000$ | $-0.0002 \pm 0.0000$ |
| MAE [$\mathrm{mm\,h^{-1}}$] | $0.0204 \pm 0.0000$ | $0.0123 \pm 0.0000$ | $0.0085 \pm 0.0000$ |
| MSE [$\mathrm{mm\,h^{-1}}$] | $0.0186 \pm 0.0000$ | $0.0123 \pm 0.0000$ | $0.0053 \pm 0.0000$ |
| $\mathrm{MAPE_{0.001}}$ [%] | $88.26 \pm 0.0312$ | $67.3705 \pm 0.0305$ | $58.5831 \pm 0.0334$ |
| Correlation | $0.7897$ | $0.8637$ | $0.9350$ |

**Table C4.** The same as Table 4 but for CWP.

| Metric | GPROF | GPROF-NN 1D | GPROF-NN 3D |
|---|---|---|---|
| Bias [$\mathrm{mm\,h^{-1}}$] | $-0.0019 \pm 0.0000$ | $-0.0005 \pm 0.0000$ | $-0.0005 \pm 0.0000$ |
| MAE [$\mathrm{mm\,h^{-1}}$] | $0.0268 \pm 0.0000$ | $0.0157 \pm 0.0000$ | $0.0115 \pm 0.0000$ |
| MSE [$\mathrm{mm\,h^{-1}}$] | $0.0027 \pm 0.0000$ | $0.0015 \pm 0.0000$ | $0.0009 \pm 0.0000$ |
| MAPE$_{0.001}$ [%] | $62.2267 \pm 0.0100$ | $36.6584 \pm 0.0078$ | $27.9016 \pm 0.0087$ |
| Correlation | 0.8709 | 0.9265 | 0.9531 |

**Table C5.** The same as Table 5 but for convective precipitation.

| Metric | GPROF | GPROF-NN 1D | GPROF-NN 3D |
|---|---|---|---|
| Bias [$\mathrm{mm\,h^{-1}}$] | $-0.0046 \pm 0.0001$ | $-0.0023 \pm 0.0001$ | $-0.0012 \pm 0.0001$ |
| MAE [$\mathrm{mm\,h^{-1}}$] | $0.0330 \pm 0.0001$ | $0.0281 \pm 0.0001$ | $0.0210 \pm 0.0001$ |
| MSE [$\mathrm{mm\,h^{-1}}$] | $0.1674 \pm 0.0001$ | $0.1337 \pm 0.0001$ | $0.0824 \pm 0.0001$ |
| SMAPE$_{0.01}$ [%] | $108.8755 \pm 0.0480$ | $104.2921 \pm 0.0507$ | $94.0801 \pm 0.1057$ |
| Correlation | 0.5927 | 0.6839 | 0.7336 |

**Table C6.** The same as Table 5 but for RWP.

| Metric | GPROF | GPROF-NN 1D | GPROF-NN 3D |
|---|---|---|---|
| Bias [$\mathrm{mm\,h^{-1}}$] | $-0.0002 \pm 0.0000$ | $-0.0015 \pm 0.0000$ | $-0.0005 \pm 0.0000$ |
| MAE [$\mathrm{mm\,h^{-1}}$] | $0.0210 \pm 0.0000$ | $0.0144 \pm 0.0000$ | $0.0116 \pm 0.0000$ |
| MSE [$\mathrm{mm\,h^{-1}}$] | $0.0143 \pm 0.0000$ | $0.0102 \pm 0.0000$ | $0.0060 \pm 0.0000$ |
| SMAPE$_{0.001}$ [%] | $88.1093 \pm 0.0327$ | $75.4804 \pm 0.0335$ | $72.0101 \pm 0.0703$ |
| Correlation | 0.7591 | 0.8346 | 0.8785 |

**Table C7.** The same as Table 5 but for IWP.

| Metric | GPROF | GPROF-NN 1D | GPROF-NN 3D |
|---|---|---|---|
| Bias [$\mathrm{mm\,h^{-1}}$] | $-0.0035 \pm 0.0000$ | $-0.0009 \pm 0.0000$ | $-0.0008 \pm 0.0000$ |
| MAE [$\mathrm{mm\,h^{-1}}$] | $0.0222 \pm 0.0000$ | $0.0123 \pm 0.0000$ | $0.0100 \pm 0.0000$ |
| MSE [$\mathrm{mm\,h^{-1}}$] | $0.0137 \pm 0.0000$ | $0.0093 \pm 0.0000$ | $0.0060 \pm 0.0000$ |
| SMAPE$_{0.001}$ [%] | $92.0949 \pm 0.0357$ | $74.1056 \pm 0.0362$ | $69.5782 \pm 0.0762$ |
| Correlation | 0.8372 | 0.8878 | 0.9129 |

**Table C8.** The same as Table 5 but for CWP.

| Metric | GPROF | GPROF-NN 1D | GPROF-NN 3D |
|---|---|---|---|
| Bias [$\mathrm{mm\,h^{-1}}$] | $-0.0019 \pm 0.0000$ | $0.0000 \pm 0.0000$ | $-0.0004 \pm 0.0000$ |
| MAE [$\mathrm{mm\,h^{-1}}$] | $0.0268 \pm 0.0000$ | $0.0195 \pm 0.0000$ | $0.0149 \pm 0.0000$ |
| MSE [$\mathrm{mm\,h^{-1}}$] | $0.0027 \pm 0.0000$ | $0.0016 \pm 0.0000$ | $0.0011 \pm 0.0000$ |
| SMAPE$_{0.001}$ [%] | $62.219 \pm 0.0130$ | $47.2591 \pm 0.0114$ | $38.3892 \pm 0.0237$ |
| Correlation | 0.8701 | 0.9194 | 0.9369 |

*Code availability.* The implementation of the GPROF-NN retrievals is published as free software online in Pfreundschuh (2022, https://doi.org/10.5281/zenodo.5819297).

*Data availability.* Because of their size and unlikely usefulness for other researchers, it was deemed impractical to make the GPROF retrieval databases, which were used to generate training and test data, publicly available. However, we are happy to provide access to the data upon request. The training data for the GPROF-NN retrievals, which are derived from these databases, are even larger, which is why we do not store them persistently. However, the required code to generate the training data is publicly available (https://doi.org/10.5281/zenodo.5819297, Pfreundschuh, 2022).

GPM L1C observations and GPM CMB were obtained from https://doi.org/10.5067/GPM/GMI/GPM/1C/07 (Berg, 2022a), https://doi.org/10.5067/GPM/MHS/NOAA18/1C/07 (Berg, 2022b), and https://doi.org/10.5067/GPM/DPRGMI/CMB/2B/07 (Olson, 2022), respectively. MRMS QPE (Smith et al., 2016) data were downloaded from https://mtarchive.geol.iastate.edu/2022/01/01/mrms/ncep/PrecipRate/ (Multi-RADAR Multi-Sensor Archiving, 2022).

*Author contributions.* SP implemented the GPROF-NN algorithms, performed the data analysis, and wrote the manuscript. PJB developed the GPROF 2021 retrieval. CDK supervised the project and provided feedback. PE initiated the project and supervised it. TN implemented the first prototype of the GPROF-NN retrievals.

*Competing interests.* The contact author has declared that none of the authors has any competing interests.

*Acknowledgements.* The computations for this study were performed using several freely available programming languages and software packages, most prominently the Python language (The Python Language Foundation, 2018), the IPython computing environment (Perez and Granger, 2007), the numpy package for numerical computing (van der Walt et al., 2011), xarray (Hoyer and Hamman, 2017) and satpy (Raspaud et al., 2021) for the processing of satellite data, PyTorch (Paszke et al., 2019) for implementing the machine learning models, and Matplotlib (Hunter, 2007) and cartopy (Met Office, 2010–2015) for generating figures.

*Financial support.* This research has been supported by the Swedish National Space Agency (grant no. 154/19), the National Aeronautics and Space Administration (grant no. 80NSSC19K0680), and the Royal Swedish Academy of Engineering Sciences (Hans Werthén Fund Scholarship).

*Review statement.* This paper was edited by Marloes Penning de Vries and reviewed by two anonymous referees.

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

TS2    Please note that the link in the data section and the correspoding reference list entry need to be identical.