# Peer review of "GPROF-NN: A neural network based implementation of the Goddard Profiling Algorithm"

_Atmospheric Measurement Techniques, 2022_

## Referee Comment (RC1)

**GPROF-NN: A neural network based implementation of the Goddard Profiling Algorithm**

The manuscript presents an investigation on using Neural Networks to develop a passive microwave precipitation retrieval model. The authors evaluate their proposed global-scale models using GMI and MHS brightness temperatures compared to the widely used Bayesian-based Goddard Profiling algorithm. In this study, one year of data is utilized to train and validate the NNs models and GPM combined product and MRMS datasets are considered as the reference/ground truth.

The topic is certainly attractive. Satellite precipitation estimation using NNs has been a popular line of research and some papers have recently appeared in the literature. The manuscript is relevant to AMT and follows on to previous NN-related articles by authors including this journal (e.g., Simon Pfreundschus et al. 2018). The manuscript is well written and well organized, the NN methodology sounds, and I do not see any issue with their discussions and general analytical approach. However, the introduction lacks an adequate review of the literature of recent studies, and also data preparation for inputs into the CNN model is poorly explained.

There are some points that should be brought to the attention of the authors that should be easy to address.

Many of your readers may not be familiar with NNs terminology. Would you please highlight the advantages of the NNs, e.g., related CNN and QRNN methods, that today are popular in the satellite precipitation community compared to other ML techniques.
As mentioned before, the study lacks an adequate review of the recent literature about using NNs for satellite precipitation estimation. I suggest some relevant papers (but are not limited to) that are worth reviewing. Please briefly explain already published works in the literature, their challenges/their methodologies, etc., and mention how your work is different from them. What are the open questions you try to address that the previous studies have not considered?
As an example, in Lines 65-70: I understand that you specifically explore the potentials for NNs algorithm in GPROF, so please acknowledge other studies that have already discussed using spatial features in retrieving precipitation.

- *Li, Z., Wen, Y., Schreier, M., Behrangi, A., Hong, Y. and Lambrigtsen, B., 2021. Advancing satellite precipitation retrievals with data-driven approaches: Is black box model explainable?. Earth and Space Science, 8(2), p.e2020EA001423.*
- *Afzali Gorooh, V., Akbari Asanjan, A., Nguyen, P., Hsu, K. and Sorooshian, S., 2022. Deep neural network high SpatioTEmporal resolution Precipitation estimation (Deep-STEP) using Passive Microwave and Infrared Data. Journal of Hydrometeorology.*
- *Sanò, P., Panegrossi, G., Casella, D., Marra, A.C., D'Adderio, L.P., Rysman, J.F. and Dietrich, S., 2018. The passive microwave neural network precipitation retrieval (PNPR) algorithm for the CONICAL scanning Global Microwave Imager (GMI) radiometer. Remote Sensing, 10(7), p.1122.*
- *Ehsani, M.R., Zarei, A., Gupta, H.V., Barnard, K., Lyons, E. and Behrangi, A., 2022. NowCasting-nets: Representation Learning to Mitigate Latency Gap of Satellite Precipitation Products using Convolutional and Recurrent Neural Networks. IEEE Transactions on Geoscience and Remote Sensing.*
-

Lines 13 and 435: How do you define accuracy? Please elaborate on the reported improvements.
Line 15: how do you see the spatial consistency in precipitation retrievals? Does this sentence refer to visualization of derived precipitation rates over Hurricane Harvey for one or two orbital tracks? Please report some statistics for the general spatial detection skills of your proposed models.

Lines 25, Section 3.3: I think the authors need to be cautious in reporting processing time and computational cost comparisons. It is obvious that pixel-wise predictions are faster compared to convolutional-based systems when models are trained and are ready to use. I mean the comparison between GPROF and NN 1D makes sense but including NN 3D is 'Comparing Apples to Oranges'. Processing time means when we have everything set up and ready, let's say we receive one or more orbital tracks (different channels have different footprints, etc.), how long does it take from getting a set of brightness temperatures (Level 1 product) to get the corresponding precipitation maps.

The data preprocessing steps are not clear in the methodology. I suggest summarizing all the training process and prediction (here means after train and validation stage) steps in a numbered list, especially for the CNN algorithm in the methodology section.

Data and method Section: Please clearly explain how many channels are used as inputs to the NN models? What type of resampling/rescaling/interpolation methods do you use? Different radiometers/imagers/sounders have different bands and resolutions, how do you address this problem?
How do you define 18 surface types? Are they generated by TELSEM classification algorithm? Please provide a clear picture of the source of data, pre-processing steps, etc. in this section.
Background material for GPROF Algorithm is well described and cited in previous papers. So please summarize Section 2.2 and please explain more about the innovative parts of your investigation and the proposed models.
Please define all acronyms just the first time you use them. Then use the acronyms in the rest of the manuscript.

Line 200: How many trainable parameters do the NNs algorithms have? Is one year of information enough for training and validating the models?
Line 220: Please add some information about the training stage of models. For example, what are the size of 3D inputs to the CNN model in the training stage? how do you pre-process data to come up with input training samples?

Figure 5. Define regions A and B in the figure.
Please explain the augmentation process in the training stage of CNN model development.

Results Section: from my understanding, the authors only used one year of data for training and validation of NN models. Also, the first three days of 12 months (36 days) are used for test purposes. It is not clear, how many samples (bins) are exactly utilized for testing and reporting the scatter plots and biases? What is the threshold for rain/no-rain samples?

Figures 6 and 7. I do not see a good reason for the color used in these figures, and I find it confusing, commonly blue-red colors would reveal more features. Please find similar figures as the example in Utsumi et al 2020 paper. Also, please report some common statistical indices (related to scatterplots) or detection skill metrics to reveal the discrepancies/improvements. It is better to judge the performance based on statistical indices along with visual assessments.

- *Utsumi, N., Turk, F.J., Haddad, Z.S., Kirstetter, P.-E., Kim, H., 2020. Evaluation of Precipitation Vertical Profiles Estimated by GPM-Era Satellite-Based Passive Microwave Retrievals. Journal of Hydrometeorology 22, 95–112.* https://doi.org/10.1175/JHM-D-20-0160.1

Figure 7: What are the vertical white lines in the last panel of the figure? (Lowest right scatter plot)?

Figure 8 and the associated discussions in this section: The authors mentioned that they have used 18 surfaces classed. Did they regroup the precipitation over different surface types in order to report the statistics? Or here they just report 4 types out of 18? How do the proposed models perform on arid land surface types?

Line 303: Again, how many samples are used to calculate Bias, MSE, etc in each pixels/5-degree box?

Section 3.2: This section presents a visualization of precipitation rates over one or two orbital tracks during Hurricane Harvey. Would you please report some basic statistical indices such as the probability of detection, missed ratio, etc.

Figures 13 and 14. Please show CMB and MRMS products in both figures. Please use the commonly used blue red colorbar and colormap for presenting precipitation rates. Revise the figure in a way that the rain rates less than 1 mm/h are not eliminated. I see that the figures are patchy, and the spatial patterns of precipitation rates are not obvious.
Please remove the colorful background from figure 14. and again, it is miss leading when the precipitation rates less than 1 mm/h in panels c, d, g, h is not shown in the figures.

Section 3.3 as mentioned before, I suggest removing this part or please add more information for different stages of developing NN 1D and NN 3D models, to avoid confusion for the readers. I understand that GPUs, TPUs, etc. can be used to train deep neural networks, and the processing time when everything is ready for the model can be fast for pixel-wise NN 1D. Using NN 3D may be relatively fast in precipitation estimation (prediction phase), but the data preprocessing takes time and is not mentioned here.

Line 413, 461: Please avoid using "simply" replacing or developing. It is not simple!
Line 440: Again, please review the study by Li et al. 2021 and more recent ones that use CNN and PMW data are a part of their input datasets. It is worth mentioning previous works at least in the introduction. Also, it is already established that using neighboring information (spatial features) improves the satellite retrievals both in capturing the amount and the location of events.

- *Li, Z., Wen, Y., Schreier, M., Behrangi, A., Hong, Y. and Lambrigtsen, B., 2021. Advancing satellite precipitation retrievals with data-driven approaches: Is black box model explainable?. Earth and Space Science, 8(2), p.e2020EA001423.*
- *Afzali Gorooh, V., Akbari Asanjan, A., Nguyen, P., Hsu, K. and Sorooshian, S., 2022. Deep neural network high SpatioTEmporal resolution Precipitation estimation (Deep-STEP) using Passive Microwave and Infrared Data. Journal of Hydrometeorology.*
  *and many more,…*

Line 440-445, 452: No evidence has been reported or shown that the model is trained properly. At least please mention the number of samples in the training and testing process, how do the authors

select the hyperparameters? How many parameters do the NN models have compared to GPROF? The Hurricane Harvey event was just a visual representation of retrievals. By adding statistic indices such as pixel- or window-wise correlation, false alarm, missed ratio, etc., the reader can find the improvements and the differences (not only by reporting average bias and visual assessments).

Line 455: Quoting from the manuscript "an additional neural network model was required to transform the data from the retrieval database into a form that is amenable for training a CNN…", I invite the authors to clearly explain the process in the manuscript. It is not clear!

Line 477: I suggest replacing "warming climate" with something like changing climate.

---

## Referee Comment (RC2)

This work (GPROF-NN: A neural network based implementation of the Goddard Profiling Algorithm) presented a retrieval algorithm for two passive microwave sensors (GMI and MHS). The paper is pretty well written and easy to follow. The results are quite impressive. However, I do have two major concerns and some minors comments.

Majors:

1. The validation scheme is not quite convincing. What you did is: using part of the training as the validation dataset (near L255, *first three days of every month from the retrieval database*). This can be a major issue since it is shown that GPROF-NN and GPROF-3D is better than GPGORF-Bayesian. The better performance from GPROF-NN and 3D may result from the over-fitting of the Neural network. I am particularly concerned about the over-fitting issue for surface precipitation from GPROF-NN-3D (Fig. 6, bottom left panel, it seems that the vast majority of the pixels are on 1-by-1 line from 0.1 to 10 mm/hr)

Why not use 1-yr independent data (say, 2020 DPR) to validate your results? Based on Fig. 15, it takes about 120 ~ 250 seconds per orbit to get the results. I highly recommend to redo the validation.

2. The most noticeable improve from NN method is for the very light precipitation (<0.1 mm/hr to 0.01 mm/hr, Fig. 6, 1$^{st}$ column). Then the question is: such light precipitation is really beyond the detection capability of both GMI and MHS. Many previous studies showed that the detection threshold value is around 0.2 mm/hr (e.g., *Munchak, S. Joseph, and Gail Skofronick-Jackson. "Evaluation of precipitation detection over various surfaces from passive microwave imagers and sounders." Atmospheric Research 131 (2013): 81-94*.).

In other words, even if GPROF-NN and GPROF-NN-3D can make this light surface precipitation retrieval better, it is difficult to justify physically you did correctly since these light precipitation are beyond the GMI/MHS detection capability.

Minors:

Line 3: "at such high temporal resolution" to "at three hours temporal resolution", because the temporal resolution from PMWs is rather low (even with the constellation), compared with IR (can be 10 minutes or less).

Line 23: "can be expect" to "can be expected"

Line 33: "3 hours" to "three hours" to be consistent with what you have used in the abstract.

Line 34: "GPM level 3 retrieval products" probably need to change to "GPM level 3 retrieval product". My understanding is that: there is only one Level 3 product (ie.., IMERG). Also, it may be better to briefly introduce IMERG via one sentence since IMERG is more widely used and known. But not so many studies realized that PMWs form the foundation for IMERG.

Line 134: I believe there are two typos in the multiple-variate normal distribution: (1) ni should be 1; and (2) 2pi, should be (2pi)^n (n is the variable number, should be 13 TBs). Please double check.

Line 157: "as well" to "as well as"

Fig. 5. I don't understand what is the color squares. In the caption, it is mentioned "Grey squares mark equilaterals with …", what are the colored squares? I guess grey and color squares are the same??

Fig. 5: Please explain more what are the Fig. 5(c), and why you are doing this way?

Line 250: *To obtain two-dimensional training scenes that are sufficiently wide to train a CNN, we make use of an intermediate CNN based model to 'retrieve' simulated brightness temperatures across the full GMI swath.* Please explain in more details how you did this (i.e., extend from DPR swath to the whole GMI swath).

Both Figure 6 and Figure 7 are over all surface types (i.e., land, ocean, coast, ect)? Please clarify.

Throughout the paper, I did not find which MHS you used (maybe I missed it). Please specify MHS onboard which satellite (there are 5 MHSs, I think).

Line 440: *we are not aware of any other operational PMW algorithms that incorporate structural information using CNNs.* Yes, you are probably correct that nobody is using structural information via CNN. However, structure information has long been used for retrieval from the TRMM era. The land algorithm did by Ferrao group used quite a bit structural information (spatial information) before GPROF transitioned into all Bayesian technique. (see "Estimation of convective/stratiform ratio for TMI pixels" in Gopalan, Kaushik, et al. "Status of the TRMM 2A12 land precipitation algorithm." *Journal of Atmospheric and Oceanic Technology* 27.8 (2010): 1343-1354.) A more recent paper to use the spatial information (Guilloteau, Clément, and Efi Foufoula-Georgiou. "Beyond the pixel: Using patterns and multiscale spatial information to improve the retrieval of precipitation from spaceborne passive microwave imagers." *Journal of atmospheric and oceanic technology* 37.9 (2020): 1571-1591.).

It will be good to briefly discuss how previous studies are using the structural information.

---

## Author Comment (AC1)

**GPROF-NN: A neural network based implementation of the Goddard Profiling Algorithm**

Response to reviewer comments

**1 Comments from reviewer 1**

We want to thank the reviewer for taking the time to read our manuscript and provide valuable feedback.

**1.1 Principal changes**

Many comments from both reviewers were related to the description of the generation of the training data and the implementation of the retrievals. In order to address these comments and avoid excessive growth of the main text of the manuscript, we have added appendices that explain the generation of the training data, the training itself and the application of the retrievals. We have also rewritten much of Sec. 1 and 2 to improve the presentation of both the scope of the manuscript and the design of the GPROF-NN retrievals.

In addition to that, an error in the simulated brightness temperatures for MHS was discovered and corrected. We have updated the results for MHS. While this improved the overall accuracy of the retrieval, it did not affect the study's main findings. Furthermore, we identified and corrected a minor issue in our evaluation of GPROF that caused a slight overestimation of its accuracy. Again, this correction did not affect the conclusions of the paper.

**1.2 Specific comments**

**Reviewer comment 1**

Many of your readers may not be familiar with NNs terminology. Would you please highlight the advantages of the NNs, e.g., related CNN and QRNN methods, that today are popular in the satellite precipitation community compared to other ML techniques.

**Author response:**

We will add two paragraphs to the introduction, which discuss the advantages of neural networks for remote sensing retrievals in general as well as the specific advantages of CNNs and QRNNs.

**Changes in manuscript:**

**Changes starting in line 64:**

 Deep neural networks  (DNNs), which have enabled a number of  significant breakthroughs in different scientific fields (Silver et al., 2016; Jumper et al., 2021), have in recent years been explored for retrieving precipitation from satellite observations. Especially convolutional neural networks (CNNs) are appealing for this application because of their ability to leverage spatial patterns in image data. This property sets them apart from traditional retrieval methods and shallow machine-learning techniques, which are limited in their ability to use this information by computational complexity (Duncan et al., 2019) or the need for feature engineering or manual incorporation of spatial information through techniques such as convective-stratiform discrimination (Gopalan et al., 2010).

**Changes starting in line 97:**

The proposed algorithms are based on quantile regression neural networks (QRNNs, Pfreundschuh et al., 2018), which can be used to predict the posterior distribution of a Bayesian solution of the retrieval, given that the assumed a priori distribution of the Bayesian solution is the same as the distribution of neural network's training data. Because of this, the GPROF-NN  retrievals can produce all of GPROF's retrieval outputs, which include a probability of precipitation and an uncertainty estimate of the predicted precipitation in the form of terciles of the posterior distribution.

**Reviewer comment 2**

As mentioned before, the study lacks an adequate review of the recent literature about using NNs for satellite precipitation estimation. I suggest some relevant papers (but are not limited to) that are worth reviewing. Please briefly explain already published works in the literature, their challenges/their methodologies, etc., and mention how your work is different from them. What are the open questions you try to address that the previous studies have not considered? As an example, in Lines 65-70: I understand that you specifically explore the potentials for NNs algorithm in GPROF, so please acknowledge other studies that have already discussed using spatial features in retrieving precipitation.

- Li, Z., Wen, Y., Schreier, M., Behrangi, A., Hong, Y. and Lambrigtsen, B., 2021. Advancing satellite precipitation retrievals with data-driven approaches: Is black box model explainable?. Earth and Space Science, 8(2), p.e2020EA001423.

- Afzali Gorooh, V., Akbari Asanjan, A., Nguyen, P., Hsu, K. and Sorooshian, S., 2022. Deep neural network high SpatioTEmporal resolution Precipitation estimation (Deep-STEP) using Passive Microwave and Infrared Data. Journal of Hydrometeorology.

- Sanò, P., Panegrossi, G., Casella, D., Marra, A.C., D'Adderio, L.P., Rysman, J.F. and Dietrich, S., 2018. The passive microwave neural network precipitation retrieval (PNPR) algorithm for the CONICAL scanning Global Microwave Imager (GMI) radiometer. Remote Sensing, 10(7), p.1122.

- Ehsani, M.R., Zarei, A., Gupta, H.V., Barnard, K., Lyons, E. and Behrangi, A., 2022. NowCasting-nets: Representation Learning to Mitigate Latency Gap of Satellite Precipitation Products using Convolutional and Recurrent Neural Networks. IEEE Transactions on Geoscience and Remote Sensing.

**Author response:**

We agree with the reviewer that the discussion of previous work on precipitation retrieval with neural networks was insufficient in the first version of the manuscript. We will add a brief summary of previous work to the introduction and discuss the relevant differences to our work.

**Changes in manuscript:**

**Changes starting in line 72:**

 Shallow neural networks have long been used to retrieve precipitation ~~and hydrometeor profiles. Since the retrieval database has grown to a size of several hundred million entries it is perfectly suited for the application of deep neural networks , which scale very well to large amounts of data and are capable of learning complex relationships from them. In addition to this, a neural network based implementation has the advantage of allowing the integration of spatial information into~~ from PMW observations (Staelin and Chen, 2000; Surussavadee and Staelin, 2008) . The Passive microwave Neural network Precipitation Retrieval (PNPR) presented in Sanò et al. (2015); Sanò et al. (2016); Sanò et al. (2018) and the work by Tang et al. (2018) are among the more recent algorithms that use neural networks for retrieving precipitation from PMW observations. They employ relatively shallow neural networks and retrieve precipitation in a pixel-wise manner, thus neglecting spatial structure of the observations. Other recent work demonstrates the ability of CNNs to leverage spatial information in satellite observations. Examples of this are IR-based retrievals by Sadeghi et al. (2019) , PMW-based precipitation detection (Li et al., 2021) and retrievals combining PMW with IR observations (Gorooh et al., 2022) and gauge measurements (Moraux et al., 2019) .

A shortcoming of the aforementioned studies is that none of them addresses the inherent uncertainty of the precipitation retrievals. Retrieving precipitation from PMW observations constitutes an inverse problem, whose ill-posed character leads to significant uncertainties in the retrieval results. Traditionally, these uncertainties are handled using Bayesian statistics. However, because the algorithms mentioned above neglect the probabilistic character of the retrieval,

1.

2.

there is no way to reconcile them with the Bayesian approach.
 Moreover, existing precipitation retrievals that make use of DNNs (Moraux et al., 2019; Sadeghi et al., 2019; Li et al., 2021; Gorooh et al., 2022) are experimental retrievals that are currently not used operationally. The design of an operational retrieval algorithm for the GPM PMW observations needs to address a number of additional requirements, such as the handling of observations from different sensors and the retrieval of multiple output variables. Furthermore, because

GPM is an ongoing mission, continuity of the output variables must be ensured, which further constrains the design of the retrieval algorithm.

**Reviewer comment 3**

Lines 13 and 435: How do you define accuracy? Please elaborate on the reported improvements.

**Author response:**

We will rewrite the sentences to clearly state the observed improvements in the various metrics.

**Changes in manuscript:**

**Changes starting in line 14:**

precipitation retrievals. Despite using the same input information as GPROF, the GPROF-NN  1D  retrieval improves the accuracy of the retrieved surface precipitation  for the GPM Microwave Imager (GMI)  from $0.079$ mmh$^{-1}$ to $0.059$ mmh$^{-1}$ in terms of  mean absolute error (MAE), from 76.1 % to 69.5 % in terms of symmetric mean absolute percentage error  (SMAPE) and from 0.797 to 0.847 in terms of correlation. The improvements for the Microwave Humidity Sounder (MHS) are from $0.085$ mmh$^{-1}$ to $0.061$ mmh$^{-1}$ in terms of MAE, from 81 % to 70.1 % for SMAPE and from 0.724 to 0.804 in terms of correlation. Comparable improvements are  found for the retrieved hydrometeor profiles and their column integrals  as well as  the detection of precipitation. Moreover, the ability of the retrievals to resolve small-scale variability is improved by more than 40 % for GMI and 29 % for MHS. The GPROF-NN 3D ~~retrieval over the performance of the GPROF-NN 1D retrieval, showing the added benefits of incorporating structural information into the retrieval. The effective resolution in along-track direction of the GPROF-NN 3D algorithm is reduced to 13.5 km, which is the upper limit imposed by the along track separation of consecutive scan lines. Comparable improvements are found also when the algorithms are applied to synthetic observations from the cross track scanning Microwave Humidity Sounder (MHS) sensor.~~ retrieval further improves the MAE to $0.043$ mmh$^{-1}$, the SMAPE to 48.67 (mmh$^{-1}$)$^2$ and the correlation to 0.897 for GMI and $0.043$ mmh$^{-1}$, 0.112 (mm h$^{-1}$)$^2$ and 0.83 for MHS.

**Changes starting in line 483:**

The second important finding from this study is that by extending the retrieval to incorporate structural information, its accuracy can be further improved by  about 20 % in terms of MAE, MSE and SMAPE and 5 % in terms of correlation compared to the GPROF-NN 1D retrieval at the same time as the effective resolution in along track direction is decreased to its lower limit of 13.5 km for GMI and improved by 70 % for MHS.

**Reviewer comment 4**

Line 15: how do you see the spatial consistency in precipitation retrievals? Does this sentence refer to visualization of derived precipitation rates over Hurricane Harvey for one or two orbital tracks? Please report some statistics for the general spatial detection skills of your proposed models.

**Author response:**

The sentence that the reviewer is referring to was badly formulated. It was meant to refer to Fig. 9 and A1 from the original manuscript, which show that the improvements are consistent across the globe. However, since we consider this information to be of minor importance we will remove it from the revised version of the manuscript.

**Changes in manuscript**

- The sentence will be removed from the manuscript.

**Reviewer comment 5**

Lines 25, Section 3.3: I think the authors need to be cautious in reporting processing time and computational cost comparisons. It is obvious that pixel-wise predictions are faster compared to convolutional-based systems when models are trained and are ready to use. I mean the comparison between GPROF and NN 1D makes sense but including NN 3D is 'Comparing Apples to Oranges'. Processing time means when we have everything set up and ready, let's say we receive one or more orbital tracks (different channels have different footprints, etc.), how long does it take from getting a set of brightness temperatures (Level 1 product) to get the corresponding precipitation maps.

**Author response:**

The reported processing time measure exactly what the reviewer is requesting, i.e., the time required to process a full orbit of L1C observations augmented with ancillary data. Input and output data are the same for all retrievals and the processing time includes reading and writing of the data. The times are therefore directly comparable.

It seems that this has not been made sufficiently clear in the manuscript. We will rewrite the paragraph to emphasize that the timings are, in fact, comparable.

**Changes in manuscript:**

**Changes starting in line 401:**

 GPROF is used  to process PMW observations from a constellation of sensors spanning several decades of observations. Therefore, the processing time must not be excessively high. Although neural networks are  known to be efficient to evaluate, this often assumes dedicated hardware, which  can not yet be expected to be available at the processing centers.

 the processing time  required for retrieving precipitation from a full orbit of observations using a single CPU core of an Intel Xeon Gold 6234 CPU to assess the computational complexity of the three retrievals. The processing time here includes all steps from reading a GPROF input file to writing the corresponding output file. The input and output files are the same for all three algorithms, excluding, of course, differences in the retrieval results.

The results are displayed in Fig. 8.  The processing of a single GMI file takes about 4 minutes for GPROF but only about 2 minutes for the GPROF-NN  retrievals. Because of the lower number of pixels in a single orbit, all retrievals are significantly faster for MHS. However, also here the GPROF-NN retrievals are significantly faster than GPROF. This shows that, even in the absence of dedicated hardware, the GPROF-NN retrievals process observations faster than the current implementation.

**Reviewer comment 6**

The data preprocessing steps are not clear in the methodology. I suggest summarizing all the training process and prediction (here means after train and validation stage) steps in a numbered list, especially for the CNN algorithm in the methodology section.

**Author response:**

We will add descriptions of the training and evaluation processes of the GPROF-NN retrievals to the manuscript. However, to avoid excessively increasing the length of the manuscript's main text, we will add these sections to the appendix.

**Changes in manuscript:**

- Section B2 will be added to the appendix, which describes the training processes for the GPROF-NN retrievals.

- Section B3 will be added to the appendix, which describes the retrieval processing of the GPROF-NN algorithms.

**Reviewer comment 7**

Data and method Section: Please clearly explain how many channels are used as inputs to the NN models? What type of resampling/rescaling/interpolation methods do you use? Different radiometers/imagers/sounders have different bands and resolutions, how do you address this problem?

**Author response:**

The GPROF-NN retrieval use the same channels as GPROF and don't apply any interpolation apart what is done anyways by the GPROF preprocessing software. We will add a paragraph with this information as well as a table with the channels of the GMI and MHS sensors used in the study to the manuscript.

**Changes in manuscript:**

- Tab. 1.1 will be added to Sec. 2.1 of the manuscript, which shows the channels of the GMI and MHS sensors that are used in this study. In addition to that, the following paragraph will be added to Sec. 2.1 which discusses the handling of different channels in the retrieval database.

  > **Changes starting in line 135:**
  >
  > Since the available channels and the viewing geometries vary between the sensors of the GPM constellation, a separate database is generated for each sensor type. A crucial difference between the retrieval databases for GMI and the other sensors of the GPM constellation is that the database for GMI uses real observations, while the databases for the other sensors are constructed using simulations. The varying resolutions and viewing geometries of different sensors are taken into account by resampling and averaging the simulated observations and retrieval results to the observation footprints of the corresponding sensor. The channels of the GMI and MHS sensors that are used in this study are listed in Tab. 1.1.

- We will rewrite the first paragraph of Sec. 2.3 as follows.

**Changes starting in line 163:**

The principal objective guiding the design of the GPROF-NN algorithms was to develop a  neural-network-based retrieval that operates on the same input data and provides the same output as GPROF so that it can  replace the current implementation in a future update. Although GPROF's retrieval scheme is defined on independent pixels, the algorithm operates on full orbits of observations and corresponding ancillary data. Both GPROF-NN retrievals therefore operate on the same input format as GPROF, which corresponds to each sensor's level 1C observations in their native spatial sampling, which, where required, is remapped to a common grid.

Table 1.1: Channels of the GMI and MHS sensors used for the retrievals in this study.

| Channel | Freq. [GHz] | Pol. |
|---|---|---|
| GMI-1 | 10.6 | V |
| GMI-2 | 10.6 | H |
| GMI-3 | 18.7 | V |
| GMI-4 | 18.7 | H |
| GMI-5 | 23 | V |
| GMI-6 | 37 | V |
| GMI-7 | 37 | H |
| GMI-8 | 89 | V |
| GMI-9 | 89 | H |
| GMI-10 | 166 | V |
| GMI-11 | 166 | H |
| GMI-12 | $183 \pm 3$ | V |
| GMI-13 | $183 \pm 7$ | V |

| Sensor | Freq. [GHz] | Pol. |
|---|---|---|
| MHS-1 | 89 | V |
| MHS-2 | 157 | V |
| MHS-3 | $183 \pm 1$ | H |
| MHS-4 | $183 \pm 3$ | H |
| MHS-5 | 190.31 | V |

**Reviewer comment 8**

How do you define 18 surface types? Are they generated by TELSEM classification algorithm? Please provide a clear picture of the source of data, pre-processing steps, etc. in this section. Background material for GPROF Algorithm is well described and cited in previous papers. So please summarize Section 2.2 and please explain more about the innovative parts of your investigation and the proposed models.

**Author response:**

The surface type information used by GPROF is described in detail in the ATBD. Instead of repeating the content of the ATBS in the manuscript, we will include a reference in the description of the retrieval database.

We will replace Sec. 2.2 with a brief summary and move the detailed description of GPROF to the appendix.

**Changes in manuscript**

- We will add the following paragraph to the end of Sec. 2.1

  > **Changes starting in line 108:**
  >
  > A detailed description of the retrieval data base and the derivation of the data it contains can found in the GPROF ATBD (Passive Microwave Algorithm Team Facility, 2022). The training data for the GPROF-NN retrievals consists of the data from the GPROF retrieval database. To simplify the loading of the data during training it is brought into an intermediate format, which is described in detail in Sec. B1 in the appendix.

- Sec. 2.2 is moved to Sec. A1 in the appendix and replaced with the following summary.

  > **Changes starting in line 110:**
  >
  > The current implementation of GPROF uses a Bayesian scheme to retrieve precipitation and hydrometeor profiles, which works by resampling the profiles in the database based on the similarity of the observations and ancillary data. GPROF uses ancillary data to split the database into separate bins. This reduces the number of profiles for which weights must be computed and helps to constrain the retrieval. Moreover, the profiles in each bin are clustered to limit the number of profiles that need to be processed. A detailed description of the implementation of GPROF is provided in Sec. A in the appendix.

**Reviewer comment 9**

Please define all acronyms just the first time you use them. Then use the acronyms in the rest of the manuscript.

**Author response:**

We will revise the manuscript to make the use of acronyms more consistent.

**Reviewer comment 9**

Line 200: How many trainable parameters do the NNs algorithms have? Is one year of information enough for training and validating the models?

**Author response:**

The GPROF-NN 1D model has about 5 million, and the GPROF-NN 3D model has about 25 million parameters. The training data contains about 2 billion pixels with precipitation information. The fact that the trained models generalize well to unseen test data suggests that it is possible to train these models sufficiently well with the available data.

The number of pixels used to evaluate the retrievals varies between 50 and 3 million for different sensors and retrieval types. Although the samples are not independent, we expect the number to be large enough to yield reliable statistics.

**Change in manuscript**

- We will add the numbers of parameters and sizes of the training datasets to the appendix.

- We will add the table shown in Tab. 1.2 containing the number of pixels used for the evaluation to the beginning of Sec. 3.1.

- We will add the following paragraph to the beginning of Sec. 3.1:

  > **Changes starting in line 288:**
  >
  > Tab. 1.2 lists the number of pixels with precipitation information used for the testing of the retrievals. Spatially contiguous scenes of the same size as the ones used during training are used for the evaluation of GPROF-NN 3D retrievals. Since these scenes generally may not cover all of the pixels with precipitation information, the test data for the GPROF-NN 3D retrievals contains less pixels that can be used for the evaluation. The lower number of test pixels for MHS is due the coarser resolution of the observations which leads to a smaller number of observations over sea-ice and snow and an additional reduction of the pixels available for evaluation of the GPROF-NN 3D retrieval.

Table 1.2: The number of pixels with precipitation information in the test datasets used to evaluate the retrievals.

| Sensor | GPROF & GPROF-NN 1D | GPROF-NN 3D |
|--------|---------------------|-------------|
| GMI    | 50 435 584          | 14 218 203  |
| MHS    | 24 975 877          | 4 945 165   |

**Reviewer comment 11**

Line 220: Please add some information about the training stage of models. For example, what are the size of 3D inputs to the CNN model in the training stage? how do you pre-process data to come up with input training samples?

**Author response:**

To provide a fuller picture of the training processes for the GPROF-NN retrievals, we will include a dedicated section in the appendix and move all information related to the training there.

**Changes in manuscript:**

- We will add a detailed description of the training processes for the retrievals to Sec. B2 in the appendix, which contains the information requested by the reviewer.

**Reviewer comment 11**

Figure 5. Define regions A and B in the figure. Please explain the augmentation process in the training stage of CNN model development.

**Author response:**

We will add the missing labels for the two regions. A detailed description of the augmentation process will be provided in the appendix.

**Changes in manuscript:**

- We have updated Fig. 5 in the manuscript, which now looks as shown in Fig. 1.1

- We will include a detailed description of the augmentation process in Sec. B2.3 the appendix.

[Figure]

Figure 1.1: The updated Fig. 5, which will be included in the revised manuscript.

**Reviewer comment 12**

Figures 6 and 7. I do not see a good reason for the color used in these figures, and I find it confusing, commonly blue-red colors would reveal more features. Please find similar figures as the example in Utsumi et al 2020 paper. Also, please report some common statistical indices (related to scatterplots) or detection skill metrics to reveal the discrepancies/improvements. It is better to judge the performance based on statistical indices along with visual assessments. 3 - Utsumi, N., Turk, F.J., Haddad, Z.S., Kirstetter, P.-E., Kim, H., 2020. Evaluation of Precipitation Vertical Profiles Estimated by GPM-Era Satellite-Based Passive Microwave Retrievals. Journal of Hydrometeorology 22, 95–112. https://doi.org/10.1175/JHM-D-20-0160.1

**Author response:**

It seems that Utsumi et al. (2021) use a spectral colormap that is similar or identical to the 'jet' colormap in their scatter plots. We would like to point out that the 'jet' colormap visually distorts the displayed data due to its non-linear and non-monotonic lightness profile (Thyng et al., 2016). The colormap that is used in the manuscript is perceptually uniform and ordered (matplotlib, 2022). This should make the color map less confusing and less misleading and is in accordance with general guidelines for data visualization (Borland and Ii, 2007).

Since we are not aware of any objective criteria that would justify the use of the 'jet' color map, we will keep the current color map in the revised version of the manuscript. The manuscript already reports a range of common statistical indices (Bias, MSE, MAE, SMAPE) to assess the accuracy of the retrieval. To provide an additional metric that is related to scatter plots, we will extend this to include the correlation.

We will, however, not include detection metrics here because the detection of precipitation is handled by the probability of precipitation and precipitation flag outputs of GPROF. These are evaluated separately in Sec. 3.1.2 of the manuscript.

**Changes in manuscript:**

- We will add the correlation to tables 2, 3, A1, A2, A3, A4, A6, A7, A8 as well as Fig. 8.

**Reviewer comment 13**

Figure 7: What are the vertical white lines in the last panel of the figure? (Lowest right scatter plot)?

**Author response:**

The white lines in the scatter plot were caused by missing test samples at the corresponding cloud water path values. Because the input data for the GPROF-NN 3D retrieval must be assembled into spatially coherent scenes, the test data can't be fully identical to

that used for GPROF and the GPROF-NN 1D retrievals. This is why the missing values only occurred for the results of the GPROF-NN 3D retrieval.

For the revised manuscript we will ensure that the bin sizes for the scatter plots are chosen in a way to avoid these white lines in the revised version of the manuscript.

**Changes in manuscript:**

- Fig. 7 has been updated and now looks like shown in Fig. 1.2.

[Figure]

Figure 1.2: The updated Fig. 7, which will be included in the revised version of the manuscript.

**Reviewer comment 14**

Figure 8 and the associated discussions in this section: The authors mentioned that they have used 18 surfaces classed. Did they regroup the precipitation over different surface types in order to report the statistics? Or here they just report 4 types out of 18? How do the proposed models perform on arid land surface types?

**Author response:**

We regrouped the land surface classes for Fig. 8 because we considered the plot to be too busy with all 18 classes included. We will add an explanation of the regrouping in the text.

The GPROF surface classes do not have an explicit class for arid surfaces but instead a range of classes of increasingly dense vegetation cover. To accomodate the reviewer's suggestion, we will split the 'vegetation' group into densely and sparsely vegetated surfaces.

**Changes in manuscript:**

- We will add the following sentence to the description of Fig. 8.

**Changes starting in line 323:**

The figure displays bias, MSE, MAE, SMAPE, and correlation for principal surface types. The original GPROF surface types have been grouped into ocean (surface type 1), dense vegetation (surface types 3 - 5), sparse vegetation (6 - 7), snow (surface types 8-11), and coast (surface types 12-15).

- We will update the figure to display the accuracy over densely and sparsely vegetated land. The updated plot is shown in Fig. 1.3.

[Figure]

Figure 1.3: The updated Fig. 8 from the original manuscript.

**Reviewer comment 15**

Line 303: Again, how many samples are used to calculate Bias, MSE, etc in each pixels/5-degree box?

**Author response:**

We will add an additional row of panels to the plot that shows the number of samples in each bin.

**Changes in manuscript:**

- Fig. 9 and Fig. A1 will be updated. The updated figures are shown in Fig. 1.4 and Fig. 1.5, respectively.

[Figure]

Figure 1.4: The updated Fig. 9, which will be included in the revised manuscript.

[Figure]

Figure 1.5: The updated Fig. A1, which will be included in the revised manuscript.

**Reviewer comment 16**

Section 3.2: This section presents a visualization of precipitation rates over one or two orbital tracks during Hurricane Harvey. Would you please report some basic statistical indices such as the probability of detection, missed ratio, etc.

**Author response:**

We will add tables with the requested statistics to the evaluation.

**Changes in manuscript:**

- We will add two tables shown in Tab. 1.3 and Tab. 1.4 which the Bias, MSE, MAE, Correlation, Precision and Recall for the GMI and MHS overpasses, respectively

- The discussion of the results in Sec. 3.2 will be rewritten to include these new results.

  > **Changes starting in line 420:**
  >
  > A quantitative assessment of the retrieval results is provided in Tab. 7, which shows bias, MSE and correlation as well as the precision and recall of the retrieved precipitation flag. The precision is just the fraction of correctly detected raining pixels of all pixels predicted to be raining and the recall is the fraction of all raining that is correctly detected.
  >
  > All statistics were calculate using both the CMB product and the MRMS ground-based measurements as reference. The reference measurements were averaged to the footprint of the GMI 18.7 GHz channel taking into account the rotation of the pixels across the swath. Only measurements with a radar quality index exceeding 0.8 were used for the comparison against MRMS retrievals.
  >
  > The accuracy of all retrievals is lower when compared to MRMS than when compared to CMB. This is likely because all GPROF retrievals are designed to reproduce the retrieval database, which is to large extent derived from the CMB product. The GPROF-NN retrievals yield more accurate results than GPROF across all considered metrics except for the recall, which is lower for GPROF-NN 1D than for GPROF. Interestingly, GPROF-NN 1D achieves lower MAE, MSE and Bias as well as higher correlation in the comparison against MRMS, while the two perform very similar in the comparison against CMB.

  > **Changes starting in line 439:**
  >
  > Accuracy metrics for the comparison of the MHS retrievals with MRMS are shown in Tab. 1.3. The MRMS measurements were averaged to the MHS observation footprints taking into account the changes in footprint size and shape across the swath. For MHS, GPROF has the lowest Bias, MAE and MSE as well and higher recall than GPROF-NN 1D. These results do not show any clear improvements for the GPROF-NN retrievals. However, the GPROF-NN 3D retrievals nonetheless improves the retrieval in terms of all metrics compared to GPROF-NN 1D, suggesting that the GPROF-NN 3D is able to make use of the spatial information in the observations despite being

trained on simulated observations.

Table 1.3: Accuracy metrics for surface precipitation retrieved from GMI PMW observations of hurricane Harvey for the overpass on 2017-08-25 at 11:50:00 UTC. Each metric is calculated with respect to the surface precipitation from the CMB product as well as the surface precipitation from MRMS as reference.

| Retrieval | Bias [mm h$^{-1}$] | | MSE [(mm h$^{-1}$)$^2$] | | Correlation | | Precision | | Recall | |
|---|---|---|---|---|---|---|---|---|---|---|
| | CMB | MRMS | CMB | MRMS | CMB | MRMS | CMB | MRMS | CMB | MRM |
| GPROF | 0.346 | 0.355 | 2.691 | 8.299 | 0.892 | 0.651 | 0.9 | 0.82 | 0.82 | 0. |
| GPROF-NN 1D | 0.245 | 0.145 | 1.944 | 4.927 | 0.914 | 0.701 | 0.95 | 0.9 | 0.90 | 0. |
| GPROF-NN 3D | 0.248 | 0.184 | 1.953 | 6.12 | 0.923 | 0.676 | 0.95 | 0.9 | 0.90 | 0. |

Table 1.4: Accuracy metrics for surface precipitation retrieved from MHS PMW observations of hurricane Harvey for the overpass on 2017-08-25 at 13:58 UTC. The metrics calculated against the MRMS surface precipitation estimates.

| Retrieval | Bias [mm h$^{-1}$] | MSE [(mm h$^{-1}$)$^2$] | Correlation | Precision | Recall |
|---|---|---|---|---|---|
| GPROF | 0.11 | 2.602 | 0.749 | 0.88 | 0.12 |
| GPROF-NN 1D | 0.259 | 4.031 | 0.751 | 0.9057 | 0.094 |
| GPROF-NN 3D | 0.152 | 3.168 | 0.759 | 0.948 | 0.052 |

**Reviewer comment 17**

Figures 13 and 14. Please show CMB and MRMS products in both figures. Please use the commonly used blue red colorbar and colormap for presenting precipitation rates. Revise the figure in a way that the rain rates less than 1 mm/h are not eliminated. I see that the figures are patchy, and the spatial patterns of precipitation rates are not obvious. Please remove the colorful background from figure 14. and again, it is miss leading when the precipitation rates less than 1 mm/h in panels c, d, g, h is not shown in the figures.

**Author response:**

While it is not possible to show the CMB product for the overpass of MHS since the GPM overpass occurred at a different time, we will add the MRMS measurements to the GPM overpass. Moreover, we will revise the plots to show precipitation rates across the full swaths on a logarithmic color scale without omitting the precipitation rates less than 1 mm/h.
We will not change the colormap to jet plots based on the arguments presented in the response to comment 12.

**Change in manuscript:**

We will update Fig. 13 and Fig. 14 according to the reviewers suggestions. The updated figures are shown in Fig. 1.6 and Fig. 1.7, respectively.

[Figure]

Figure 1.6: The updated Fig. 13, which will be included in the revised manuscript.

**Reviewer comment 18**

Section 3.3 as mentioned before, I suggest removing this part or please add more information for different stages of developing NN 1D and NN 3D models, to avoid confusion for the readers. I understand that GPUs, TPUs, etc. can be used to train deep neural networks, and the processing time when everything is ready for the model can be fast for pixel-wise NN 1D. Using NN 3D may be relatively fast in precipitation estimation (prediction phase), but the data preprocessing takes time and is not mentioned here.

**Author response:**

See response to comment 5.

**Reviewer comment 19**

Line 413, 461: Please avoid using "simply" replacing or developing. It is not simple!

**Author response:**

We will rewrite this sentence in the revised version of the manuscript.

[Figure]

Figure 1.7: The updated Fig. 14, which will be included in the revised manuscript.

**Changes in manuscript**

**Changes starting in line 461:**

The evaluation of the GPROF-NN 1D algorithm against GPROF, showed that retrieval accuracy as well as effective resolution can be improved  by replacing the current retrieval method with a fully-connected neural network.

**Changes starting in line 421:**

 Both GPROF-NN retrievals have been designed as a drop-in replacement for GPROF and can be directly used in the operational GPM processing pipeline. The results presented in this study show that, given a perfect retrieval database, considerable improvements in the accuracy of GPROF can be achieved by replacing the current  Bayesian scheme with a deep neural network that processes pixels independently.

**Reviewer comment 20**

Line 440: Again, please review the study by Li et al. 2021 and more recent ones that use CNN and PMW data are a PMW data are a part of their input datasets. It is worth mentioning previous works at least in the introduction. Also, it is already established that using neighboring information (spatial features) improves the satellite retrievals both in capturing the amount and the location of events.

- Li, Z., Wen, Y., Schreier, M., Behrangi, A., Hong, Y. and Lambrigtsen, B., 2021. Advancing satellite precipitation retrievals with data-driven approaches: Is black

box model explainable?. Earth and Space Science, 8(2), p.e2020EA001423.

- Afzali Gorooh, V., Akbari Asanjan, A., Nguyen, P., Hsu, K. and Sorooshian, S., 2022. Deep neural network high SpatioTEmporal resolution Precipitation estimation (Deep-STEP) using Passive Microwave and Infrared Data. Journal of Hydrometeorology.

and many more,...

**Author response:**

It was, of course, not our intent to claim that we were the first to make use of spatial information in a PMW retrieval. We will reformulate this paragraph to avoid this misunderstanding.

**Changes in manuscript:**

> **Changes starting in line 509:**
>
> ~~The use of structural information for precipitation retrievals is common practice in algorithms based on infrared observations (Sorooshian et al., 2000; Hong et al., 2004) and the potential benefits of CNN based retrievals have been shown in Sadeghi et al. (2019). While basic structural information has been used in earlier PMW precipitation retrieval algorithms, as e.g. by Kummerow and Giglio (1994), we are not aware of any other operational PMW algorithms that incorporate structural information using CNNs~~ Because precipitation exhibits distinct spatial patterns in satellite observations, many algorithms make use of this information to improve precipitation retrievals (Kummerow and Giglio, 1994; Sorooshian et al., 2000; Hong et al., 2004). Our results confirm that CNNs learn to leverage this information directly from the satellite imagery and that it can notably improve the retrieval accuracy, which is in agreement with the findings from other precipitation retrievals that employ CNNs (Tang et al., 2018; Sadeghi et al., 2
> .

**1.2.1 Reviewer comment 21**

Line 440-445, 452: No evidence has been reported or shown that the model is trained properly. At least please mention the number of samples in the training and testing process, how do the authors select the hyperparameters? How many parameters do the NN models have compared to GPROF? The Hurricane Harvey event was just a visual representation of retrievals. By adding statistic indices such as pixel- or window-wise correlation, false alarm, missed ratio, etc., the reader can find the improvements and the differences (not only by reporting average bias and visual assessments).

**Author response:**

The results presented in Sec. 3 of the manuscript clearly show that the neural networks achieve higher retrieval accuracy on the unseen test data than GPROF. This would not be possible if the networks weren't trained properly. While the assessment of the retrievals over hurricane Harvey can give some indication over how well the retrievals work, a quantitative assessment against independent measurements has to be interpreted carefully because the reference measurements will themselves be affected by uncertainties. In addition to that, the the MHS retrievals are affected by simulation errors, which may limit the accuracy of the GPROF-NN retrievals.

We will reformulate the discussion of the overpasses to emphasize the above points. As mentioned in the response to comment 17, we will extend the evaluation of the hurricane Harvey overpasses to include the requested metrics. We will also include the sizes of the GPROF-NN neural networks and the training data in the new section in the appendix that describes the training data.

**Changes in manuscript**

- We will add tables with the requested statistics to the assessment of the Hurricane overpasses. See response to reviewer comment 16.

- The discussion of the results from the hurricane overpasses will be extended as follows.

  > **Changes starting in line 457:**
  >
  > The quantitative assessment of the accuracy of the MHS retrievals of hurricane Harvey did not show any clear improvements for the GPROF-NN retrievals compared to GPROF. This can be due to multiple reasons. Firstly, the hurricane constitutes an extreme event and it is likely that the instantaneous MRMS precipitation rates used as reference measurements are themselves affected by considerable uncertainties. Secondly, given that the bulk of the precipitation in the considered scene is intense and over ocean, GPROF can be expected to work well. This makes it less likely to find clear improvements in this particular scenario. Finally, the accuracy of the neural-network based retrievals may be limited by the modeling error of the simulations in the retrieval database. In principle, simulation errors could even cause the GPROF-NN retrievals to be less accurate than GPROF for real observations. Should this really be the case, the demonstrated potential of the GPROF-NN retrievals would imply that the quality of the simulations in the GPROF database limits the accuracy of the GPM PMW precipitation measurements and that future work to should focus on improving the simulations.

- The number of parameters of the models will be included in the newly added Sec. B2 in the appendix that described the training of the neural network models.

**1.2.2 Reviewer comment 22**

Line 455: Quoting from the manuscript "an additional neural network model was required to transform the data from the retrieval database into a form that is amenable for training a CNN...", I invite the authors to clearly explain the process in the manuscript. It is not clear!

**Author response:**

We will add an extended description of the generation of the training data in the revised version of the manuscript. It will describe the intermediate retrieval that is used to generate the training data for sensors other than GMI.

**Changes in manuscript**

- Section B1 will be added to the manuscript, which describes the generation of the training data for all sensors including the retrieval used to extend the simulated brightness temperatures.

**1.2.3 Reviewer comment 23**

Line 477: I suggest replacing "warming climate" with something like changing climate.

**Author response:**

While we acknowledge the reviewer's suggestion, we are not aware of any objective arguments for such a change and will therefore not implement it in the revised version of the manuscript.

**Bibliography**

Borland, D. and Ii, R. M. T.: Rainbow color map (still) considered harmful, IEEE computer graphics and applications, 27, 14–17, 2007.

Duncan, D. I., Eriksson, P., and Pfreundschuh, S.: An experimental 2D-Var retrieval using AMSR2, Atmospheric Measurement Techniques, 12, 6341–6359, https://doi.org/10.5194/amt-12-6341-2019, 2019.

Gopalan, K., Wang, N.-Y., Ferraro, R., and Liu, C.: Status of the TRMM 2A12 Land Precipitation Algorithm, Journal of Atmospheric and Oceanic Technology, 27, 1343 – 1354, https://doi.org/10.1175/2010JTECHA1454.1, 2010.

Gorooh, V. A., Asanjan, A. A., Nguyen, P., Hsu, K., and Sorooshian, S.: Deep Neural Network High Spatiotemporal Resolution Precipitation Estimation (Deep-STEP) Using Passive Microwave and Infrared Data, Journal of Hydrometeorology, 23, 597 – 617, https://doi.org/10.1175/JHM-D-21-0194.1, 2022.

Hong, Y., Hsu, K. L., Sorooshian, S., and Gao, X. G.: Precipitation estimation from remotely sensed imagery using an artificial neural network cloud classification system, J. Appl. Meteor., 43, 1834–1852, 2004.

Jumper, J., Evans, R., Pritzel, A., Green, T., Figurnov, M., Ronneberger, O., Tunyasuvunakool, K., Bates, R., Žídek, A., Potapenko, A., et al.: Highly accurate protein structure prediction with AlphaFold, Nature, 596, 583–589, 2021.

Kummerow, C. and Giglio, L.: A Passive Microwave Technique for Estimating Rainfall and Vertical Structure Information from Space. Part I: Algorithm Description, Journal of Applied Meteorology and Climatology, 33, 3 – 18, https://doi.org/10.1175/1520-0450(1994)033<0003:APMTFE>2.0.CO;2, 1994.

Li, Z., Wen, Y., Schreier, M., Behrangi, A., Hong, Y., and Lambrigtsen, B.: Advancing Satellite Precipitation Retrievals With Data Driven Approaches: Is Black Box Model Explainable?, Earth and Space Science, 8, e2020EA001 423, https://doi.org/https://doi.org/10.1029/2020EA001423, e2020EA001423 2020EA001423, 2021.

matplotlib: Choosing Colormaps in Matplotlib, `https://matplotlib.org/stable/tutorials/colors/colormaps.htm`, 2022.

Moraux, A., Dewitte, S., Cornelis, B., and Munteanu, A.: Deep Learning for Precipitation Estimation from Satellite and Rain Gauges Measurements, Remote Sensing, 11, https://doi.org/10.3390/rs11212463, 2019.

Passive Microwave Algorithm Team Facility: GLOBAL PRECIPITATION MEASURE-MENT (GPM) MISSION, `https://gpm.nasa.gov/sites/default/files/2022-06/ATBD_GPM_V7_GPROF.pdf`, 2022.

Pfreundschuh, S., Eriksson, P., Duncan, D., Rydberg, B., Håkansson, N., and Thoss, A.: A neural network approach to estimating a posteriori distributions of Bayesian retrieval problems, Atmos. Meas. Tech., 11, 4627–4643, https://doi.org/10.5194/amt-11-4627-2018, 2018.

Sadeghi, M., Asanjan, A. A., Faridzad, M., Nguyen, P., Hsu, K., Sorooshian, S., and Braithwaite, D.: PERSIANN-CNN: Precipitation estimation from remotely sensed information using artificial neural networks–convolutional neural networks, Journal of Hydrometeorology, 20, 2273–2289, 2019.

Sanò, P., Panegrossi, G., Casella, D., Di Paola, F., Milani, L., Mugnai, A., Petracca, M., and Dietrich, S.: The Passive microwave Neural network Precipitation Retrieval (PNPR) algorithm for AMSU/MHS observations: description and application to European case studies, Atmospheric Measurement Techniques, 8, 837–857, 2015.

Sanò, P., Panegrossi, G., Casella, D., Marra, A. C., Di Paola, F., and Dietrich, S.: The new Passive microwave Neural network Precipitation Retrieval (PNPR) algorithm for the cross-track scanning ATMS radiometer: description and verification study over Europe and Africa
using GPM and TRMM spaceborne radars, Atmospheric Measurement Techniques, 9, 5441–5460, https://doi.org/10.5194/amt-9-5441-2016, 2016.

Sanò, P., Panegrossi, G., Casella, D., Marra, A. C., D'Adderio, L. P., Rysman, J. F., and Dietrich, S.: The passive microwave neural network precipitation retrieval (PNPR) algorithm for the CONICAL scanning Global Microwave Imager (GMI) radiometer, Remote Sensing, 10, 1122, 2018.

Silver, D., Huang, A., Maddison, C. J., Guez, A., Sifre, L., Van Den Driessche, G., Schrittwieser, J., Antonoglou, I., Panneershelvam, V., Lanctot, M., et al.: Mastering the game of Go with deep neural networks and tree search, nature, 529, 484–489, 2016.

Sorooshian, S., Hsu, K. L., Gao, X., Gupta, H. V., Imam, B., and Braithwaite, D.: Evaluation of PERSIANN system satellite based estimates of tropical rainfall, Bull. Amer. Meteor. Soc., 81, 2035–2046, 2000.

Staelin, D. and Chen, F.: Precipitation observations near 54 and 183 GHz using the NOAA-15 satellite, IEEE Transactions on Geoscience and Remote Sensing, 38, 2322–2332, https://doi.org/10.1109/36.868889, 2000.

Surussavadee, C. and Staelin, D. H.: Global Millimeter-Wave Precipitation Retrievals Trained With a Cloud-Resolving Numerical Weather Prediction Model, Part I: Retrieval Design, IEEE Transactions on Geoscience and Remote Sensing, 46, 99–108, https://doi.org/10.1109/TGRS.2007.908302, 2008.

Tang, G., Long, D., Behrangi, A., Wang, C., and Hong, Y.: Exploring Deep Neural Networks to Retrieve Rain and Snow in High Latitudes Using Multisensor and Reanalysis Data, Water Resources Research, 54, 8253–8278, https://doi.org/https://doi.org/10.1029/2018WR023830, 2018.

Thyng, K. M., Greene, C. A., Hetland, R. D., Zimmerle, H. M., and DiMarco, S. F.: True Colors of Oceanography: Guidelines for Effective and Accurate Colormap Selection, Oceanography, 29, 9–13, URL `http://www.jstor.org/stable/24862699`, 2016.

Utsumi, N., Turk, F. J., Haddad, Z. S., Kirstetter, P.-E., and Kim, H.: Evaluation of Precipitation Vertical Profiles Estimated by GPM-Era Satellite-Based Passive Microwave Retrievals, Journal of Hydrometeorology, 22, 95 – 112, https://doi.org/10.1175/JHM-D-20-0160.1, 2021.

---

## Author Response (AR1)

**GPROF-NN: A neural network based implementation of the Goddard Profiling Algorithm**

**Response to reviewer comments**

This document contains the responses to the comments of each reviewer followed by the marked-up differences of the manuscript and the revised version. For each comment the author's response and, if applicable, the corresponding changes in the manuscript are listed. Line numbers of changes are given with respect to the revised manuscript.

The two most prominent changes in the revised manuscript are:

- Addition of a figure showing the temporal delay between the active and passive observations, which we suppose explains the larger residuals for the C159 and C161 flight. Extension of the discussion of the residuals for those flights to mention the temporal offsets between the active and passive observations.

- Addition of a figure showing scatter plots of IWP and residuals of the $243 \pm 2.5$ GHz and $325 \pm 9.5$ GHz channels for flight B984, which hint at a weak relationship particle shape and residuals for this flight.

We want to thank both reviewers for taking the time to read our manuscript and provide valuable feedback. The feedback was very helpful in guiding our efforts to improve the manuscript.

**0.1 Principal changes**

Many comments from both reviewers were related to the description of the generation of the training data and the implementation of the retrievals. In order to address these comments and avoid excessive growth of the main text of the manuscript, we have added appendices that explain the generation of the training data, the training itself and the application of the retrievals. We have also rewritten much of Sec. 1 and 2 to improve the presentation of both the scope of the manuscript and the design of the GPROF-NN retrievals.

In addition to that, an error in the simulated brightness temperatures for MHS was discovered and corrected. We have updated the results for MHS. While this improved the overall accuracy of the MHS retrievals, it did not affect the study's main findings. Furthermore, we identified and corrected a minor issue in our evaluation of GPROF that caused a slight overestimation of its accuracy. Again, this correction did not affect the conclusions of the paper.

**1 Comments from reviewer 1**

In what follows, line and figure numbers are given with respect to the revised manuscript.

**1.1 Specific comments**

**Reviewer comment 1**

Many of your readers may not be familiar with NNs terminology. Would you please highlight the advantages of the NNs, e.g., related CNN and QRNN methods, that today are popular in the satellite precipitation community compared to other ML techniques.

**Author response:**

We will add two paragraphs to the introduction, which discuss the advantages of neural networks for remote sensing retrievals in general as well as the specific advantages of CNNs and QRNNs.

**Changes in manuscript:**

**Changes starting in line 64:**

While GPROF is currently based on a data-driven method to solve Bayesian inverse problems, more general machine learning techniques have recently gained popularity for application in precipitation retrievals. Deep neural networks  (DNNs), which have enabled a number of  significant breakthroughs in different scientific fields (Silver et al., 2016; Jumper et al., 2021), have in recent years been explored for retrieving precipitation from satellite observations. Especially convolutional neural networks (CNNs) are appealing for this application because of their ability to leverage spatial patterns in image data. This property sets them apart from traditional retrieval methods and shallow machine-learning techniques, which are limited in their ability to use this information by computational complexity (Duncan et al., 2019) or the need for feature engineering or manual incorporation of spatial information through techniques such as convective-stratiform discrimination (Gopalan et al., 2010).

**Changes starting in line 97:**

The proposed algorithms are based on quantile regression neural networks (QRNNs, Pfreundschuh et al., 2018), which can be used to predict the posterior distribution of a Bayesian solution of the retrieval, given that the assumed a priori distribution of the Bayesian solution is the same as the distribution of the neural network's training data. Because of this, the GPROF-NN  retrievals can produce all of GPROF's retrieval outputs, which include a probability of precipitation and an uncertainty estimate of the predicted precipitation in the form of terciles of the posterior distribution.

**Reviewer comment 2**

As mentioned before, the study lacks an adequate review of the recent literature about using NNs for satellite precipitation estimation. I suggest some relevant papers (but are not limited to) that are worth reviewing. Please briefly explain already published works in the literature, their challenges/their methodologies, etc., and mention how your work is different from them. What are the open questions you try to address that the previous studies have not considered? As an example, in Lines 65-70: I understand that you specifically explore the potentials for NNs algorithm in GPROF, so please acknowledge other studies that have already discussed using spatial features in retrieving precipitation.

- Li, Z., Wen, Y., Schreier, M., Behrangi, A., Hong, Y. and Lambrigtsen, B., 2021. Advancing satellite precipitation retrievals with data-driven approaches: Is black box model explainable?. Earth and Space Science, 8(2), p.e2020EA001423.

- Afzali Gorooh, V., Akbari Asanjan, A., Nguyen, P., Hsu, K. and Sorooshian, S., 2022. Deep neural network high SpatioTEmporal resolution Precipitation estimation (Deep-STEP) using Passive Microwave and Infrared Data. Journal of Hydrometeorology.

- Sanò, P., Panegrossi, G., Casella, D., Marra, A.C., DAdderio, L.P., Rysman, J.F. and Dietrich, S., 2018. The passive microwave neural network precipitation retrieval (PNPR) algorithm for the CONICAL scanning Global Microwave Imager (GMI) radiometer. Remote Sensing, 10(7), p.1122.

- Ehsani, M.R., Zarei, A., Gupta, H.V., Barnard, K., Lyons, E. and Behrangi, A., 2022. NowCasting-nets: Representation Learning to Mitigate Latency Gap of Satellite Precipitation Products using Convolutional and Recurrent Neural Networks. IEEE Transactions on Geoscience and Remote Sensing.

**Author response:**

We agree with the reviewer that the discussion of previous work on precipitation retrieval with neural networks was insufficient in the first version of the manuscript. We will add a brief summary of previous work to the introduction and discuss the relevant differences to our work.

**Changes in manuscript:**

**Changes starting in line 72:**

 Shallow neural networks have long been used to retrieve precipitation ~~and hydrometeor profiles. Since the retrieval database has grown to a size of several hundred million entries it is perfectly suited for the application of deep neural networks , which scale very well to large amounts of data and are capable of learning complex relationships from them . In addition to this, a neural network based implementation has the advantage of allowing the integration of spatial information into~~ from PMW observations (Staelin and Chen, 2000; Surussavadee and Staelin, 2008) . The Passive microwave Neural network Precipitation Retrieval (PNPR) presented in Sanò et al. (2015); Sanò et al. (2016); Sanò et al. (2018) and the work by Tang et al. (2018) are among the more recent algorithms that use neural networks for retrieving precipitation from PMW observations. They employ relatively shallow neural networks and retrieve precipitation in a pixel-wise manner, thus neglecting spatial structure of the observations. Other recent work demonstrates the ability of CNNs to leverage spatial information in satellite observations. Examples of this are IR-based retrievals by Sadeghi et al. (2019), PMW-based precipitation detection (Li et al., 2021) and retrievals combining PMW with IR observations (Gorooh et al., 2022) and gauge measurements (Moraux et al., 2019).
A shortcoming of the aforementioned studies is that none of them addresses the inherent uncertainty of the precipitation retrievals. Retrieving precipitation from PMW observations constitutes an inverse problem, whose ill-posed character leads to significant uncertainties in the retrieval results. Traditionally, these uncertainties are handled using Bayesian statistics. However, because the algorithms mentioned above neglect the probabilistic character of the retrieval,

1.

2.

there is no way to reconcile them with the Bayesian approach.
 Moreover, existing precipitation retrievals that make use of DNNs (Moraux et al., 2019; Sadeghi et al., 2019; Li et al., 2021; Gorooh et al., 2022) are experimental retrievals that are currently not used operationally. The design of an operational retrieval algorithm for the GPM PMW observations needs to address a number of additional requirements, such as the handling of observations

from different sensors and the retrieval of multiple output variables. Furthermore, because GPM is an ongoing mission, continuity of the output variables must be ensured, which further constrains the design of the retrieval algorithm.

**Reviewer comment 3**

Lines 13 and 435: How do you define accuracy? Please elaborate on the reported improvements.

**Author response:**

We will rewrite the sentences to clearly state the observed improvements in the various metrics.

**Changes in manuscript:**

**Changes starting in line 14:**

Despite using the same input information as GPROF, the GPROF-NN  1D  retrieval improves the accuracy of the retrieved surface precipitation  for the GPM Microwave Imager (GMI)  from 0.079 mmh$^{-1}$ to 0.059 mmh$^{-1}$ in terms of  mean absolute error (MAE), from 76.1 % to 69.5 % in terms of symmetric mean absolute percentage error  (SMAPE) and from 0.797 to 0.847 in terms of correlation. The improvements for the Microwave Humidity Sounder (MHS) are from 0.085 mmh$^{-1}$ to 0.061 mmh$^{-1}$ in terms of MAE, from 81 % to 70.1 % for SMAPE and from 0.724 to 0.804 in terms of correlation. Comparable improvements are  found for the retrieved hydrometeor profiles and their column integrals  as well as  the detection of precipitation. Moreover, the ability of the retrievals to resolve small-scale variability is improved by more than 40 % for GMI and 29 % for MHS. The GPROF-NN 3D ~~retrieval over the performance of the GPROF-NN 1D retrieval, showing the added benefits of incorporating structural information into the retrieval. The effective resolution in along-track direction of the GPROF-NN 3D algorithm is reduced to 13.5 km, which is the upper limit imposed by the along track separation of consecutive scan lines. Comparable improvements are found also when the algorithms are applied to synthetic observations from the cross track scanning Microwave Humidity Sounder (MHS) sensor.~~ retrieval further improves the MAE to 0.043 mmh$^{-1}$, the SMAPE to 48.67 % and the correlation to 0.897 for GMI and 0.043 mmh$^{-1}$, 63.42 %$^2$ and 0.83 for MHS.

**Changes starting in line 483:**

The second important finding from this study is that by extending the retrieval to incorporate structural information, its accuracy can be further improved by  about 20 % in terms of MAE, MSE and SMAPE and 5 % in terms of correlation compared to the GPROF-NN 1D retrieval at the same time as the effective resolution in along track direction is decreased to its lower limit of 13.5 km for GMI and improved by 70 % for MHS.

**Reviewer comment 4**

Line 15: how do you see the spatial consistency in precipitation retrievals? Does this sentence refer to visualization of derived precipitation rates over Hurricane Harvey for one or two orbital tracks? Please report some statistics for the general spatial detection skills of your proposed models.

**Author response:**

The sentence that the reviewer is referring to was badly formulated. It was meant to refer to Fig. 9 and A1 from the original manuscript, which show that the improvements are consistent across the globe. However, since we consider this information to be of minor importance we will remove it from the revised version of the manuscript.

**Changes in manuscript**

- The sentence will be removed from the manuscript.

**Reviewer comment 5**

Lines 25, Section 3.3: I think the authors need to be cautious in reporting processing time and computational cost comparisons. It is obvious that pixel-wise predictions are faster compared to convolutional-based systems when models are trained and are ready to use. I mean the comparison between GPROF and NN 1D makes sense but including NN 3D is Comparing Apples to Oranges'. Processing time means when we have everything set up and ready, lets say we receive one or more orbital tracks (different channels have different footprints, etc.), how long does it take from getting a set of brightness temperatures (Level 1 product) to get the corresponding precipitation maps.

**Author response:**

The reported processing time measure exactly what the reviewer is requesting, i.e., the time required to process a full orbit of L1C observations augmented with ancillary data. Input and output data are the same for all retrievals and the processing time includes reading and writing of the data. The times are therefore directly comparable.

It seems that this has not been made sufficiently clear in the manuscript. We will rewrite the paragraph to emphasize that the timings are, in fact, comparable.

**Changes in manuscript:**

> **Changes starting in line 446:**
>
>  GPROF is used  to process PMW observations from a constellation of sensors spanning several decades of observations. Therefore, the processing time must not be excessively high. Although neural networks are  generally efficient to evaluate, this often assumes dedicated hardware, which  can not yet be expected to be available at the processing centers.
>
> We measure the processing time  required for retrieving precipitation from a full orbit of observations using a single CPU core of an Intel Xeon Gold 6234 CPU to assess the computational complexity of the three retrievals. The processing time here includes all steps from reading a GPROF input file to writing the corresponding output file. The input and output files are the same for all three algorithms, excluding, of course, differences in the retrieval results.
>
> The results are displayed in Fig. 14.  The processing of a single GMI file takes about 4 minutes for GPROF but only about 2 minutes for the GPROF-NN  retrievals. Because of the lower number of pixels in a single orbit, all retrievals are significantly faster for MHS. However, also here the GPROF-NN retrievals are significantly faster than GPROF. This shows that, even in the absence of dedicated hardware, the GPROF-NN retrievals process observations faster than the current implementation.

**Reviewer comment 6**

The data preprocessing steps are not clear in the methodology. I suggest summarizing all the training process and prediction (here means after train and validation stage) steps in a numbered list, especially for the CNN algorithm in the methodology section.

**Author response:**

We will add descriptions of the training and evaluation processes of the GPROF-NN retrievals to the manuscript. However, to avoid excessively increasing the length of the manuscript's main text, we will add these sections to the appendix.

**Changes in manuscript:**

- Section B2 will be added to the appendix, which describes the training processes for the GPROF-NN retrievals.

**Changes starting in line 600:**

**B.2 Training**

[revised manuscript text omitted]

- Section B3 will be added to the appendix, which describes the retrieval processing of the GPROF-NN algorithms.

**Changes starting in line 675:**

**B.3 Retrieval processing**

The data flow for the application of the GPROF and GPROF-NN retrievals is displayed in Fig. 1.1. The first step, which is common for all three retrievals, is the augmentation of the GPM L1C data with ancillary data. This process is performed by the GPROF preprocessor application. A detailed description of the ancillary data and its derivation can be found in the GPROF ATBD (Passive Microwave Algorithm Team Facility, 2022). The GPROF preprocessor produces a binary file containing the observations and ancillary data. This file serves as input for both GPROF and the GPROF-NN retrievals.

**GPROF-NN 1D**

The processing of input observations for the GPROF-NN 1D retrieval involves the following steps.

1. Flattening of retrieval inputs
2. Input normalization and encoding
3. Batch-wise evaluation of network and calculation of posterior statistics

4. Re-assembly into swath structure
5. Writing of GPROF binary output file

The observations and corresponding ancillary data are flattened into a list of inputs (1). All inputs are normalized and the categorical input variables are one-hot encoded using the same statistics as during training (2). The GPROF-NN 1D network is then used to calculate the posterior distributions of the retrieval targets from which the relevant posterior statistics are derived (3). Finally, the results for each pixel are re-assembled into the original swath structure and written to the GPROF binary output format, which is converted to HDF5 format in a separate step.

**GPROF-NN 3D**

The processing of input observations for the GPROF-NN 3D retrieval involves the following steps.

1. Input normalization and encoding
2. Input padding
3. Evaluation of network and calculation of the posterior statistics
4. Removal of padding

The input observations and ancillary data are normalized and encoded using the same statistics as during the training. The input observations are then padded using symmetric padding so that the dimension of the input data are a multiple of 32, which is required to ensure of symmetry requirements of the down- and up-sampling transformation in the neural network. The GPROF-NN 3D network is then evaluated and the posterior statistics are calculated. Because the GPROF-NN 3D network employs a fully-convolutional architecture, the results can be calculated for a full orbit of observations at once. However, since this may require excessive amounts of memory, the processing allows for optional tiling of the processing in along-track direction. After removal of the padding, the retrieval results are written to the same binary format that is used by GPROF-NN 1D and GPROF.

**Reviewer comment 7**

Data and method Section: Please clearly explain how many channels are used as inputs to the NN models? What type of resampling/rescaling/interpolation methods do you use? Different radiometers/imagers/sounders have different bands and resolutions, how do you address this problem?

[Figure]

Figure 1.1:  Data flow diagram for the application of the GPROF and GPROF-NN retrievals. The input for all retrieval is a GPROF preprocessor file, which is a binary file that contains the brightness temperatures and corresponding ancillary data. From this input all retrievals produce the retrieval results, which are stored in a common binary format before being converted to HDF5 files.

**Author response:**

The GPROF-NN retrieval use the same channels as GPROF and don't apply any interpolation apart what is done anyways by the GPROF preprocessing software. We will add a paragraph with this information as well as a table with the channels of the GMI and MHS sensors used in the study to the manuscript.

**Changes in manuscript:**

- Tab. 1.2 will be added to Sec. 2.1 of the manuscript, which shows the channels of the GMI and MHS sensors that are used in this study. In addition to that, the following paragraph will be added to Sec. 2.1 which discusses the handling of different channels in the retrieval database.

  > **Changes starting in line 135:**
  >
  > Since the available channels and the viewing geometries vary between the sensors of the GPM constellation, a separate database is generated for each sensor type. A crucial difference between the retrieval databases for GMI and the other sensors of the GPM constellation is that the database for GMI uses real observations, while the databases for the other sensors are constructed using simulations. The varying resolutions and viewing geometries of different sensors are taken into account by resampling and averaging the simulated observations and retrieval results to the observation footprints of the corresponding sensor. The channels of the GMI and MHS sensors that are used in this study are listed in Tab. 1.2.

- We will rewrite the first paragraph of Sec. 2.3 as follows.

  > **Changes starting in line 163:**
  >
  > The principal objective guiding the design of the GPROF-NN algorithms was to develop a  neural-network-based retrieval that operates on the same input data and provides the same output as GPROF so that it can  replace the current implementation in a future update.  Although GPROF's retrieval scheme is defined on independent pixels, the algorithm processes full orbits of observations and corresponding ancillary data. Both GPROF-NN retrievals were therefore designed to process the same input format as GPROF, which corresponds to each sensor's level 1C observations in their native spatial sampling, which, where required, is remapped to a common grid. The output from all retrievals is on the same grid as the input.

Table 1.2: Channels of the GMI and MHS sensors used for the retrievals in this study.

| Channel | Freq. [GHz] | Pol. |
|---------|-------------|------|
| GMI-1   | 10.6        | V    |
| GMI-2   | 10.6        | H    |
| GMI-3   | 18.7        | V    |
| GMI-4   | 18.7        | H    |
| GMI-5   | 23          | V    |
| GMI-6   | 37          | V    |
| GMI-7   | 37          | H    |
| GMI-8   | 89          | V    |
| GMI-9   | 89          | H    |
| GMI-10  | 166         | V    |
| GMI-11  | 166         | H    |
| GMI-12  | $183 \pm 3$ | V    |
| GMI-13  | $183 \pm 7$ | V    |

| Sensor | Freq. [GHz] | Pol. |
|--------|-------------|------|
| MHS-1  | 89          | V    |
| MHS-2  | 157         | V    |
| MHS-3  | $183 \pm 1$ | H    |
| MHS-4  | $183 \pm 3$ | H    |
| MHS-5  | 190.31      | V    |

**Reviewer comment 8**

How do you define 18 surface types? Are they generated by TELSEM classification algorithm? Please provide a clear picture of the source of data, pre-processing steps, etc. in this section. Background material for GPROF Algorithm is well described and cited in previous papers. So please summarize Section 2.2 and please explain more about the innovative parts of your investigation and the proposed models.

**Author response:**

The surface type information used by GPROF is described in detail in the ATBD. Instead of repeating the content of the ATBS in the manuscript, we will include a reference in the description of the retrieval database.

We will replace Sec. 2.2 with a brief summary and move the detailed description of GPROF to the appendix.

**Changes in manuscript**

- We will add the following paragraph to the end of Sec. 2.1

  **Changes starting in line 152:**

  A detailed description of the retrieval database and the derivation of the data it contains can be found in the GPROF ATBD (Passive Microwave Algorithm Team Facility, 2022). The training data for the GPROF-NN retrievals consists of the data from the retrieval database. The training data is stored in an intermediate format to simplify the loading of the data during training of the neural network. The format and the creation process of the training data is

described in detail in Sec. B.1 in the appendix.

- Sec. 2.2 is moved to Sec. A1 in the appendix and replaced with the following summary.

  **Changes starting in line 156:**

  The current implementation of GPROF uses a Bayesian scheme to retrieve precipitation and hydrometeor profiles, which works by resampling the profiles in the database based on the similarity of the observations and ancillary data. GPROF uses ancillary data to split the database into separate bins. This reduces the number of profiles for which weights must be computed and helps to constrain the retrieval. Moreover, the profiles in each bin are clustered to limit the number of profiles that need to be processed. A detailed description of the implementation of GPROF is provided in Sec. A in the appendix.

**Reviewer comment 9**

Please define all acronyms just the first time you use them. Then use the acronyms in the rest of the manuscript.

**Author response:**

We will revise the manuscript to make the use of acronyms more consistent.

**Reviewer comment 9**

Line 200: How many trainable parameters do the NNs algorithms have? Is one year of information enough for training and validating the models?

**Author response:**

The GPROF-NN 1D model has about 5 million, and the GPROF-NN 3D model has about 25 million parameters. The training data contains about 2 billion pixels with precipitation information. The fact that the trained models generalize well to unseen test data suggests that it is, in fact, possible to train these models sufficiently well with the available data.
The number of pixels used to evaluate the retrievals varies between 50 and 3 million for different sensors and retrieval types. Although the samples are not independent, we expect the number to be large enough to yield reliable statistics.

**Change in manuscript**

- We will add the numbers of parameters and sizes of the training datasets to the appendix. See changes in response to comment 6.

- We will add the table shown in Tab. 1.3 containing the number of pixels used for the evaluation to the beginning of Sec. 3.1.

- We will add the following paragraph to the beginning of Sec. 3.1:

  **Changes starting in line 288:**

  Tab. 1.3 lists the number of pixels with precipitation information used for testing the retrievals. The evaluation of the GPROF-NN 3D retrieval uses spatially contiguous scenes of the same size as the ones used during its training. Since these scenes typically do not cover all of the pixels with precipitation information, the test data for the GPROF-NN 3D retrievals contain fewer pixels that can be used for evaluation. The lower number of test pixels for MHS is due to the coarser resolution of the observations, which leads to a smaller number of observations over sea-ice and snow and an additional reduction of the pixels available for evaluation of the GPROF-NN 3D retrieval.

Table 1.3: The number of pixels with precipitation information in the test datasets used to evaluate the retrievals.

| Sensor | GPROF & GPROF-NN 1D | GPROF-NN 3D |
|--------|--------------------:|------------:|
| GMI    | 50 435 584          | 14 218 203  |
| MHS    | 24 975 877          | 4 945 165   |

**Reviewer comment 11**

Line 220: Please add some information about the training stage of models. For example, what are the size of 3D inputs to the CNN model in the training stage? how do you pre-process data to come up with input training samples?

**Author response:**

To provide a fuller picture of the training processes for the GPROF-NN retrievals, we will include a dedicated section in the appendix and move all information related to the training there.

**Changes in manuscript:**

- We will add a detailed description of the training processes for the retrievals to Sec. B2 in the appendix, which contains the information requested by the reviewer. See changes in response to comment 6.

**Reviewer comment 11**

Figure 5. Define regions A and B in the figure. Please explain the augmentation process in the training stage of CNN model development.

**Author response:**

We will add the missing labels for the two regions. A detailed description of the augmentation process will be provided in the appendix.

**Changes in manuscript:**

- We have updated Fig. 5 in the manuscript, which now looks as shown in Fig. 2.2

- We will include a detailed description of the augmentation process in Sec. B2.3 the appendix.

[Figure]

Figure 1.2: The updated Fig. 4, which will be included in the revised manuscript.

**Reviewer comment 12**

Figures 6 and 7. I do not see a good reason for the color used in these figures, and I find it confusing, commonly blue-red colors would reveal more features. Please find similar figures as the example in Utsumi et al 2020 paper. Also, please report some common statistical indices (related to scatterplots) or detection skill metrics to reveal the discrepancies/improvements. It is better to judge the performance based on statistical indices along with visual assessments. 3 - Utsumi, N., Turk, F.J., Haddad, Z.S., Kirstetter, P.-E., Kim, H., 2020. Evaluation of Precipitation Vertical Profiles Estimated by GPM-Era Satellite-Based Passive Microwave Retrievals. Journal of Hydrometeorology 22, 95112. https://doi.org/10.1175/JHM-D-20-0160.1

**Author response:**

It seems that Utsumi et al. (2021) use a spectral colormap that is similar or identical to the 'jet' colormap in their scatter plots. We would like to point out that the 'jet' colormap visually distorts the displayed data due to its non-linear and non-monotonic lightness profile (Thyng et al., 2016). The colormap that is used in the manuscript is perceptually uniform and ordered (matplotlib, 2022). This should make the color map less confusing and less misleading and is in accordance with general guidelines for data visualization (Borland and Ii, 2007).

Since we are not aware of any objective criteria that would justify the use of the 'jet' color map, we will keep the current color map in the revised version of the manuscript. The manuscript already reports a range of common statistical indices (Bias, MSE, MAE, SMAPE) to assess the accuracy of the retrieval. To provide an additional metric that is related to scatter plots, we will extend the manuscript to include the correlation.

We will, however, not include detection metrics here because the detection of precipitation is handled by the probability of precipitation and precipitation flag outputs of GPROF. These are evaluated separately in Sec. 3.1.2 of the manuscript.

**Changes in manuscript:**

- We will add the correlation to tables 2, 3, A1, A2, A3, A4, A6, A7, A8 as well as Fig. 7 (Fig. 8 in first version) and Fig. 11 (Fig. 12 in first version). The updated tables are shown in Tab. 1.4-1.13. The updated figures are shown in Fig. 1.5 and Fig. 1.3, respectively.

Table 1.4:  Mean error metrics and estimated standard deviation for  surface precipitation retrieved from GMI observations.

| Metric | GPROF | GPROF-NN 1D | GPROF-NN 3D |
|---|---|---|---|
| Bias [mm h$^{-1}$] |  $-0.0029 \pm 0.0001$ | $-0.0024 \pm 0.0001$ | $-0.0006 \pm 0.0001$ |
| MAE [mm h$^{-1}$] |  $0.0788 \pm 0.0001$ | $0.0585 \pm 0.0001$ | $0.0444 \pm 0.0001$ |
| MSE [mm h$^{-1}$] |  $0.1965 \pm 0.0001$ | $0.1379 \pm 0.0001$ | $0.0983 \pm 0.0001$ |
| SMAPE$_{0.01}$ [%] |  $76.0598 \pm 0.0139$ | $69.5382 \pm 0.0127$ | $56.0040 \pm 0.0181$ |
| Correlation | $0.7971$ | $0.8470$ | $0.8966$ |

**Reviewer comment 13**

Figure 7: What are the vertical white lines in the last panel of the figure? (Lowest right scatter plot)?

[Figure]

Figure 1.3: The updated Fig. 11 from the revised manuscript.

Table 1.5: Mean error metrics and estimated standard deviation for surface precipitation retrieved from MHS observations.

| Metric | GPROF | GPROF-NN 1D | GPROF-NN 3D |
|---|---|---|---|
| Bias   [mm h$^{-1}$] | 0.0070 ± 0.0001 | −0.0053 ± 0.0001 | 0.0017 ± 0.0002 |
|  | −0.0110 ± 0.0001 | −0.0066 ± 0.0001 | −0.0018 ± 0.0001 |
| MAE [mm h$^{-1}$] | 0.0948 ± 0.0001 | 0.0610 ± 0.0001 | 0.0524 ± 0.0002 |
|  | 0.0846 ± 0.0001 | 0.0609 ± 0.0001 | 0.0487 ± 0.0001 |
| MSE  [mm h$^{-1}$] | 0.3078 ± 0.0002 | 0.2088 ± 0.0002 | 0.1373 ± 0.0002 |
|  | 0.2317 ± 0.0001 | 0.1682 ± 0.0001 | 0.1087 ± 0.0001 |
| SMAPE$_{0.01}$  [%] | 80.9690 ± 0.0192 | 70.1140 ± 0.0189 | 63.4292 ± 0.0414 |
|  | 80.8641 ± 0.0190 | 68.4961 ± 0.0185 | 62.3086 ± 0.0377 |
| Correlation | 0.7239 ± 0.0000 | 0.8040 ± 0.0000 | 0.8400 ± 0.0000 |

Table 1.6: Like Tab. 1.4 but for convective precipitation.

| Metric | GPROF | GPROF-NN 1D | GPROF-NN 3D |
|---|---|---|---|
| Bias [mm h$^{-1}$] |  $-0.0007 \pm 0.0001$ | $-0.0015 \pm 0.0001$ | $-0.0011 \pm 0.0001$ |
| MAE [mm h$^{-1}$] |  $0.0322 \pm 0.0001$ | $0.0239 \pm 0.0001$ | $0.0204 \pm 0.0001$ |
| MSE [mm h$^{-1}$] |  $0.1927 \pm 0.0001$ | $0.1298 \pm 0.0001$ | $0.0854 \pm 0.0001$ |
| MAPE$_{0.01}$ [%] |  $118.151 \pm 0.0391$ | $107.1976 \pm 0.0378$ | $92.8343 \pm 0.0542$ |
| Correlation | $0.6380$ | $0.7467$ | $0.8152$ |

Table 1.7: Like Tab. 1.4 but for RWP.

| Metric | GPROF | GPROF-NN 1D | GPROF-NN 3D |
|---|---|---|---|
| Bias [mm h$^{-1}$] |  $0.0016 \pm 0.0000$ | $-0.0005 \pm 0.0000$ | $-0.0003 \pm 0.0000$ |
| MAE [mm h$^{-1}$] |  $0.0185 \pm 0.0000$ | $0.0127 \pm 0.0000$ | $0.0094 \pm 0.0000$ |
| MSE [mm h$^{-1}$] |  $0.0120 \pm 0.0000$ | $0.0086 \pm 0.0000$ | $0.0047 \pm 0.0000$ |
| MAPE$_{0.001}$ [%] |  $84.072 \pm 0.0287$ | $69.6918 \pm 0.0284$ | $61.8979 \pm 0.0315$ |
| Correlation | $0.8308$ | $0.8777$ | $0.9241$ |

Table 1.8: Like Tab. 1.4 but for IWP.

| Metric | GPROF | GPROF-NN 1D | GPROF-NN 3D |
|---|---|---|---|
| Bias [mm h$^{-1}$] | $-0.0022 \pm 0.0000$ | $-0.0006 \pm 0.0000$ | $-0.0002 \pm 0.0000$ |
| MAE [mm h$^{-1}$] |  $0.0204 \pm 0.0000$ | $0.0123 \pm 0.0000$ | $0.0085 \pm 0.0000$ |
| MSE [mm h$^{-1}$] |  $0.0186 \pm 0.0000$ | $0.0123 \pm 0.0000$ | $0.0053 \pm 0.0000$ |
| MAPE$_{0.001}$ [%] |  $88.26 \pm 0.0312$ | $67.3705 \pm 0.0305$ | $58.5831 \pm 0.0334$ |
| Correlation | $0.7897$ | $0.8637$ | $0.9350$ |

Table 1.9: Like Tab. 1.4 but for CWP.

| Metric | GPROF | GPROF-NN 1D | GPROF-NN 3D |
|---|---|---|---|
| Bias [mm h$^{-1}$] | $-0.0019 \pm 0.0000$ | $-0.0005 \pm 0.0000$ | $-0.0005 \pm 0.0000$ |
| MAE [mm h$^{-1}$] | $0.0268 \pm 0.0000$ | $0.0157 \pm 0.0000$ | $0.0115 \pm 0.0000$ |
| MSE [mm h$^{-1}$] | $0.0027 \pm 0.0000$ | $0.0015 \pm 0.0000$ | $0.0009 \pm 0.0000$ |
| MAPE$_{0.001}$ [%] | $62.2267 \pm 0.0100$ | $36.6584 \pm 0.0078$ | $27.9016 \pm 0.0087$ |
| Correlation | $0.8709$ | $0.9265$ | $0.9531$ |

Table 1.10: Like Tab. 1.5 but for convective precipitation.

| Metric | GPROF | GPROF-NN 1D | GPROF-NN 3D |
|---|---|---|---|
| Bias [mm h$^{-1}$] | $0.0072 \pm 0.0001$ $-0.0046 \pm 0.0001$ | $-0.0033 \pm 0.0001$ $-0.0023 \pm 0.0001$ | $0.0019 \pm 0.0001$ $-0.0012 \pm 0.0001$ |
| MAE [mm h$^{-1}$] | $0.0404 \pm 0.0001$ $0.0330 \pm 0.0001$ | $0.0280 \pm 0.0001$ $0.0281 \pm 0.0001$ | $0.0240 \pm 0.0001$ $0.0210 \pm 0.0001$ |
| MSE [mm h$^{-1}$] | $0.2119 \pm 0.0001$ $0.1674 \pm 0.0001$ | $0.1417 \pm 0.0001$ $0.1337 \pm 0.0001$ | $0.0908 \pm 0.0001$ $0.0824 \pm 0.0001$ |
| SMAPE$_{0.01}$ [%] | $109.8088 \pm 0.0483$ $108.8755 \pm 0.0480$ | $110.0042 \pm 0.0506$ $104.2921 \pm 0.0507$ | $95.5691 \pm 0.1124$ $94.0801 \pm 0.1057$ |
| Correlation | $0.5927$ | $0.6839$ | $0.7336$ |

Table 1.11: Like Tab. 1.5 but for RWP.

| Metric | GPROF | GPROF-NN 1D | GPROF-NN 3D |
|---|---|---|---|
| Bias [mm h$^{-1}$] | $0.0038 \pm 0.0000$ $-0.0002 \pm 0.0000$ | $-0.0012 \pm 0.0000$ $-0.0015 \pm 0.0000$ | $0.0004 \pm 0.0000$ $-0.0005 \pm 0.0000$ |
| MAE [mm h$^{-1}$] | $0.0246 \pm 0.0000$ $0.0210 \pm 0.0000$ | $0.0145 \pm 0.0000$ $0.0144 \pm 0.0000$ | $0.0120 \pm 0.0000$ $0.0116 \pm 0.0000$ |
| MSE [mm h$^{-1}$] | $0.0196 \pm 0.0000$ $0.0143 \pm 0.0000$ | $0.0126 \pm 0.0000$ $0.0102 \pm 0.0000$ | $0.0071 \pm 0.0000$ $0.0060 \pm 0.0000$ |
| SMAPE$_{0.001}$ [%] | $91.8829 \pm 0.0350$ $88.1093 \pm 0.0327$ | $76.0732 \pm 0.0349$ $75.4804 \pm 0.0335$ | $73.0157 \pm 0.0750$ $72.0101 \pm 0.0703$ |
| Correlation | $0.7591$ | $0.8346$ | $0.8785$ |

Table 1.12: Like Tab. 1.5 but for IWP.

| Metric | GPROF | GPROF-NN 1D | GPROF-NN 3D |
|---|---|---|---|
| Bias [mm h$^{-1}$] | $0.0051 \pm 0.0000$ | $-0.0011 \pm 0.0000$ | $0.0015 \pm 0.0001$ |
| | $-0.0035 \pm 0.0000$ | $-0.0009 \pm 0.0000$ | $-0.0008 \pm 0.0000$ |
| MAE [mm h$^{-1}$] | $0.0290 \pm 0.0000$ | $0.0120 \pm 0.0000$ | $0.0114 \pm 0.0001$ |
| | $0.0222 \pm 0.0000$ | $0.0123 \pm 0.0000$ | $0.0100 \pm 0.0000$ |
| MSE [mm h$^{-1}$] | $0.0270 \pm 0.0001$ | $0.0119 \pm 0.0001$ | $0.0078 \pm 0.0001$ |
| | $0.0137 \pm 0.0000$ | $0.0093 \pm 0.0000$ | $0.0060 \pm 0.0000$ |
| SMAPE$_{0.001}$ [%] | $92.5347 \pm 0.0364$ | $74.4043 \pm 0.0371$ | $68.5830 \pm 0.0760$ |
| | $92.0949 \pm 0.0357$ | $74.1056 \pm 0.0362$ | $69.5782 \pm 0.0762$ |
| Correlation | $0.8372$ | $0.8878$ | $0.9129$ |

Table 1.13: Like Tab. 1.5 but for CWP.

| Metric | GPROF | GPROF-NN 1D | GPROF-NN 3D |
|---|---|---|---|
| Bias [mm h$^{-1}$] | $0.0051 \pm 0.0000$ | $-0.0003 \pm 0.0000$ | $-0.0003 \pm 0.0000$ |
| | $-0.0008 \pm 0.0000$ | $0.0000 \pm 0.0000$ | $-0.0004 \pm 0.0000$ |
| MAE [mm h$^{-1}$] | $0.0299 \pm 0.0000$ | $0.0193 \pm 0.0000$ | $0.0156 \pm 0.0000$ |
| | $0.0264 \pm 0.0000$ | $0.0195 \pm 0.0000$ | $0.0149 \pm 0.0000$ |
| MSE [mm h$^{-1}$] | $0.0033 \pm 0.0000$ | $0.0016 \pm 0.0000$ | $0.0012 \pm 0.0000$ |
| | $0.0027 \pm 0.0000$ | | $0.0011 \pm 0.0000$ |
| SMAPE$_{0.001}$ [%] | $64.4350 \pm 0.0137$ | $46.5956 \pm 0.0112$ | $39.0331 \pm 0.0238$ |
| | $60.3897 \pm 0.0130$ | $47.2591 \pm 0.0114$ | $38.3892 \pm 0.0237$ |
| Correlation | $0.8598$ | $0.9194$ | $0.9369$ |

**Author response:**

The white lines in the scatter plot were caused by missing test samples at the corresponding cloud water path values. Because the input data for the GPROF-NN 3D retrieval must be assembled into spatially coherent scenes, the test data can't be fully identical to that used for GPROF and the GPROF-NN 1D retrievals. This is why the missing values only occurred for the results of the GPROF-NN 3D retrieval.

For the revised manuscript we will ensure that the bin sizes for the scatter plots are chosen in a way to avoid these white lines in the revised version of the manuscript.

**Changes in manuscript:**

- Fig. 7 has been updated and now looks like shown in Fig. 1.4.

[Figure]

Figure 1.4: The updated Fig. 6, which will be included in the revised version of the manuscript.

**Reviewer comment 14**

Figure 8 and the associated discussions in this section: The authors mentioned that they have used 18 surfaces classed. Did they regroup the precipitation over different surface types in order to report the statistics? Or here they just report 4 types out of 18? How do the proposed models perform on arid land surface types?

**Author response:**

We regrouped the land surface classes for Fig. 8 because we considered the plot to be too busy with all 18 classes included. We will add an explanation of the regrouping in the text.

The GPROF surface classes do not have an explicit class for arid surfaces but instead a range of classes of increasingly dense vegetation cover. To accomodate the reviewer's suggestion, we will split the 'vegetation' group into densely and sparsely vegetated surfaces.

**Changes in manuscript:**

- We will add the following sentence to the description of Fig. 8.

  > **Changes starting in line 323:**
  >
  > The figure displays bias, MSE, MAE, SMAPE, and correlation for principal surface types. The original GPROF surface types have been grouped into ocean (surface type 1), dense vegetation (surface types 3 - 5), sparse vegetation (6 - 7), snow (surface types 8-11), and coast (surface types 12-15).

- We will update the figure to display the accuracy over densely and sparsely vegetated land. The updated plot is shown in Fig. 1.5.

**Reviewer comment 15**

Line 303: Again, how many samples are used to calculate Bias, MSE, etc in each pixels/5-degree box?

**Author response:**

We will add an additional row of panels to the plot that shows the number of samples in each bin.

**Changes in manuscript:**

- Fig. 9 and Fig. A1 will be updated. The updated figures are shown in Fig. 1.6 and Fig. 1.7, respectively.

**Reviewer comment 16**

Section 3.2: This section presents a visualization of precipitation rates over one or two orbital tracks during Hurricane Harvey. Would you please report some basic statistical indices such as the probability of detection, missed ratio, etc.

[Figure]

Figure 1.5: The updated Fig. 7 from the original manuscript.

[Figure]

Figure 1.6: The updated Fig. 8, which will be included in the revised manuscript.

[Figure]

Figure 1.7: The updated Fig. C1, which will be included in the revised manuscript.

**Author response:**

We will add tables with the requested statistics to the evaluation.

**Changes in manuscript:**

- We will add two tables shown in Tab. 1.14 and Tab. 1.15 which the Bias, MSE, MAE, Correlation, Precision and Recall for the GMI and MHS overpasses, respectively

- The discussion of the results in Sec. 3.2 will be rewritten to include these new results.

> **Changes starting in line 420:**
>
> A quantitative assessment of the retrieval results is provided in Tab. 1.14, which shows bias, MSE, and correlation, as well as the precision and recall of the retrieved precipitation flag. The precision is the fraction of correctly detected raining pixels of all pixels predicted to be raining, and the recall is the fraction of all truly raining that is correctly detected.
>
> All statistics were calculated using the CMB product and the MRMS ground-based measurements as a reference. The reference measurements were averaged to the footprint of the GMI 18.7 GHz channel, taking into account the rotation of the pixels across the swath. Only measurements with a radar quality index is at least 0.8 were used for the comparison against MRMS retrievals.
>
> The accuracy of all retrievals is lower when compared to MRMS than when compared to CMB. This is likely because all GPROF retrievals are designed to reproduce the retrieval database, which is to a large extent derived from the CMB product. The GPROF-NN retrievals yield more accurate results than

GPROF across all considered metrics except for the recall, which is lower for GPROF-NN 1D than for GPROF. Interestingly, GPROF-NN 1D achieves lower MAE, MSE, and bias as well as higher correlation in the comparison against MRMS, while the two perform similarly in the comparison against CMB.

**Changes starting in line 439:**

in Accuracy metrics for comparing the MHS retrievals with MRMS are shown in Tab. 1.15. The MRMS measurements were averaged to the MHS observation footprints taking into account the changes in footprint size and shape across the swath. For MHS, GPROF has the lowest Bias, MAE, and MSE and higher recall than GPROF-NN 1D. These results do not show any clear improvements for the GPROF-NN retrievals. However, the GPROF-NN 3D retrievals improve the retrieval in terms of all metrics compared to GPROF-NN 1D, suggesting that the GPROF-NN 3D can make use of the spatial information in the observations despite being trained on simulated observations.

Table 1.14: Accuracy metrics for surface precipitation retrieved from GMI PMW observations of hurricane Harvey for the overpass on 2017-08-25 at 11:50:00 UTC. Each metric is calculated with respect to the surface precipitation from the CMB product as well as the surface precipitation from MRMS as reference.

| Retrieval | Bias [mm h$^{-1}$] | | MSE [(mm h$^{-1}$)$^2$] | | Correlation | | Precision | | Recall | |
|-----------|------|------|------|------|------|------|------|------|------|------|
| | CMB | MRMS | CMB | MRMS | CMB | MRMS | CMB | MRMS | CMB | MRMS |
| GPROF | 0.346 | 0.355 | 2.691 | 8.299 | 0.892 | 0.651 | 0.9 | 0.82 | 0.82 | 0 |
| GPROF-NN 1D | 0.245 | 0.145 | 1.944 | 4.927 | 0.914 | 0.701 | 0.95 | 0.9 | 0.90 | 0 |
| GPROF-NN 3D | 0.248 | 0.184 | 1.953 | 6.12 | 0.923 | 0.676 | 0.95 | 0.9 | 0.90 | 0 |

Table 1.15: Accuracy metrics for surface precipitation retrieved from MHS PMW observations of hurricane Harvey for the overpass on 2017-08-25 at 13:58 UTC. The metrics calculated against the MRMS surface precipitation estimates.

| Retrieval | Bias [mm h$^{-1}$] | MSE [(mm h$^{-1}$)$^2$] | Correlation | Precision | Recall |
|-----------|------|------|------|------|------|
| GPROF | 0.11 | 2.602 | 0.749 | 0.88 | 0.12 |
| GPROF-NN 1D | 0.259 | 4.031 | 0.751 | 0.9057 | 0.094 |
| GPROF-NN 3D | 0.152 | 3.168 | 0.759 | 0.948 | 0.052 |

**Reviewer comment 17**

Figures 13 and 14. Please show CMB and MRMS products in both figures. Please use the commonly used blue red colorbar and colormap for presenting precipitation rates.

Revise the figure in a way that the rain rates less than 1 mm/h are not eliminated. I see that the figures are patchy, and the spatial patterns of precipitation rates are not obvious. Please remove the colorful background from figure 14. and again, it is miss leading when the precipitation rates less than 1 mm/h in panels c, d, g, h is not shown in the figures.

**Author response:**

While it is not possible to show the CMB product for the overpass of MHS since the GPM overpass occurred at a different time, we will add the MRMS measurements to the GPM overpass. Moreover, we will revise the plots to show precipitation rates across the full swaths on a logarithmic color scale without omitting the precipitation rates less than 1 mm/h.
We will not change the colormap to jet plots based on the arguments presented in the response to comment 12.

**Change in manuscript:**

- We will update Fig. 13 and Fig. 14 according to the reviewers suggestions. The updated figures are shown in Fig. 1.8 and Fig. 1.9, respectively.

[Figure]

Figure 1.8: The updated Fig. 12, which will be included in the revised manuscript.

**Reviewer comment 18**

Section 3.3 as mentioned before, I suggest removing this part or please add more information for different stages of developing NN 1D and NN 3D models, to avoid confusion for the readers. I understand that GPUs, TPUs, etc. can be used to train deep neural

[Figure]

Figure 1.9: The updated Fig. 13, which will be included in the revised manuscript.

networks, and the processing time when everything is ready for the model can be fast for pixel-wise NN 1D. Using NN 3D may be relatively fast in precipitation estimation (prediction phase), but the data preprocessing takes time and is not mentioned here.

**Author response:**

See response to comment 5.

**Reviewer comment 19**

Line 413, 461: Please avoid using simply replacing or developing. It is not simple!

**Author response:**

We will rewrite this sentence in the revised version of the manuscript.

**Changes in manuscript**

> **Changes starting in line 461:**
>
> The evaluation of the GPROF-NN 1D algorithm against GPROF, showed that retrieval accuracy as well as effective resolution can be improved  by replacing the current retrieval method with a fully-connected neural network.

> **Changes starting in line 522:**
>
>  Both GPROF-NN retrievals have been designed

as a drop-in replacement for GPROF and can be directly used in the operational GPM processing pipeline. The results presented in this study show that, given a perfect retrieval database, considerable improvements in the accuracy of GPROF can be achieved by replacing the current  Bayesian scheme with a deep neural network that processes pixels independently.

**Reviewer comment 20**

Line 440: Again, please review the study by Li et al. 2021 and more recent ones that use CNN and PMW data are a PMW data are a part of their input datasets. It is worth mentioning previous works at least in the introduction. Also, it is already established that using neighboring information (spatial features) improves the satellite retrievals both in capturing the amount and the location of events.

- Li, Z., Wen, Y., Schreier, M., Behrangi, A., Hong, Y. and Lambrigtsen, B., 2021. Advancing satellite precipitation retrievals with data-driven approaches: Is black box model explainable?. Earth and Space Science, 8(2), p.e2020EA001423.

- Afzali Gorooh, V., Akbari Asanjan, A., Nguyen, P., Hsu, K. and Sorooshian, S., 2022. Deep neural network high SpatioTEmporal resolution Precipitation estimation (Deep-STEP) using Passive Microwave and Infrared Data. Journal of Hydrometeorology.

and many more,...

**Author response:**

It was, of course, not our intent to claim that we were the first to make use of spatial information in a PMW retrieval. We will reformulate this paragraph to avoid this misunderstanding.

**Changes in manuscript:**

> **Changes starting in line 509:**
>
> ~~The use of structural information for precipitation retrievals is common practice in algorithms based on infrared observations (Sorooshian et al., 2000; Hong et al., 2004) and the potential benefits of CNN based retrievals have been shown in Sadeghi et al. (2019). While basic structural information has been used in earlier PMW precipitation retrieval algorithms, as e.g. by Kummerow and Giglio (1994), we are not aware of any other operational PMW algorithms that incorporate structural information using CNNs~~ Because precipitation exhibits distinct spatial patterns in satellite observations, many algorithms make use of this information to improve precipitation retrievals (Kummerow and Giglio, 1994; Sorooshian et al., 2000; Hong et al., 2004). Our results confirm that CNNs learn to leverage this information directly from the satellite

imagery and that it can notably improve the retrieval accuracy, which is in agreement with the findings from other precipitation retrievals that employ CNNs (Tang et al., 2018; Sadeghi et al., .

**1.1.4 Reviewer comment 21**

Line 440-445, 452: No evidence has been reported or shown that the model is trained properly. At least please mention the number of samples in the training and testing process, how do the authors select the hyperparameters? How many parameters do the NN models have compared to GPROF? The Hurricane Harvey event was just a visual representation of retrievals. By adding statistic indices such as pixel- or window-wise correlation, false alarm, missed ratio, etc., the reader can find the improvements and the differences (not only by reporting average bias and visual assessments).

**Author response:**

The results presented in Sec. 3 of the manuscript clearly show that the neural networks achieve higher retrieval accuracy on the unseen test data than GPROF. This would not be possible if the networks weren't trained properly. While the assessment of the retrievals over hurricane Harvey can give some indication over how well the retrievals work, a quantitative assessment against independent measurements has to be interpreted carefully because the reference measurements will themselves be affected by uncertainties. In addition to that, the the MHS retrievals are affected by simulation errors, which may limit the accuracy of the GPROF-NN retrievals.

We will reformulate the discussion of the overpasses to emphasize the above points. As mentioned in the response to comment 17, we will extend the evaluation of the hurricane Harvey overpasses to include the requested metrics. We will also include the sizes of the GPROF-NN neural networks and the training data in the new section in the appendix that describes the training data.

**Changes in manuscript**

- We will add tables with the requested statistics to the assessment of the Hurricane overpasses. See response to reviewer comment 16.

- The discussion of the results from the hurricane overpasses will be extended as follows.

  **Changes starting in line 457:**

  The quantitative assessment of the accuracy of the MHS retrievals of hurricane Harvey did not show any clear improvements for the GPROF-NN retrievals compared to GPROF. This can be due to multiple reasons. Firstly, the hurricane constitutes an extreme event and it is likely that the instantaneous MRMS precipitation rates used as reference measurements are themselves

affected by considerable uncertainties. Secondly, given that the bulk of the precipitation in the considered scene is intense and over ocean, GPROF can be expected to work quite well. This makes it less likely to find clear improvements in this particular scenario. Finally, the accuracy of the neural-network based retrievals may be limited by the modeling error of the simulations in the retrieval database. In principle, simulation errors could even cause the GPROF-NN retrievals to be less accurate than GPROF for real observations. Should this really be the case, the demonstrated potential of the GPROF-NN retrievals would imply that the quality of the simulations in the GPROF database limits the accuracy of the GPM PMW precipitation measurements and that future work to should focus on improving the simulations.

- The number of parameters of the models will be included in the newly added Sec. B2 in the appendix that described the training of the neural network models. See response to comment 6.

**1.1.5 Reviewer comment 22**

Line 455: Quoting from the manuscript an additional neural network model was required to transform the data from the retrieval database into a form that is amenable for training a CNN..., I invite the authors to clearly explain the process in the manuscript. It is not clear!

**Author response:**

We will add an extended description of the generation of the training data in the revised version of the manuscript. It will describe the intermediate retrieval that is used to generate the training data for sensors other than GMI.

**Changes in manuscript**

- Section B1 will be added to the manuscript, which describes the generation of the training data for all sensors including the retrieval used to extend the simulated brightness temperatures.

  **Changes starting in line 580:**

  **B.6 Training data**

  **Structure**

  The training data for the GPROF-NN retrievals is stored in an intermediate format to simplify the loading of the data during the training process. The data is organized into scenes measuring 221 contiguous GMI pixels in both alongand across-track directions. Each scene contains the GMI L1C brightness temperatures and the corresponding values of the retrieval quantities at the center of the GMI swath. For sensors other than GMI, each scene also contains the simulated brightness temperatures of the corresponding sensor.

**Generation**

An overview of the data flow for the training data generation for the GPROF-NN retrievals is displayed in Fig. 2.3. The training data originates from four primary sources: The GPROF simulator files, which contain surface precipitation, hydrometeor profiles, and simulated brightness temperatures for an orbit of the GPM combined product. Surface precipitation over snow surfaces and sea-ice are derived from MRMS and ERA5 data, respectively. This data is matched with GMI L1C-R brightness temperatures. The data is split into non-overlapping scenes measuring 221 scans and 221 pixels. For sensors other than GMI, the brightness temperature differences between actual and simulated GMI observations are included and added to the simulated observations to provide a first-order correction for the modeling error in the observations.

Simulated brightness temperatures are only available where the hydrometeor profiles and surface precipitation is known, i.e., at the center of the training scenes. Because this is insufficient to train a CNN with 2D convolutions for sensors other than GMI, an intermediate simulator retrieval is trained to retrieve simulated brightness temperatures from GMI observations. This retrieval the applied to the training data to fill in the simulated brightness temperatures across the entire GMI swath. The simulator neural network uses the same architecture as GPROF-NN 3D retrieval.

**1.1.7 Reviewer comment 23**

Line 477: I suggest replacing warming climate with something like changing climate.

**Author response:**

While we acknowledge the reviewer's suggestion, we are not aware of any objective arguments for such a change and will therefore not implement it in the revised version of the manuscript.

[Figure]

Figure 1.10: Data flow diagram for the generation and organization of the GPROF-NN training data. Grey rectangles represent datasets, and colored rectangles with rounded corners represent algorithms.

**2 Comments from reviewer 2**

In what follows, line and figure numbers are given with respect to the revised manuscript.

**2.1 Major comments**

**Reviewer comment 1**

The validation scheme is not quite convincing. What you did is: using part of the training as the validation dataset (near L255, first three days of every month from the retrieval database). This can be a major issue since it is shown that GPROF-NN and GPROF-3D is better than GPGORF-Bayesian. The better performance from GPROF-NN and 3D may result from the over- fitting of the Neural network. I am particularly concerned about the over-fitting issue for surface precipitation from GPROF-NN-3D (Fig. 6, bottom left panel, it seems that the vast majority of the pixels are on 1-by-1 line from 0.1 to 10 mm/hr)
Why not use 1-yr independent data (say, 2020 DPR) to validate your results? Based on Fig. 15, it takes about 120 250 seconds per orbit to get the results. I highly recommend to redo the validation.

**Author response:**

It seems that the reviewer has misunderstood our evaluation scheme. We have, of course, not evaluated the model on a sub-set of the data that was used for training. Instead, only days 6 until 31 of every month have been used for training, while days 1 until 3 were used for the evaluation. We will revise this section to make this more clear.
The alternative validation proposed by the reviewer is not really suitable for this study. Firstly, it is not clear whether one year of DPR data would provide sufficiently many collocations with MHS. Secondly, the use of independent validation data introduces an additional error source into the evaluation. Since the declared aim of the study was to assess only the impact of the retrieval method, we consider the validation against independent measurements outside the scope of this study.
We will extend the introduction of the manuscript to highlight these difficulties and better define the scope of the manuscript.

**Changes in manuscript:**

- We will add a paragraph to the introduction that discusses the difficulties of evaluating precipitation retrievals and explains the motivation for our evaluation scheme.

**Changes starting in line 102:**

Before a retrieval can replace the current operational version of GPROF, it is imperative to establish its ability to improve the retrieval accuracy to avoid degradation of the GPM products. A balanced evaluation of the accuracy of precipitation retrievals is difficult because it depends on the statistics of the data used in the assessment. Data-driven retrievals generally yield the most accurate results when evaluated on data with the same distribution as the data used for their training. At the same time, evaluation against independent measurements may distort the evaluation when these measurements deviate significantly from the training data. In this study, the retrieval performance of the GPROF-NN algorithms is evaluated and compared to that of GPROF using a held-out part of the retrieval database. This provides the most direct estimate of the benefits of the neural network based retrievals because it avoids the distorting effects of using test data from a different origin. Moreover, the nominal accuracy of both the GPROF and GPROF-NN algorithms provides a reference for future validation against independent measurements.

- We will add a paragraph that clearly states that the data we use for evaluation is not used during the training of the neural network retrievals.

**Changes starting in line 285:**

The held-out test data comprises observations from the first three days of every month from the retrieval database. ~~It should be noted that we have deliberately limited this evaluation to data from the retrieval database in order to isolate the effect of the retrieval algorithm from that of the database. We conclude this section with a case study of overpasses of Hurricane Harvey. These results are based on real observations and thus provide an indication to what extent the performance on the retrieval database can be expected to generalize to real observations~~ This data has not been used for training the neural network retrievals. It is, however, derived from the same data sources and thus stems from the same distribution as the training data.

**Reviewer comment 2**

The most noticeable improve from NN method is for the very light precipitation (<0.1 mm/hr to 0.01 mm/hr, Fig. 6, 1st column). Then the question is: such light precipitation is really beyond the detection capability of both GMI and MHS. Many previous studies showed that the detection threshold value is around 0.2 mm/hr (e.g., Munchak, S. Joseph, and Gail Skofronick- Jackson. "Evaluation of precipitation detection over various surfaces from passive microwave imagers and sounders." Atmospheric Research

131 (2013): 81-94.). In other words, even if GPROF-NN and GPROF-NN-3D can make this light surface precipitation retrieval better, it is difficult to justify physically you did correctly since these light precipitation are beyond the GMI/MHS detection capability.

**Author response:**

We do not agree with the reviewer on this point. The findings from Munchak and Skofronick-Jackson (2013) are themselves based on a retrieval. It is therefore possible that a more advanced retrieval method can improve the detection threshold of the sensors.

In fact, when we apply the technique from Munchak and Skofronick-Jackson (2013) but instead of the cost function of their variational retrieval use the probability of precipitation retrieved by GPROF, we obtain the graph shown in Fig. 2.1. The detection thresholds for GPROF, GPROF-NN 1D and GPROF-NN 3D are about 0.15, 0.08 and $0.04\,\mathrm{mm\,h^{-1}}$, respectively, as can be seen from the graph. This indicates that the GPROF-NN 1D (3D) retrieval increases the minimum sensitivity of GMI by a factor of 2 (4) and that there is a precipitation signal even at precipitation rates below $0.1\,\mathrm{mm\,h^{-1}}$ Moreover, the simple fact that the neural network based retrievals can improve the retrieval of weak precipitation indicates the presence of a signal from that precipitation. If that wouldn't be the case, there would be no way for the neural network based retrievals to make better predictions than GPROF.

[Figure]

Figure 2.1: Factional occurence of rain (solid lines, left y-axis) and corresponding mean precipitation (dotted lines, right y-axis). This figure is similar to Fig. 6 in Munchak and Skofronick-Jackson (2013) but uses the retrieved probability of precipitation instead of the OEM cost.

**2.2 Minor comments**

**Reviewer comment 1**

Line 3: at such high temporal resolution to at three hours temporal resolution, because the temporal resolution from PMWs is rather low (even with the constellation), compared with IR (can be 10 minutes or less).

**Author response:**

We will reformulate this first part of the abstract to improve the description of the role of PMW observations.

**Changes in manuscript:**

- We will reformulate the first paragraph of the abstract.

   **Changes starting in line 1:**

   The Global Precipitation Measurement (GPM) mission  measures global precipitation at a temporal resolution of  a few hours to enable close monitoring of the global hydrological cycle.  GPM achieves this by combining observations from a space-borne precipitation radar, a constellation of passive microwave (PMW) sensors and geostationary satellites.

**Reviewer comment 2**

Line 23: can be expect to can be expected

**Author response:**

We will reformulate the corresponding paragraph and corrected the mistake.

**Changes in manuscript**

   **Changes starting in line 23:**

   Application of the  retrievals to GMI observations of hurricane Harvey shows moderate improvements when compared to co-located GPM combined and ground-based radar measurements indicating that the improvements at least partially carry over to  assessment against independent measurements.

**Reviewer comment 3**

Line 33: 3 hours to three hours to be consistent with what you have used in the abstract.

**Author response:**

We will replace 'three' with 'few' in the revised version of the manuscript because IMERG actually achieves a temporal resolution of 30 minutes.

**Changes in manuscript**

**Changes starting in line 21:**

The Goddard Profiling Algorithm (GPROF, Kummerow et al. (2015)) is the operational precipitation retrieval algorithm for the passive microwave (PMW) observations from the  radiometer constellation of the Global Precipitation Measurement (GPM, Hou et al. (2014)), whose objective is to provide consistent global measurements of precipitation at a temporal resolution of  a few hours .

**Reviewer comment 4**

Line 34: GPM level 3 retrieval products probably need to change to GPM level 3 retrieval product. My understanding is that: there is only one Level 3 product (ie.., IMERG). Also, it may be better to briefly introduce IMERG via one sentence since IMERG is more widely used and known. But not so many studies realized that PMWs form the foundation for IMERG.

**Author response:**

Although, officially, there are many GPM level three products it is true that IMERG is probably the most popular one. We will therefore reformulate the sentence in the revised version of the manuscript to mention IMERG.

**Changes in manuscript**

**Changes starting in line 36:**

 a few hours. The precipitation retrieved by GPROF serves as input for the Integrated Multi-Satellite Retrievals for GPM (IMERG), which can be considered the state-of-the-art of global precipitation measurements.

**Reviewer comment 5**

Line 134: I believe there are two typos in the multiple-variate normal distribution: (1) $n_i$ should be 1; and (2) $2\pi$, should be $(2\pi)^n$ (n is the variable number, should be 13 TBs). Please double check.

**Author response:**

We would like to thank the reviewer for pointing out this mistake. However, instead of removing $n_i$ from the Eq. (2), we will remove it from Eq. (1) and move the $2\pi$ inside the determinant.

**Changes in manuscript:**

- Equation (1), which has been renamed to (A1), will look as follows in the revised version of the manuscript:

$$\int_{\mathbf{x}} \mathbf{x} p(\mathbf{x}|\mathbf{y}) \, d\mathbf{x} = \int_{\mathbf{x}} \mathbf{x} \frac{p(\mathbf{y}|\mathbf{x})p(\mathbf{x})}{p(\mathbf{y})} \, d\mathbf{x} \approx \frac{\sum_i p(\mathbf{y}|\mathbf{x}_i)\mathbf{x}_i}{\sum_i p(\mathbf{y}|\mathbf{x}_i)}. \tag{2.1}$$

- Equation (2), which has been renamed to (A2), will look as follows in the revised version of the manuscript:

$$p(\mathbf{y}|\mathbf{x}_i) = \frac{n_i}{\sqrt{\det(2\pi\mathbf{S})}} \exp\left\{ -\frac{1}{2}(\mathbf{y} - \mathbf{y}_i)^T \mathbf{S}^{-1} (\mathbf{y} - \mathbf{y}_i) \right\} \tag{2.2}$$

**Reviewer comment 6**

Line 157: as well to as well as

**Author response:**

We will correct this in the revised version of the manuscript.

**Changes in manuscript:**

**Changes starting in line 179:**

For the GPROF-NN retrievals, the predicted CDF is used to derive most likely and mean surface precipitation (the latter of which is identical to the solution that would have been obtained with common mean squared error regression), the terciles of the posterior distribution as well as the probability of precipitation.

**Reviewer comment 7**

Fig. 5. I dont understand what is the color squares. In the caption, it is mentioned Grey squares mark equilaterals with ..., what are the colored squares? I guess grey and color squares are the same??

**Author response:**

The shading in the background just shows the GMI brightness temperatures. Grey squares are drawn on top to better show the distorting effect of the conical viewing geometry. We will update the figure caption to hopefully make the figure easier to understand.

**Changes in manuscript**

- The caption of Fig. 4 in the manuscript will be updated. The updated caption is shown in Fig. 2.2

[Figure]

Figure 2.2: The effect of GMIs conical viewing geometry on observed features. Panel (a) displays geolocated observations of the 10.6 GHz channel (colored background). Grey squares mark equilaterals with a side length of 200km oriented along the swath. The highlighted stripe located at the swath center marks the region where the values of the retrieved variables are known. Panel (b) shows the same observations viewed as an image on a uniform grid. Panel (c) shows six synthetically generated training inputs based on two input regions marked in Panel (b). The first row shows three synthetic samples that simulate the effect of viewing the input in region A at a different position across the GMI swath. The second row shows the corresponding transformations for the input in region B.

**Reviewer comment 8**

Line 250: To obtain two-dimensional training scenes that are sufficiently wide to train a CNN, we make use of an intermediate CNN based model to retrieve simulated brightness temperatures across the full GMI swath. Please explain in more details how you did this (i.e., extend from DPR swath to the whole GMI swath).

**Author response:**

We will add a section to the newly added appendix which describes the process of generating the GPROF-NN 3D training data for sensors other than GMI.

**Changes in manuscript:**

- A description of the generation of the training data will be added to Sec. B1 of the revised manuscript.

  **Changes starting in line 580:**

  ## B.1 Training data

  ### Structure

  The training data for the GPROF-NN retrievals is stored in an intermediate format to simplify the loading of the data during the training process. The data is organized into scenes measuring 221 contiguous GMI pixels in both along- and across-track directions. Each scene contains the GMI L1C brightness temperatures and the corresponding values of the retrieval quantities at the center of the GMI swath. For sensors other than GMI, each scene also contains the simulated brightness temperatures of the corresponding sensor.

  ### Generation

  An overview of the data flow for the training data generation for the GPROF-NN retrievals is displayed in Fig. 2.3. The training data originates from four primary sources: The GPROF simulator files, which contain surface precipitation, hydrometeor profiles, and simulated brightness temperatures for an orbit of the GPM combined product. Surface precipitation over snow surfaces and sea-ice are derived from MRMS and ERA5 data, respectively. This data is matched with GMI L1C-R brightness temperatures. The data is split into non-overlapping scenes measuring 221 scans and 221 pixels. For sensors other than GMI, the brightness temperature differences between actual and simulated GMI observations are included and added to the simulated observations to provide a first-order correction for the modeling error in the observations.

  Simulated brightness temperatures are only available where the hydrometeor profiles and surface precipitation is known, i.e., at the center of the training scenes. Because this is insufficient to train a CNN with 2D convolutions for sensors other than GMI, an intermediate simulator retrieval is trained to retrieve simulated brightness temperatures from GMI observations. This retrieval the applied to the training data to fill in the simulated brightness

[Figure]

Figure 2.3: Data flow diagram for the generation and organization of the GPROF-NN training data. Grey rectangles represent datasets, and colored rectangles with rounded corners represent algorithms.

temperatures across the entire GMI swath. The simulator neural network uses the same architecture as GPROF-NN 3D retrieval.

**Reviewer comment 9**

Both Figure 6 and Figure 7 are over all surface types (i.e., land, ocean, coast, ect)? Please clarify.

**Author response:**

Yes, both plots use all surface types. We will add the clarification to the manuscript.

**Changes in manuscript:**

**Changes starting in line 297:**

Scatter plots of the retrieval results for these five quantities over all surfaces are displayed in Fig. 5 for GMI and Fig. 6 for MHS.

**Reviewer comment 10**

Throughout the paper, I did not find which MHS you used (maybe I missed it). Please specify MHS onboard which satellite (there are 5 MHSs, I think).

**Author response:**

The GPROF database doesn't distinguish between the different instances of the MHS sensors, which is why the platform is not stated in the manuscript. For the observations of hurricane Harvey the platform is stated in l. 432.

**Reviewer comment 11**

Line 440: we are not aware of any other operational PMW algorithms that incorporate structural information using CNNs. Yes, you are probably correct that nobody is using structural information via CNN. However, structure information has long been used for retrieval from the TRMM era. The land algorithm did by Ferrao group used quite a bit structural information (spatial information) before GPROF transitioned into all Bayesian technique. (see Estimation of convective/stratiform ratio for TMI pixels in Gopalan, Kaushik, et al. "Status of the TRMM 2A12 land precipitation algorithm." Journal of Atmospheric and Oceanic Technology 27.8 (2010): 1343-1354.) A more recent paper to use the spatial information (Guilloteau, Clément, and Efi Foufoula-Georgiou. "Beyond the pixel: Using patterns and multiscale spatial information to improve the retrieval of precipitation from spaceborne passive microwave imagers." Journal of atmospheric and oceanic technology 37.9 (2020): 1571-1591.). It will be good to briefly discuss how previous studies are using the structural information.

**Author response**

We would like to thank the reviewer for this suggestion and the provided references. We will extend our discussion of the use of spatial information in previous retrievals.

**Changes in manuscript:**

- We will add a paragraph to the introduction that discusses machine learning and the use of spatial information in remote sensing retrievals.

  **Changes starting in line 64:**

  While GPROF is currently based on a data-driven method to solve Bayesian inverse problems, more general machine learning techniques have recently gained popularity for application in precipitation retrievals. Deep neural networks  (DNNs), which have enabled a number of  significant breakthroughs in different scientific fields (Silver et al., 2016; Jumper et al., 2021), have in recent years been explored for retrieving precipitation from satellite observations. Especially convolutional neural networks (CNNs) are appealing for this application because of their ability to leverage spatial patterns in image data. This property sets them

apart from traditional retrieval methods and shallow machine-learning techniques, which are limited in their ability to use this information by computational complexity (Duncan et al., 2019) or the need for feature engineering or manual incorporation of spatial information through techniques such as convective-stratiform discrimination (Gopalan et al., 2010).

- We will also reformulate the discussion of the use of spatial information to include the reference provided by the reviewer.

> **Changes starting in line 509:**
>
> ~~The use of structural information for precipitation retrievals is common practice in algorithms based on infrared observations (Sorooshian et al., 2000; Hong et al., 2004) and the potential benefits of CNN based retrievals have been shown in Sadeghi et al. (2019). While basic structural information has been used in earlier PMW precipitation retrieval algorithms, as e.g. by Kummerow and Giglio (1994), we are not aware of any other operational PMW algorithms that incorporate structural information using CNNs~~ Because precipitation exhibits distinct spatial patterns in satellite observations, many algorithms make use of this information to improve precipitation retrievals (Kummerow and Giglio, 1994; Sorooshian et al., 2000; Hong et al., . Our results confirm that CNNs learn to leverage this information directly from the satellite imagery and that it can notably improve the retrieval accuracy, which is in agreement with the findings from other precipitation retrievals that employ CNNs (Tang et al., 2018; Sadeghi et al., 2019; Gorooh et al., 2022; Sanò et al., 2018).